# The price of unfairness in linear bandits with biased feedback

**Solenne Gaucher**
Département de Mathématiques d'Orsay, Université Paris-Saclay, Orsay, France
solenne.gaucher@math.u-psud.fr

**Alexandra Carpentier**
Institut für Mathematik, Universität Potsdam, Potsdam, Germany
carpentier@uni-potsdam.de

**Christophe Giraud**
Département de Mathématiques d'Orsay, Université Paris-Saclay, Orsay, France
christophe.giraud@universite-paris-saclay.fr

## Abstract

In this paper, we study the problem of fair sequential decision making with biased linear bandit feedback. At each round, a player selects an action described by a covariate and by a sensitive attribute. The perceived reward is a linear combination of the covariates of the chosen action, but the player only observes a biased evaluation of this reward, depending on the sensitive attribute. To characterize the difficulty of this problem, we design a phased elimination algorithm that corrects the unfair evaluations, and establish upper bounds on its regret. We show that the worst-case regret is smaller than $\mathcal{O}(\kappa_*^{1/3} \log(T)^{1/3} T^{2/3})$, where $\kappa_*$ is an explicit geometrical constant characterizing the difficulty of bias estimation. We prove lower bounds on the worst-case regret for some sets of actions showing that this rate is tight up to a possible sub-logarithmic factor. We also derive gap-dependent upper bounds on the regret, and matching lower bounds for some problem instance. Interestingly, these results reveal a transition between a regime where the problem is as difficult as its unbiased counterpart, and a regime where it can be much harder.

## 1 Introduction

Artificial intelligence is increasingly used in a wide range of decision making scenarii with higher and higher stakes, with application in online advertisement [31], credit [3], health care [11], education [28] and job interviews [35], in the hope of improving accuracy and efficiency. Recent works have shown that the decisions made by algorithms can be dangerously biased against certain categories of people, and have endeavored to mitigate this behavior [22, 14, 6, 27]. Studies have underlined that the main cause of algorithmic unfairness is the presence of bias in the training set [27], which led to the development of methods aiming to guarantee the fairness of the algorithms. This paper, in lines with these works, addresses the problem of online decision making under biased feedback.

Linear bandits have become a very popular tool in online decision making problems, when side information on the actions is available in the form of covariates. In the present paper, we consider a variant of this problem, where the agent only has access to an unfair assessment of the action taken, that is systematically biased against a group of actions. For example, examiners may be prejudiced against people from a minority group, and give them lower grades; similarly, algorithms trained on

36th Conference on Neural Information Processing Systems (NeurIPS 2022).

biased data may produce unfair assessments of the credit risk of individuals belonging to a minority group. Note that not correcting biased evaluation can have adverse effects for all parties: on the one hand, actions disadvantaged by the evaluation mechanism will be unfairly discriminated against; on the other hand, the agent may spend his budget on an unfairly advantaged action that is actually sub-optimal. The problem of sequential decision making under biased feedback can be formalized as follows.

**Biased linear bandit problem**    A player is presented with a set of $k$ distinct actions characterized by covariates $x \in \mathcal{X} \subset \mathbb{R}^d$, and by known sensitive attributes $z_x \in \{-1, 1\}$ indicating the group of the action. For the sake of clarity, we consider here a two-group model (respectively privileged or discriminated against), and we defer to Appendix D discussions on how to extend this model and our algorithm to more than two groups. At each round $t \leq T$, the player chooses the action $x_t$ and receives an unobserved reward $x_t^\top \gamma^*$, where $\gamma^* \in \mathbb{R}^d$ is the regression parameter specifying the true value of the action. The regret of the player is given by

$$R_T = \mathbb{E}\Big[ \sum_{t \leq T} (x^* - x_t)^\top \gamma^* \Big], \quad \text{where} \quad x^* \in \underset{x \in \mathcal{X}}{\operatorname{argmax}}\, x^\top \gamma^*. \tag{1}$$

By contrast to the classical linear bandit, the player does not observe a noisy version of the unbiased reward $x_t^\top \gamma^*$. Instead, she observes an unfair evaluation $y_t$ of the value of the action $x_t^\top \gamma^*$, given by the following biased linear model:

$$y_t = x_t^\top \gamma^* + z_{x_t} \omega^* + \xi_t$$

where $\xi_t \overset{i.i.d}{\sim} \mathcal{N}(0, 1)$ is a noise term. The evaluation are systematically biased against a certain group: this unequal treatment of the groups is captured by the bias parameter $\omega^* \in \mathbb{R}$.

**Preliminary discussion**    The biased linear bandit is a variant of the linear bandit. By contrast, in the classical linear bandit model, the agent observes a noisy version of the reward. Obviously, applying directly an algorithm designed for linear bandit to biased linear bandits without correcting the evaluations would lead to a linear regret if the evaluation mechanism is prejudiced against the group of the best action in terms of reward, and if the best action in terms of feedback belongs to the advantaged group. To avoid this pitfall, one must estimate the bias in order to correct the evaluations. This implies a change in the exploration-exploitation trade-off, as exploration becomes more expensive. Indeed, in classical bandit problems, one can compare the rewards of two actions by repeatedly sampling them - or, to put it differently, one can find the best action by sampling only those actions that seem optimal. This does not hold in the biased linear bandit: if, at some point, the set of potentially optimal actions contains representatives from both groups, and does not span $\mathbb{R}^d$, one is forced to sample sub-optimal actions to estimate the bias and improve the estimation of the unbiased rewards. For this reason, classical algorithm for linear bandit that only sample actions considered as potentially optimal, such as OFUL [1] or Phase Elimination [24], can suffer linear regret. This underlines the necessity to ensure sufficient estimation of the bias parameter, even when it implies sampling sub-optimal actions.

## 1.1   Related work

Fairness in bandit problems has mostly been studied from the perspective of fair budget allocation between actions. This problem is motivated by the fact that classical bandit algorithms select sub-optimal actions only a vanishing fraction of the time, which may be undesirable in many situations. To mitigate this problem and guarantee diversity in the actions selected, some papers [4, 29, 9, 16, 43] have proposed new algorithms ensuring fairness of the selection frequency of each action. The framework studied in this paper is different: we consider here that the mechanism for observing the rewards is unfair, and we aim at correcting it in order to maximize a (fair) true cumulative reward.

In this work, we consider that the agent knows the sensible attributes, and that she can treat actions differently according to their sensible attributes, in order to correct the prejudice caused by the unfair bias in the evaluation. This situation falls into the awareness framework, by contrast to the unawareness one, where using the sensitive attributes is prohibited. Whether or not it is preferable to treat different groups differently remains a controversial question. While using sensitive attributes at the time of prediction is sometimes forbidden by law, some recent works have highlighted critical

issues related to unawareness. For example, empirical evidence [26] have shown that classification algorithms based on disparate learning processes use non-sensitive features correlated with the sensitive attribute as a proxy for the later. These empirical findings have recently been supported by theoretical results established in [15] in the case of demographic parity. Similarly, the authors of [7] study a problem of fair online learning, and show that some problems feasible in the awareness framework become infeasible in the unawareness one (such as no-regret learning under demographic parity constraints). These examples advocate for the use of the sensitive attribute, as it allows for better fairness guarantees while preventing unfair discrimination based on (possibly irrelevant) non-sensitive features correlated with the sensitive attribute. Without taking a position in this debate, we underline that, in practice, this attribute (gender or minority status) is often known to the decision-maker, and that its use is in some cases allowed or even encouraged (e.g. for affirmative action).

By contrast to a line of work on statistical fairness, the aim of our model is not to correct for the possibly unequal distribution of features $x$ and values $x^\top \gamma^*$ across the different groups. Our approach is instead related to causal fairness [18]: in the causal fairness framework, the dependencies between prediction, sensitive attributes and non-sensitive attributes are captured by a causal model. The goal is then to ensure that the sensitive attribute does not *directly* influence the prediction (in other words, that conditionally on selected resolving variables, the prediction is independent of the sensitive attribute). Here, the resolving variables may depend on the sensitive attribute in a manner that is considered as non-discriminatory. For example, one group may have, on average, more physical strength than the other one, and this skill can be considered as fair when it comes to recruit a piano mover. The biased linear model studied in this paper is a simple example of causal model with linear structural model equations $x = f(z, \xi')$ and $y = x^\top \gamma^* + z\omega^* + \xi$, where $\xi$ and $\xi'$ are noise terms: the covariates $x$ may depend on the sensitive attribute $z$, and the biased evaluation $y$ depends on both. In our work, we treat $x^\top \gamma^*$ as a fair evaluation of the value of action $x$, since it is independent of $z$ conditionally on the resolving variable $x$.

The biased linear model has been studied in the batch setting in [8], where the authors investigate the optimal trade-off between minimax risk and Demographic Parity. Detection of systematic bias, interpreted as a treatment effect, has been investigated in a batch setting in [18]. In [2], the authors consider a similar model, with unobserved sensitive attribute $z$ and known bias parameter $\omega^*$, under additional assumption that the sensitive attribute $z$ is independent from the covariate $x$. By contrast, we show that bias estimation is one of the main difficulties of the biased bandit problem.

The linear bandit with biased feedback can be viewed as a stochastic partial monitoring game. With the terminology of partial monitoring, the biased problem considered in the present paper is globally observable but not locally observable: in this case, the optimal worst-case regret rate typically increases as $\tilde{O}(T^{2/3})$. This regret rate is for example achieved in the related problem of partial linear monitoring with linear feedback and linear reward using an Information Directed Sampling algorithm [20]. However, the dependence of the regret on the geometry of the action set and on the dimension $d$ remains in most cases an open question [25, 5, 20]. In this paper, we characterize the geometry of the biased linear bandit problem, and we investigate dependence of the regret on the gaps.

## 1.2 Contribution and outline

In this paper, we introduce the linear bandit problem with biased feedback. We design a new algorithm based on optimal design for this problem. We derive an upper bound on the worst case regret of this algorithm of order $\kappa_*^{1/3} \log(T)^{1/3} T^{2/3}$ for large $T$, where $\kappa_*$ is an explicit constant depending on the geometry of the action set. We provide matching lower bounds on some problem instances, showing that the constant $\kappa_*$ characterizes the difficulty of the action set. Note that this regret is higher than the classical rates of order $\tilde{O}(dT^{1/2})$ obtained for $d$-dimensional linear bandits: this increase corresponds to the price to pay for debiasing the unfair evaluations.

We also characterize the gap-depend regret, showing that it is of order $(d/\Delta_{\min} \vee \kappa(\Delta)/\Delta_{\neq}^2) \log(T)$, where $\Delta_{\min}$ is the minimum gap, $\Delta_{\neq}$ is the gap between the best actions of the two groups, and $\kappa(\Delta)$ corresponds to the minimum regret to pay for estimating the bias with a given variance. This bound underlines the relative difficulties of the $d$-dimensional linear bandit and of the bias estimation. When $d/\Delta_{\min} \geq \kappa(\Delta)/\Delta_{\neq}^2$, i.e. when one group contains all near-optimal actions, the difficulty is dominated by that of the corresponding linear bandit problem. When both groups contain near-optimal actions, and $d/\Delta_{\min} \leq \kappa(\Delta)/\Delta_{\neq}^2$, the regret corresponds to the price of debiasing the rewards.

The rest of the paper is organized as follows. In Section 2, we present the FAIR PHASED ELIMI-NATION algorithm: we first discuss parameter estimation in Section 2.1, before presenting a sketch of the algorithm in Section 2.2 (a detailed version of this algorithm is provided in Appendix B). Then, in Section 3, we establish an upper bound on its worst-case regret. In Section 4, we derive a gap-dependent upper bound on the regret of our algorithm. In Section 5, we establish lower bounds on some action sets for both the worst-case and the gap-dependent regret, showing that these rates are sharp respectively up to a sub-logarithmic factor and an absolute multiplicative constant. Additional discussions on the geometry of bias estimation are postponed to Appendix A.

### 1.3 Notations and additional assumptions

We assume that all covariates $x \in \mathcal{X}$ are distinct, which implies that the group $z_x$ of action $x$ is well defined. We also assume that no group is empty, that the set $\{\begin{pmatrix} x \\ z_x \end{pmatrix} : x \in \mathcal{X}\}$ spans $\mathbb{R}^{d+1}$ (which guarantees identifiability of the parameters), and that the rewards are bounded: $\max_{x \in \mathcal{X}} |x^\top \gamma^*| \leq 1$.

When necessary, we underline the dependence of the regret on the parameter $\theta$ by denoting it $R_T^\theta$. We denote by $a_x = \begin{pmatrix} x \\ z_x \end{pmatrix}$ the vector describing an action and its group, by $\theta^* = \begin{pmatrix} \gamma^* \\ \omega^* \end{pmatrix} \in \mathbb{R}^{d+1}$ the unknown parameter, and by $\mathcal{A} = \{a_x : x \in \mathcal{X}\}$ the set of actions and of corresponding sensitive attributes. We denote by $\Delta = (\Delta_x)_{x \in \mathcal{X}}$ the vector of gaps $\Delta_x = \max_{x' \in \mathcal{X}} (x' - x)^\top \gamma^*$, and by $\mathcal{C}(\mathcal{X}) = \{\gamma \in \mathbb{R}^d : \forall x \in \mathcal{X}, |x^\top \gamma| \leq 1\}$ the set of admissible parameters. Note that for all $x \in \mathcal{C}(\mathcal{X}), \Delta_x \leq 2$. For $i \leq d+1$, let $e_i$ be the $i$-th vector of the canonical basis of $\mathbb{R}^{d+1}$, and for any matrix $M$, let $M^+$ be a generalized inverse of $M$. We denote by $\mathcal{P}^\mathcal{X}$ the set of probability measures on $\mathcal{X}$, and $\mathcal{M}^\mathcal{X} = \{\mu : \mathcal{X} \mapsto \mathbb{R}_+\}$. For any $\mu \in \mathcal{P}^\mathcal{X}$ or $\mu \in \mathcal{M}^\mathcal{X}$, we denote $V(\mu) = \sum_{x \in \mathcal{X}} \mu(x) a_x a_x^\top$ the covariance matrix corresponding to this allocation. For $u \in \mathbb{R}^{d+1}$ (resp. $\mathcal{U} \in \mathbb{R}^{d+1}$), we denote by $\mathcal{P}_u^\mathcal{X}$ (resp. $\mathcal{M}_u^\mathcal{X}$) the measures $\mu$ in $\mathcal{P}^\mathcal{X}$ (resp. in $\mathcal{M}^\mathcal{X}$) such that $u \in \text{Range}(V(\mu))$. For $\mathcal{U} \subset \mathbb{R}^{d+1}$, we denote by $\mathcal{P}_\mathcal{U}^\mathcal{X}$ (resp. $\mathcal{M}_\mathcal{U}^\mathcal{X}$) the measures $\mu$ such that $\mu \in \mathcal{P}_u^\mathcal{X}$ (resp. $\mathcal{M}_u^\mathcal{X}$) for all $u \in \mathcal{U}$.

## 2 Fair Phased Elimination algorithm

The Fair Phased Elimination algorithm belongs to the category of sequential elimination algorithms. Classical sequential elimination algorithms typically proceed by phases, indexed by $l = 1, 2, \ldots$. At phase $l$, these algorithms consider a set of potentially optimal actions $\mathcal{X}_l$. The rewards of all actions $x \in \mathcal{X}_l$ are then estimated with a given precision $O(\epsilon_l)$, typically chosen as $\epsilon_l = 2^{2-l}$, by sampling actions in $\mathcal{X}_l$. Actions sub-optimal by a gap larger than the precision level are then removed from the set $\mathcal{X}_{l+1}$ of potentially optimal actions for the phase $l + 1$.

As underlined previously, classical sequential elimination algorithms may suffer linear regret in the biased linear bandit problem if actions allowing to estimate the bias are discarded by the algorithm before the best group is identified (this happens for example if at a phase $l$, less that $d + 1$ action remains, with at least one action in each group). To mitigate this problem, we first estimate the biased evaluations of the potentially optimal actions, using ordinary least squares estimation. We then debias the estimations using an estimator for the bias relying on independent observations, which may be obtained by sampling sub-optimal actions. Before presenting the algorithm, let us discuss the estimation of the evaluations and of the bias parameter.

### 2.1 Optimal design for parameter estimation in the biased linear bandit

**G-optimal design for biased evaluation estimation**  As in the Phased Elimination algorithm [24], we rely on G-optimal design to estimate the biased evaluations $a_x^\top \theta^*$ with small error uniformly over a set of actions $\mathcal{X}_l$. More precisely, for a given set of potentially optimal actions $\mathcal{X}_l$, we compute the G-optimal design solution to the problem

$$\underset{\pi \in \mathcal{P}_{\mathcal{X}_l}^{\mathcal{X}_l}}{\text{minimize}} \max_{x \in \mathcal{X}_l} a_x^\top (V(\pi))^+ a_x . \qquad \text{(G-optimal design)} \qquad (2)$$

This can be done using polynomial-time algorithms, relying for example on interior points method [40], or on mixed integer second-order cone programming [37]. The celebrated General Equivalence theorem of Kiefer [19] and Pukelsheim [33] states that the value of Equation (2) is bounded by $d + 1$.

Let $\pi^*$ denote any design solution to the G-optimal design problem (2), and let $\widehat{\theta}$ denote the ordinary least square estimator obtained by sampling each action $x \in \mathcal{X}_l$ exactly $\lceil n\pi^*(x) \rceil$ times for a given $n > 0$. Then, for all $x \in \mathcal{X}_l$, the General Equivalence theorem implies that the variance of the estimate $a_x^\top \widehat{\theta}$ is smaller than $(d+1)/n$. Moreover, the G-optimal design $\pi^*$ can be chosen so that it is supported by at most $(d+1)(d+2)/2$ points, so the total number of samples is at most $n + (d+1)(d+2)/2$.

**$\Delta$-optimal design for bias evaluation** In this paragraph, we introduce the $\Delta$-optimal design, which is discussed in greater depth in Appendix A. $\Delta$-optimal design aims at estimating a parameter with a given accuracy and with minimal regret. Similar ideas have recently been used in [42] to solve classical linear bandit problems. To estimate the bias parameter $\omega^*$, we use the estimator $\widehat{\omega} = e_{d+1}^\top \widehat{\theta}$, where $\widehat{\theta}$ is the ordinary least square estimator for the full parameter $\theta^*$. Now, if we sample each action $x \in \mathcal{X}$ exactly $\mu(x)$ time, the variance of $\widehat{\omega}$ is equal to $e_{d+1}^\top V(\mu)^+ e_{d+1}$. Given the vector of gaps $\Delta$, the design $\mu$ minimizing the regret of this exploration phase, while ensuring that the variance of $\widehat{\omega}$ is smaller than 1, is solution of the problem

$$\underset{\mu \in \mathcal{M}_{\mathcal{X}}^{e_{d+1}}}{\text{minimize}} \sum_x \mu(x)\Delta_x \quad \text{such that} \quad e_{d+1}^\top V(\mu)^+ e_{d+1} \leq 1. \quad \text{($\Delta$-optimal design)} \quad (3)$$

In the following, we denote $\mu^\Delta$ a minimizer of (3), and $\kappa(\Delta) = \sum_{x \in \mathcal{X}} \mu^\Delta(x)\Delta_x$. Lemma 9 in Appendix A explains how to compute the design $\mu^\Delta$ in polynomial time by adapting tools from $c$-optimal design. This lemma also shows that the support of $\mu^\Delta$ can be chosen to be of cardinality at most $d + 1$. Then, choosing each action exactly $\lceil n\mu^\Delta(x) \rceil$ times for a given $n > 0$ allows us to estimate the bias with variance lower than $n^{-1}$ and a regret no larger than $n\kappa(\Delta) + 2(d+1)$. Obviously, we do not know the gap vector $\Delta$ beforehand, so we must estimate it as we go.

## 2.2 Outline of the Fair Phased Elimination algorithm

The Fair Phased Elimination algorithm, sketched in Algorithm 3, relies on the following key ideas. First, note that within a group, the order of the true rewards and of the biased evaluations are the same. Hence, within a group, we can use classical algorithms for linear bandits to choose the actions and estimate the biased evaluations with a controlled within-group regret: this is done using **G-exploration and elimination**. Second, to compare actions belonging to different groups, we independently estimate the bias parameter $\omega^*$, using **$\Delta$-exploration and elimination**. Finally, we underline that bias estimation may require to sample very sub-optimal actions. Therefore, it can be overly costly to estimate the bias up to the precision level required to identify the best group. To prevent this, we use a **stopping criterion**.

**G-exploration and elimination** At each phase $l = 1, 2, ...$, we keep two sets of potentially optimal actions belonging to the groups $+1$ and $-1$, denoted respectively $\mathcal{X}_l^{(+1)}$ and $\mathcal{X}_l^{(-1)}$. If we have not identified the group containing the best action, we run a G-EXP-ELIM routine 1 on each set $\mathcal{X}_l^{(z)}$ for $z = 1$ and $z = -1$. This routine samples actions according to a rounded G-optimal design on $\mathcal{X}_l^{(z)}$, with a total number of observations chosen so that the biased evaluations of all actions in $\mathcal{X}_l^{(z)}$ are known with an error at most $\epsilon_l$. The set $\mathcal{X}_{l+1}^{(z)}$ is obtained by removing from $\mathcal{X}_l^{(z)}$ actions whose estimated evaluations are sub-optimal by a gap larger than $3\epsilon_l$, compared to the empirical best action in the group. This allows to ensure that only actions sub-optimal by a gap $\mathcal{O}(\epsilon_l)$ remain in $\mathcal{X}_{l+1}^{(z)}$, and to estimate the gap vector $\Delta$ with a precision sufficient for $\Delta$-optimal estimation.

If the group containing the best action has been identified, we discard the other group, and run a G-EXP-ELIM routine 1 on the set of potentially optimal actions in this group.

**$\Delta$-exploration and elimination** If the group of the best action has not been found before phase $l$, we run the $\Delta$-EXP-ELIM routine 2. More precisely, relying on a previous estimate $\widehat{\Delta}^l$ of the gap vector $\Delta$, we compute the $\widehat{\Delta}^l$-optimal design $\widehat{\mu}$. We then estimate the bias using actions sampled according to a rounded version of this design, with a total number of observations chosen so that the error of bias estimation is smaller than $\epsilon_l$, and use it to debias the reward estimation. If the debiased evaluation of the best action of each group are separated by a gap larger than $4\epsilon_l$, we consider that the best group is the one containing the empirical best action in terms of biased evaluation, and we discard the other group.

---
**Routine 1** G-EXP-ELIM $(\mathcal{X}, n, \epsilon)$

---
1: Compute G-optimal design $\pi$ solution of (2) on $\mathcal{X}$, with $|\operatorname{supp}(\pi)| \leq {(d+1)(d+2)}/{2}$
2: Sample $\lceil n\pi(x) \rceil$ times each action $a_x$ for $x \in \mathcal{X}$      ▷ G-optimal parameter estimation
3: Compute the ordinary least square estimator $\widehat{\theta}$
4: $\mathcal{X}' \leftarrow \left\{ x \in \mathcal{X} : \max_{x' \in \mathcal{X}} (x' - x)^\top \widehat{\theta} \leq 3\epsilon \right\}$      ▷ Suboptimal actions elimination
5: **return** $\widehat{\theta}$ and $\mathcal{X}'$

---

If we cannot find the best group, we rely on estimates of the bias and of the biased evaluations obtained during the previous round to update the estimate of the gap vector $\widehat{\Delta}^{l+1}$.

---
**Routine 2** $\Delta$-EXP-ELIM $(\mathcal{X}, (\mathcal{X}^{(z)}, \widehat{\theta}^{(z)})_{z \in \{-1,1\}}, \widehat{\Delta}, n, \epsilon)$

---
1: Compute $\widehat{\Delta}$-optimal design $(\hat{\mu}, \kappa(\hat{\Delta}))$ solution of (3) on $\mathcal{X}$, with $|\operatorname{supp}(\hat{\mu})| \leq d + 1$
2: Sample $\lceil n\hat{\mu}(x) \rceil$ times each action $a_x$ for $x \in \mathcal{X}$      ▷ $\widehat{\Delta}$-optimal bias estimation
3: Compute $\widehat{\omega} = e_{d+1}^\top \widehat{\theta}$, where $\widehat{\theta}$ is the ordinary least square estimator
4: **for** $z \in \{-1, 1\}$ and $x \in \mathcal{X}^{(z)}$ **do** $\widehat{m}_x \leftarrow a_x^\top \widehat{\theta}^{(z)} - z\widehat{\omega}$      ▷ Debiased rewards estimation
5: **if** $\exists z \in \{-1, 1\}$ such that $\max_{x \in \mathcal{X}^{(z)}} \widehat{m}_x \geq \max_{x \in \mathcal{X}^{(-z)}} \widehat{m}_x + 4\epsilon$ **then** $\mathcal{Z} \leftarrow \{z\}$    ▷ Group elimination
6: **else** $\widehat{\Delta}_x \leftarrow 2 \wedge (\max_{x' \in \mathcal{X}^{(-1)} \cup \mathcal{X}^{(1)}} \widehat{m}_{x'} - \widehat{m}_x + 4\epsilon)$ for all $x \in \mathcal{X}^{(-1)} \cup \mathcal{X}^{(1)}$
7: **return** $\mathcal{Z}$ and $\widehat{\Delta}$

---

**Stopping criterion**    As underlined previously, the $\Delta$-EXP-ELIM routine samples actions that can be very sub-optimal. As a consequence, when the gap between the best two actions of each group is small, finding the best group can be overly costly in terms of regret. To prevent this, if the best group has not been found at stage $l$ fulfilling $\epsilon_l \leq \left( \kappa(\widehat{\Delta}^l) \log(T)/T \right)^{1/3}$, the bias estimation is stopped and the empirical best action in $\mathcal{X}_{l+1}^{(1)} \cup \mathcal{X}_{l+1}^{(-1)}$ is sampled for the remaining time (see Algorithm 3)

---
**Algorithm 3** FAIR PHASED ELIMINATION (sketched)

---
1: **input:** $\delta, T, \mathcal{X}, k = |\mathcal{X}|, \epsilon_l = 2^{2-l}$ for $l \geq 1$
2: **initialize:** $\mathcal{X}_1^{(+1)} \leftarrow \{x : z_x = 1\}$, $\mathcal{X}_1^{(-1)} \leftarrow \{x : z_x = -1\}$,
3:      $\mathcal{Z}_1 \leftarrow \{-1, +1\}$, $\widehat{\Delta}^1 \leftarrow (2, ..., 2)$, $l \leftarrow 0$
4: **while** the budget is not spent **do** $l \leftarrow l + 1$
5:      **for** $z \in \mathcal{Z}_l$ **do**
6:          $\left( \widehat{\theta}^{(z)}, \mathcal{X}_{l+1}^{(z)} \right) \leftarrow$ G-EXP-ELIM $\left( \mathcal{X}_l^{(z)}, \frac{2(d+1)}{\epsilon_l^2} \log\left( \frac{kl(l+1)}{\delta} \right), \epsilon_l \right)$
7:      **if** $\mathcal{Z}_l = \{-1, +1\}$ **then**
8:          **if** $\epsilon_l \leq \left( \kappa(\widehat{\Delta}^l) \log(T)/T \right)^{1/3}$ **then**      ▷ Stop bias estimation
9:            Sample best action in $\mathcal{X}_{l+1}^{(-1)} \cup \mathcal{X}_{l+1}^{(+1)}$ for the remaining time
10:         **else**
11:           $\left( \mathcal{Z}_{l+1}, \widehat{\Delta}^{l+1} \right) \leftarrow \Delta$-EXP-ELIM $\left( \mathcal{X}, \left( \mathcal{X}_{l+1}^{(z)}, \widehat{\theta}_l^{(z)} \right)_{z \in \{-1,1\}}, \widehat{\Delta}^l, \frac{2}{\epsilon_l^2} \log\left( \frac{l(l+1)}{\delta} \right), \epsilon_l \right)$

---

## 3   Upper bound on the worst-case regret of FAIR PHASED ELIMINATION

The regret of the FAIR PHASED ELIMINATION depends on the difficulty of estimating the bias parameter, captured by $\kappa(\Delta)$. Lemma 7 in Appendix A.6 shows that for all parameter $\gamma^* \in \mathcal{X}$, $\kappa(\Delta)$ is upper bounded by $2\kappa_*$, where $\kappa_*$ is the *minimal variance of the bias estimator* given by

$$\kappa_* = \min_{\pi \in \mathcal{P}_{e_{d+1}}^{\mathcal{X}}} e_{d+1}^\top \left( V(\pi) \right)^+ e_{d+1}.$$

The following theorem provides a bound on the worst case regret depending on $\kappa_*$. Proofs are postponed to Appendix C.2.

**Theorem 1.** *For the choice $\delta = T^{-1}$, there exists two numerical constants $C, C' > 0$ such that the following bound on the regret of the* FAIR PHASED ELIMINATION *algorithm 4 holds*

$$R_T \leq C \left( \kappa_*^{1/3} T^{2/3} \log(T)^{1/3} + (d \vee \kappa_*) \log(T) + d^2 + d\kappa_*^{-1/3} T^{1/3} \log(kT) \log(T)^{-1/3} \right)$$

$$\leq C' \kappa_*^{1/3} T^{2/3} \log(T)^{1/3} \quad for \quad T \geq \frac{((d \vee \kappa_*)^{3/2} \log(T)) \vee d^3}{\sqrt{\kappa_*}} \vee \frac{(d \log(kT))^3}{(\kappa_* \log(T))^2}.$$

In Section 5.1, we show that the upper bound obtained in Theorem 1 is sharp in some settings, up to the sub-logarithmic factor $\log(T)^{1/3}$.

Theorem 1 shows that the worst-case regret of the Fair Phased Elimination algorithm asymptotically grows as $C\kappa_*^{1/3} T^{2/3} \log(T)^{1/3}$. This worst-case regret rate is higher than the typical rate $Cd \log(T) T^{1/2}$ obtained under unbiased feedback on the rewards (see, e.g., [1]). This increase in the regret corresponds to the cost of learning from unfair evaluations. It is due to the fact that the algorithm may need to sample actions that are sub-optimal in order to estimate the bias parameter. Note that this rate $\widetilde{\mathcal{O}}(T^{2/3})$ is typical for globally observable bandit problems with partial linear monitoring, and can be obtained by applying results established in [20] for in the partial linear monitoring setting to the biased linear bandit problem.

By contrast to previous results, Theorem 1 characterizes precisely the dependence of the worst-case regret on the geometry of the action set. The relevant constant $\kappa_*$ is the minimal variance for estimating the bias, which appears when considering the related $c$-optimal design problem. While the connection between G-optimal design and the linear bandit problem has already been exploited, it is to the best of our knowledge the first time that $c$-optimal design is related to partial monitoring.

The constant $\kappa_*$ corresponds to the minimum number of samples required for estimating the bias with a variance equal to 1 (up to rounding issues). Intuitively, if the actions are very correlated with their sensitive attributes, more samples will be needed to estimate the bias with the same precision. This situation corresponds to cases where $\kappa_*$ is large, and leads to a higher regret. Lemma 1, illustrated in Figure 1, relates $\kappa_*$ to the margin between the two groups of actions.

**Lemma 1.** *$\kappa_*$ is the largest constant $\kappa \geq 0$ such that, there exists an hyperplane $\mathcal{H}$ containing zero and separating the two groups, and such that, the margin to $\mathcal{H}$ is at least $\sqrt{\kappa}-1/\sqrt{\kappa}+1$ times the maximum distance of all points to the hyperplane (see Figure 1). When no such hyperplane exists, then $\kappa_* = 1$.*

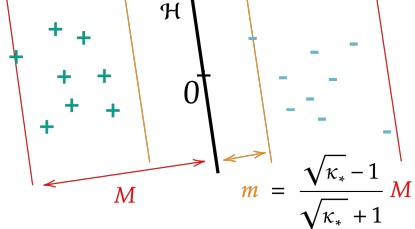
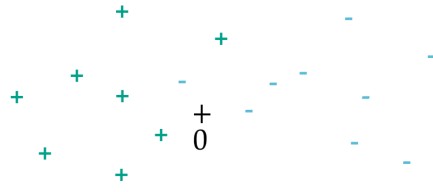

(a) The margin $m$ is equal to $\sqrt{\kappa_*}-1/\sqrt{\kappa_*}+1$ times the maximum distance $M$ of any action to the hyperplane.

(b) $\kappa_* = 1$: the groups cannot be separated by a hyperplane containing 0.

Figure 1: Interpretation of $\kappa_*$ in terms of separation of the groups.

Interestingly, Lemma 1 underlines that under reasonable assumptions, the constant $\kappa_*$ may not depend on the ambient dimension $d$, and it can even be equal to 1. By contrast, while the Information Directed Sampling algorithm can be applied to the biased linear bandit problem, the regret bounds established in [20] are of order $\alpha^{1/3} d^{1/2} T^{2/3} \log(kT)^{1/2}$, where $\alpha$ is a measure of the complexity of the action set called the worst-case alignment constant. Lemma 6 in Appendix A shows that $\alpha$ is equivalent to the minimal variance of the bias estimator $\kappa_*$. Hence, our bound improves over previous results by a factor $d^{1/2} \log(T)^{1/6} (\log(kT)/\log(T))^{1/2}$.

The gaps are not involved in the definition of the minimal variance of bias estimation $\kappa_*$. The reader may have expected to get, instead of $\kappa_*$, the minimax regret for estimating the bias

$$\widetilde{\kappa} = \max_{\gamma \in \mathcal{C}(\mathcal{X}), x' \in \mathcal{X}} \sum_{x \in \mathcal{X}} \widetilde{\mu}(x)(x' - x)^\top \gamma, \quad \text{where}$$

$$\widetilde{\mu} = \operatorname*{argmin}_{\mu} \max_{x' \in \mathcal{X}, \gamma \in \mathcal{C}(\mathcal{X})} \sum_{x \in \mathcal{X}} \mu(x)(x' - x)^\top \gamma, \text{ such that } \mu \in \mathcal{M}_{e_{d+1}}^{\mathcal{X}} \text{ and } e_{d+1}^\top V(\mu)^+ e_{d+1} \leq 1.$$

Next lemma shows that $\kappa_*$ and $\widetilde{\kappa}$ are in equivalent up to a factor 2. We refer the interested reader to Appendix A, where further discussions on the geometry of bias estimation are postponed, due to space constraints.

**Lemma 2.** $\widetilde{\kappa}/2 \leq \kappa_* \leq 2\widetilde{\kappa}$.

## 4 Upper bound on the gap-depend regret of FAIR PHASED ELIMINATION

In this section, we provide an upper bound on the worst-case regret that depends on the gap between the two best actions, and on the gap between the best actions of the two groups. Compared to instance-dependent bounds, established in the linear bandit problem in [23, 21], gap-dependent bounds characterize the dependence of the regret on a small number of parameters. They are typically less sharp than instance-dependent bounds, but allow to better highlight the influence of the parameters on the difficulty of the problem. The bound established in the following theorem relates the difficulty of the biased linear bandit to that of bias estimation, and to that of the corresponding $d$-dimensional linear bandit. Proofs are postponed to Appendix C.2.

**Theorem 2.** *Assume that* $x^* \in \operatorname{argmax}_{x \in \mathcal{X}} x^\top \gamma^*$ *is unique. Then, there exists two numerical constants* $C, C' > 0$ *such that, for the choice* $\delta = T^{-1}$, *the following bound on the regret of the* FAIR PHASED ELIMINATION *algorithm 4 holds*

$$
\begin{aligned}
R_T &\leq C \left( \left( \frac{d}{\Delta_{\min}} \vee \frac{\kappa(\Delta \vee \Delta_{\neq} \vee \varepsilon_T)}{\Delta_{\neq}^2} \right) \log(T) + d^2 + \frac{d}{\Delta_{\min}} \log(k) \right) \\
&\leq C' \left( \frac{d}{\Delta_{\min}} \vee \frac{\kappa(\Delta \vee \Delta_{\neq} \vee \varepsilon_T)}{\Delta_{\neq}^2} \right) \log(T) \quad \text{for} \quad T \geq k \vee e^{d\Delta_{\min}}
\end{aligned}
$$

*where* $\Delta_{\min} = \min_{x \in \mathcal{X} \setminus x^*} \Delta_x$, $\Delta_{\neq} = \min_{x \in \mathcal{X}: z_x = -z_{x^*}} \Delta_x$, *and* $\varepsilon_T = \left(\kappa_* \log(T)/T\right)^{1/3}$.

The term $d/\Delta_{\min} \vee \kappa(\Delta \vee \Delta_{\neq} \vee \varepsilon_T)/\Delta_{\neq}^2$ highlights the two sources of difficulty of the problem. On the one hand, the term $d/\Delta_{\min}$ is unavoidable: even if the algorithm knew beforehand the group containing the best action, it would still need to play a game of $d$-dimensional linear bandits in this group, and suffer, in the worst-case, the corresponding gap-dependent regret [1]. Note that lower bounds on gap-depend regret of classical linear bandits follow from considering a setting with one near-optimal action with gap $\Delta_{\min}$ in each of the $d$ dimensions. Then, any algorithm needs to explore each dimension up to $\Delta_{\min}^{-2} \log(T)$ times in order to find the best action, but can do so by choosing the near-optimal actions, thus having a regret $\Delta_{\min}^{-1} \log(T)$ in each direction. By contrast, the term $\kappa(\Delta \vee \Delta_{\neq} \vee \varepsilon_T)/\Delta_{\neq}^2$ is characteristic of the biased linear bandit problem: it is due to the fact that the algorithm may need to sample very sub-optimal actions in order to find the group containing the best action. Indeed, to identify this group, one must estimate the bias with a precision $\Delta_{\neq}$, i.e. sample sub-optimal actions with average regret $\kappa(\Delta)$ approximately $\Delta_{\neq}^{-2} \log(T)$ times.

When $d/\Delta_{\min} \leq \kappa(\Delta \vee \Delta_{\neq} \vee \varepsilon_T)/\Delta_{\neq}^2$, the regret corresponds to the regret of this bias estimation phase. In other words, when both groups contain near-optimal actions, the difficulty of the problem is dominated by the price to pay for debiasing the unfair evaluations. Interestingly, when $d/\Delta_{\min} > \kappa(\Delta \vee \Delta_{\neq} \vee \varepsilon_T)/\Delta_{\neq}^2$, the difficulty of the linear bandit with systematic bias is dominated by that of the classical $d$-linear bandit. In this case, the algorithm is able to find the group containing the best action, and the problem reduces to a linear bandit in dimension $d$. Thus, the linear bandit with systematic bias is a non trivial example of a globally observable game that can be locally observable around the best action.

Finally, we underline that the magnitude of the bias does not appear in the regret: intuitively, no matter its magnitude, the algorithm always need to estimate it up to the same precision (of order $\Delta_{\neq}$) in order to find the best group and to be optimal in terms of gap-depend regret. This indicates that our algorithm is robust against important discriminations in the evaluation mechanism.

# 5 Lower bounds on the regret

In this section, we derive lower bounds on the worst-case regret and the gap-dependent regret that respectively match the upper bounds established in Theorems 1 and 2 up to sub-logarithmic factors or numerical constants.

## 5.1 Lower bound on the worst-case regret

Theorems 1 and 2 underline the dependence of the regret on the geometry of the action set. Before stating our result, we begin by introducing the notion of $\kappa_*$-correlated action set.

**Definition 1** ($\kappa_*$-correlated action set). *For $\kappa_* \geq 1$, a set of actions $\mathcal{A}$ is $\kappa_*$-correlated if $\mathcal{A} \in \mathbf{A}_{\kappa_*,d}$, where*

$$
\mathbf{A}_{\kappa_*,d} = \left\{
\begin{array}{l}
\mathcal{A} = \{a_1,...,a_k\} \subset \left(\mathbb{R}^d \times \{-1,+1\}\right)^k : \\[2mm]
k \in \mathbb{N}^*,\ \min_{\pi \in \mathcal{P}^{\mathcal{A}}_{e_{d+1}}} \left\{ e_{d+1}^\top \Big( \sum_{a \in \mathcal{A}} \pi(a) a a^\top \Big)^+ e_{d+1} \right\} \geq \kappa_*
\end{array}
\right\}
$$

*is the set of actions sets such that the minimal variance of the bias estimator is larger than $\kappa_*$.*

In the following theorem, we establish a lower bound on the regret valid for all $\kappa_* \geq 1$ by designing $\kappa_*$-correlated sets of actions $\mathcal{A} \in \mathbf{A}_{\kappa_*,d}$, and obtaining lower bounds on the regret of any algorithm on these sets of actions.

**Theorem 3.** *Let $\kappa_* \geq 1$, $d \geq 2$ and $T \geq 4^3 \kappa_*$. There exists an action set $\mathcal{A} \in \mathbf{A}_{\kappa_*,d}$ such that for any algorithm, there exists a bandit problem with parameter $\theta_T \in \mathbb{R}^{d+1}$ such that the regret of this algorithm on the problem characterized by $\theta_T$ satisfies $R_T^{\theta_T} \geq \kappa_*^{1/3} T^{2/3}/8e$.*

Previous lower bounds on the regret of linear bandits with partial monitoring, established in [20], state that the regret must be at least $c_{\mathcal{A}} T^{2/3}$ for some parameter $\theta_T \in \mathbb{R}^{d+1}$, where $c_{\mathcal{A}} > 0$ is a constant depending (not explicitly) on $\mathcal{A}$. By contrast, Theorem 3 provides an explicit characterization of the dependence of the regret rate on the geometry of the problem, which matches the upper bound of Theorem 1 up to a sub-logarithmic factor. Note that the assumption $d \geq 2$ is necessary here: if $d = 1$, there are at most two potentially optimal actions (namely, $\max\{x : x \in \mathcal{X}\}$ and $\min\{x : x \in \mathcal{X}\}$). Then, the problem becomes locally observable, and regret of order $\widetilde{O}(T^{1/2})$ can be achieved [20].

## 5.2 Lower bound on the gap-dependent regret

We now present a lower bound on the gap-dependent regret. More precisely, for given values of $\Delta_{\min}$ and $\Delta_{\neq}$, we establish a lower bound on the worst case regret among parameters $\theta$ verifying $\Delta_{\min} \leq \min_{x \in \mathcal{X} \setminus x^*} \Delta_x$, and $\Delta_{\neq} \leq \min_{x \in \mathcal{X}: z_x = -z_{x^*}} \Delta_x$. Before stating formally the result, let us define the corresponding parameter set. For an action set $\mathcal{A} \in \mathbf{A}_{\kappa_*,d}$, and for $(\Delta_{\min}, \Delta_{\neq}) \in (0,1)^2$ such that $\Delta_{\min} \leq \Delta_{\neq}$, we denote

$$
\Theta^{\mathcal{A}}_{\Delta_{\min},\Delta_{\neq}} = \left\{
\begin{array}{l}
\theta = \binom{\gamma}{\omega} :\ \gamma \in \mathcal{C}(\mathcal{X}),\ \exists!\ \binom{x^*}{z_{x^*}} \in \mathrm{argmax}_{\binom{x}{z_x} \in \mathcal{A}} \{x^\top \gamma\}, \\[2mm]
\forall \binom{x'}{z_{x'}} \in \mathcal{A} \text{ such that } x' \neq x^*, (x^* - x')^\top \gamma \geq \Delta_{\min}, \\[2mm]
\forall \binom{x'}{z_{x'}} \in \mathcal{A} \text{ such that } z_{x'} \neq z_{x^*}, (x^* - x')^\top \gamma \geq \Delta_{\neq}
\end{array}
\right\}
$$

the set of parameters with minimum gap $\Delta_{\min}$, and minimum between-group-gap $\Delta_{\neq}$.

The upper bounds established in Theorem 2 underline the dependence of the gap-dependent regret on the minimal regret $\kappa(\Delta)$ for estimating the bias. Before stating our results, we define a class of problems $\Theta^{\mathcal{A}}_{\Delta_{\min},\Delta_{\neq},\kappa}$ such that $\kappa(\Delta) \leq \kappa$. For a parameter $\gamma \in \mathcal{C}(\mathcal{X})$, let us denote $\Delta(\gamma)_x = \max_{x' \in \mathcal{X}} (x' - x)^\top \gamma$, and $\Delta(\gamma) = (\Delta(\gamma)_x)_{x \in \mathcal{X}}$. Moreover, for a given set $\mathcal{A}$, let us denote

$$
\Theta^{\mathcal{A}}_{\Delta_{\min},\Delta_{\neq},\kappa} = \Theta^{\mathcal{A}}_{\Delta_{\min},\Delta_{\neq}} \cap \left\{ \theta = \binom{\gamma}{\omega} :\ \gamma \in \mathcal{C}(\mathcal{X}),\ \kappa(\Delta(\gamma)) \leq \kappa \right\}.
$$

**Theorem 4.** *For all $\kappa \geq 2$ and all $d \geq 4$, there exists a set of actions $\mathcal{A} \in \mathbb{R}^{d+1}$ such that for all $(\Delta_{\min}, \Delta_{\neq}) \in (0, 1/8)^2$ with $\Delta_{\min} \leq \Delta_{\neq}$,*

$$\liminf_{T \to \infty} \sup_{\theta \in \Theta^{\mathcal{A}}_{\Delta_{\min}, \Delta_{\neq}, \kappa}} \frac{R^{\theta}_T}{\log(T)} \geq \left[ \frac{d}{10\Delta_{\min}} \right] \vee \left[ \frac{\kappa + 2}{8\Delta^2_{\neq}} \right]. \tag{4}$$

Theorem 4 shows that for some action sets $\mathcal{A}$, the gap-depend regret of the FAIR PHASED ELIM-INATION algorithm is asymptotically optimal up to a numerical constant. Note that the assumption $d \geq 4$ is necessary in our proof to design an action set $\mathcal{A}$ such that Equation (4) holds for all $\Delta_{\min}, \Delta_{\neq} \in (0, 1/8)$. On the other hand, as discussed in Appendix C.6, for $d \geq 2$, for all $\Delta_{\min}, \Delta_{\neq} \in (0, 1/8)$, we can show that there exists action sets $\mathcal{A}$ and $\theta \in \Theta^{\mathcal{A}}_{\Delta_{\min}, \Delta_{\neq}}$ such that the lower bound in Equation (4) still holds, by considering separately the cases $d/\Delta_{\min} > \kappa/\Delta^2_{\neq}$ and $d/\Delta_{\min} \leq \kappa/\Delta^2_{\neq}$.

## 6 Conclusion

In this paper, we addressed the problem of online decision making under biased bandit feedback. We designed a new algorithm based on $\Delta$- and G-optimal design, and obtained worst-case and gap-dependent upper bounds on its regret. We obtained lower bounds on the regret for some problem instances showing that these rates are tight up to sub-logarithmic factors in some settings. These rates highlight two behaviors: on the one hand, the worst case rate $\mathcal{O}(\kappa_*^{1/3} \log(T)^{1/3} T^{2/3})$ highlights the cost induced by the biased feedback, and the need to select sub-optimal actions in order to debias it. On the other hand, the gap-dependent bound shows that for some instance, the problem can be locally observable around the best action: then, the difficulty of the problem is dominated by the difficulty of the corresponding linear bandit problem, and is no more difficult than this problem. When this is not the case, the regret scales as $\kappa(\Delta)\Delta^{-2}_{\neq} \log(T)$, where $\Delta_{\neq}$ is the gap between the best actions of the two groups, and $\kappa(\Delta)$ is the minimum regret for estimating the bias with a given precision. In Appendix D, we discuss the extension of the biased linear model and of the Fair Phased Elimination algorithm to multiple groups with different biases. This work paves the way for studying other bandit models with unfair feedback, considering for example continuous, multi-dimensional sensitive attributes.

## Broader impact

In this work, we propose a model for sequential decision making under biased feedback. Our goal is primarily to provide a good strategy for sequential learning in an unfair environment, and to characterize the difficulty of this problem by bounding the regret. On the one hand, our results reveal that maximizing the fair rewards instead of unfair evaluations may be more difficult in terms of regret, which may discourage practitioners from correcting unfair feedbacks. On the other hand, we believe that as fairness is an important long-term key objective, rather than discouraging the practitioner, it will inform them to better plan the adaptation of their methods toward this aim.

## Acknowledgements.

The authors would like to thank Evgenii Chzhen and Nicolas Verzelen for their valuable discussions and suggestions.

The work of A. Carpentier is partially supported by the Deutsche Forschungsgemeinschaft (DFG) Emmy Noether grant MuSyAD (CA 1488/1-1), by the DFG - 314838170, GRK 2297 MathCoRe, by the FG DFG, by the DFG CRC 1294 'Data Assimilation', Project A03, by the Forschungsgruppe FOR 5381 "Mathematical Statistics in the Information Age - Statistical Efficiency and Computational Tractability", Project TP 02, by the Agence Nationale de la Recherche (ANR) and the DFG on the French-German PRCI ANR ASCAI CA 1488/4-1 "Aktive und Batch-Segmentierung, Clustering und Seriation: Grundlagen der KI" and by the UFADFH through the French-German Doktorandenkolleg CDFA 01-18 and by the SFI Sachsen-Anhalt for the project RE-BCI. Christophe Giraud received partial support by grant ANR-19-CHIA-0021-01 ("BiSCottE", Agence Nationale de la Recherche) and by the ANR and the DFG on the French-German PRCI ANR-21-CE23-0035 (ASCAI).

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
