# Appendix

The Appendix is organized as follows. In Section A, we further discuss the geometry of bias estimation, and provide additional results on the constants $\kappa_*$ and $\kappa(\Delta)$. Then, we provide in Section B a detailed version of the FAIR PHASED ELIMINATION algorithm 3. In Section C, we prove the main results of this paper. Finally, in Section D, we discuss the extension of the biaised linear bandits to more than 2 groups.

## A    On the geometry of bias estimation

We begin in Section A.1 by highlighting the relationship of the constant $\kappa_*$ with the problem of $e_{d+1}$-*optimal design*. Then, in Section A.2, we show that the geometrical constant $\kappa_*$ can be expressed in terms of separation of the two groups. In Section A.3 and Section A.4, we relate $\kappa_*$ to classical geometrical measures of the difficulty of a set of actions such as the condition.ing number and the worst-case alignment constant of [20]. In Section A.5, we show that $\kappa_*$ is equivalent to the variance of the optimal design for estimation the bias against the worst parameter $\theta^*$. In Section A.6, we provide further results on $\kappa(\Delta)$, the $\Delta$-optimal regret for estimation the bias with variance 1 when the gap vector is $\Delta$. Finally, in Section A.7, we propose guidance for computing the G-optimal and $\Delta$-optimal designs.

### A.1    Bias estimation as a $e_{d+1}$-optimal design problem

Recall that $\kappa_*$ is the minimal variance of the bias estimator related to the problem of $e_{d+1}$-optimal design.

$e_{d+1}$-**optimal design**    Optimal design theory addresses the following problem: a scientist must design a set of $n$ experiments $\{x_1, ..., x_n\} \in \mathcal{X}^n$ so as to estimate at best a parameter of interest, where each experiment $x \in \mathcal{X}$ corresponds to a point $a_x \in \mathbb{R}^{d+1}$. The aim of the scientist is to choose a design, i.e. a function $\mu : \mathcal{X} \mapsto \mathbb{N}$ indicating the budget $\mu(x)$ to be allocated to each experiment $x \in \mathcal{X}$. Each experiment $x$ is then repeated exactly $\mu(x)$ times, and the corresponding observations $y_{x,1}, ..., y_{x,\mu(x)}$ are collected for each $x \in \mathcal{X}$. The law of the observations corresponding to experiment $x$ at point $a_x$ is given by

$$y_{x,i} = a_x^\top \theta^* + \xi_{x,i},$$

where $\xi_{x,i} \sim \mathcal{N}(0, 1)$ are independent noise terms, and $\theta^* \in \mathbb{R}^{d+1}$ is an unknown parameter. The aim of the scientist is to choose the design $\mu$ so as to best estimate (some features of) the parameter $\theta^*$, under a constraint on the total number of experiments $\sum_{x \in \mathcal{X}} \mu(x) \leq n$ for some $n \in \mathbb{N}$.

Different criteria can be used to characterize the optimality of a design $\mu$. For example, one may need to estimate the full parameter $\theta^*$, in order to predict the outcomes of the experiments $x \in \mathcal{X}$ with a small uniform error: this leads to the G-optimal design problem (2). Alternatively, for $c$ a vector in $\mathbb{R}^{d+1}$, one may aim at finding the best design $\mu \in \mathcal{N}^{\mathcal{X}}$ for estimating the scalar product $c^\top \theta^*$ under a budget constraint $\sum_{x \in \mathcal{X}} \mu(x) \leq n$, where $\mathcal{N}^{\mathcal{X}} = \{\mu : \mathcal{X} \to \mathbb{N}\}$. This problem is known as *c-optimal design*. Unbiased linear estimation of $c^\top \theta^*$ is possible only when $c$ belongs to the image of $V(\mu)$, and in this case the best linear unbiased estimator of the scalar product $c^\top \theta^*$ is given by $c^\top \widehat{\theta}$, where $\widehat{\theta}$ is the least-square estimator defined as

$$\widehat{\theta} = V(\mu)^+ \sum_{x \in \mathcal{X}} a_x \left( \sum_{i \leq \mu(x)} y_{x,i} \right) \quad \text{for} \quad V(\mu) = \sum_{x \in \mathcal{X}} \mu(x) a_x a_x^\top. \tag{5}$$

The variance of the estimator $c^\top \widehat{\theta}$ is then equal to $c^\top V(\mu)^+ c$.

Exact $c$-optimal design aims at choosing the allocation $\mu \in \mathcal{N}^{\mathcal{X}}$ minimizing the variance of $c^\top \widehat{\theta}$ for a given budget $\sum_x \mu(x) \leq n$, under the constraint that $c \in \text{Range}(V(\mu))$. Let us define the normalized design $\pi : x \in \mathcal{X} \mapsto \mu(x)/n$, and let us underline that $\pi$ defines a probability on $\mathcal{X}$. The variance of $c^\top \widehat{\theta}$ is then equal to $n^{-1} c^\top V(\pi)^+ c$. In the limit $n \to +\infty$, the problem is equivalent to

the problem of approximate $c$-optimal design (sometimes simply referred to as $c$-optimal design), that aims at finding a probability measure $\pi \in \mathcal{P}_c^{\mathcal{X}} := \{\pi \in \mathcal{P}^{\mathcal{X}} : c \in \text{Range}(V(\pi))\}$ solution to the following problem

$$\min_{\pi \in \mathcal{P}_c^{\mathcal{X}}} c^{\top} V(\pi)^{+} c. \qquad \text{(c-optimal design)}$$

Note that when $\{a_x : x \in \mathcal{X}\}$ spans $\mathbb{R}^{d+1}$, for any $c \in \mathbb{R}^{d+1}$, there exists a design $\pi$ such that $c \in \text{Range}(V(\pi))$, and hence the $c$-optimal design problem admits a solution.

**Computation of the $\mathbf{e_{d+1}}$-optimal design** Finding an exact optimal allocation $\mu \in \mathcal{N}^{\mathcal{X}}$ under the constraint that $\sum_{x \in \mathcal{X}} \mu(x) \leq n$ is unfortunately NP-complete. However, finding an approximate optimal design $\pi \in \mathcal{P}_c^{\mathcal{X}}$ can be done in polynomial time [41]. Several algorithms, including multiplicative algorithms [13] and a simplex method of linear programming [17], have been proposed to iteratively approximate the optimal design. More recently, [32] suggested using screening tests to remove inessential points to accelerate optimization algorithms.

Classical results from $e_{d+1}$-optimal design show that there exists a $c$-optimal design supported by at most $d + 1$ points (see, e.g., [30, 17] for a proof of this result). The following Lemma indicates how to obtain an exact design by rounding an approximate design supported by at most $d + 1$ points.

**Lemma 3.** *For any $\pi \in \mathcal{M}_{e_{d+1}}^{\mathcal{X}}$ and any $m > 0$, the estimator $e_{d+1}^{\top} \widehat{\theta}_{\mu}$ computed from the design $\mu :$ $x \mapsto \lceil m\pi(x) \rceil$ is an unbiased estimator of $e_{d+1}^{\top} \theta$ and it has a variance at most $m^{-1} e_{d+1}^{\top} V(\pi)^{+} e_{d+1}$.*

Obviously, similar results also hold for G-optimal design.

**Lemma 4.** *Let $\pi$ be a solution of the G-optimal design problem (2). Then, for any $m > 0$ and any $x \in \mathcal{X}$, the estimator $a_x^{\top} \widehat{\theta}_{\mu}$ computed from the design $\mu : x \mapsto \lceil m\pi(x) \rceil$ is an unbiased estimator of the evaluation $a_x^{\top} \theta$, and it has a variance*

$$a_x^{\top} V(\mu)^{+} a_x \leq m^{-1}(d + 1).$$

### A.2 Interpretation of $\kappa_*$ in terms of separation of the groups

Next theorem, due to Elfving, characterizes solutions to the $c$-optimal design problem.

**Theorem 5** ([10]). *Let $\mathcal{S} = convex\ hull\ \{+a_x, -a_x : x \in \mathcal{X}\}$ be the Elfving's set of $\{a_x : x \in \mathcal{X}\} \subset \mathbb{R}^{d+1}$, and let $\partial\mathcal{S}$ denote the boundary of $\mathcal{S}$. A design $\pi \in \mathcal{P}_c^{\mathcal{X}}$ is c-optimal for $c \in \mathbb{R}^{d+1}$ if and only if there exists $\zeta \in \{-1, +1\}^{\mathcal{X}}$ and $t > 0$ such that*

$$tc = \sum_{x \in \mathcal{X}} \pi(x)\zeta_x a_x \in \partial\mathcal{S}.$$

*Moreover, $t^{-2} = c^{\top} (V(\pi))^{+} c$ is value of the c-optimal design problem.*

Elfving's characterization of the $e_{d+1}$-optimal design allows us to derive the following equivalent characterization of $\kappa_*$.

**Lemma 5.** $\kappa_* = \max\limits_{u \in \mathbb{R}^d} \dfrac{1}{\max_{x \in \mathcal{X}} \left(x^{\top} u + z_x\right)^2}.$

Lemma 1 follows from the characterization in Lemma 5. When $\kappa_* > 1$, the vector $\tilde{u}$ defined as $\tilde{u} = \text{argmax}_{u \in \mathbb{R}^d} \frac{1}{\max_{x \in \mathcal{X}} (x^{\top} u + z_x)^2}$ is a normal vector of the separating hyperplane $\mathcal{H}$ in Figure 1. Moreover, as shown in the proof of Lemma 1, the margin is in this case equal to $1 - \kappa_*^{-1/2}$, while the maximum distance of all points to the hyperplane is $1 + \kappa_*^{-1/2}$.

**Application to the action set $\mathcal{A}$ of Lemma 10** To provide the reader with intuition on $\kappa_*$, we analyze here the set of actions used to derive the lower bound in Theorem 3. Let $\mathcal{A} = \left\{ \begin{pmatrix} x_1 \\ z_{x_1} \end{pmatrix}, ..., \begin{pmatrix} x_{d+1} \\ z_{x_{d+1}} \end{pmatrix} \right\}$, where $\begin{pmatrix} x_i \\ z_{x_i} \end{pmatrix} = e_i + e_{d+1}$, for $i \in \{2, ..., \lfloor d/2 \rfloor\}$, $\begin{pmatrix} x_i \\ z_{x_i} \end{pmatrix} = e_i - e_{d+1}$ for $i \in \{\lfloor d/2 \rfloor + 1, ..., d\}$, and $\begin{pmatrix} x_{d+1} \\ z_{x_{d+1}} \end{pmatrix} = -\left(1 - \frac{2}{\sqrt{\kappa_*} + 1}\right) e_1 - e_{d+1}$. We show in Lemma 10 that the minimal variance for estimating the bias on $\mathcal{A}$ is indeed $\kappa_*$.

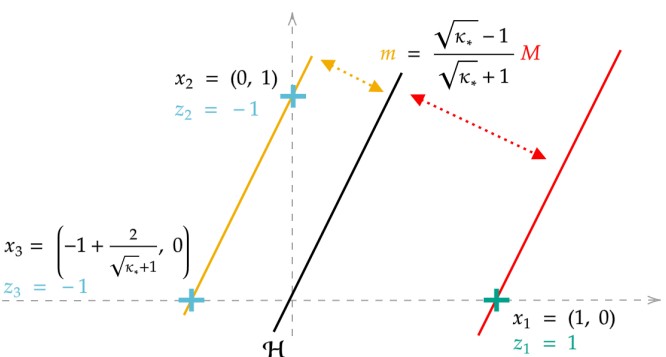

Figure 2: Illustration of Lemma 1 on the action set $\mathcal{A}$ described above for $d = 2$.

The set of actions $\mathcal{A}$ spans $\mathbb{R}^{d+1}$, however it is easy to see that only $x_1$ and $x_{d+1}$ can be used to estimate the bias. On the one hand, when $\kappa = 1$, $\begin{pmatrix} x_{d+1} \\ z_{x_{d+1}} \end{pmatrix} = \begin{pmatrix} 0 \\ -1 \end{pmatrix}$, so the bias can be evaluated just by sampling $x_{d+1}$. In the other hand, in the limit where $\kappa_* \to \infty$, the problems becomes more difficult as $\begin{pmatrix} x_{d+1} \\ z_{x_{d+1}} \end{pmatrix}$ tends to $- \begin{pmatrix} x_1 \\ z_{x_1} \end{pmatrix}$. In the limit $\kappa_* = \infty$, it is impossible to distinguish between the contribution of $\gamma^\top e_1$ and $\omega$ in the evaluations of actions 1 and $d+1$: the problem becomes not identifiable. We represent this setting for an intermediate value of $\kappa_*$ in Figure 2. We also represent the separating hyperplane, margin $m$ and distance $M$ of Lemma 1.

### A.3    Comparison to the conditioning number

By contrast to classical complexity measures such as conditioning numbers that give equal weight to all observations, optimal design gives flexibility to choose $d + 1$ best actions to estimate the bias, and therefore allows for sharper bounds.

Indeed, by definition of $\kappa_*$,

$$\kappa_* \leq e_{d+1} V(\pi^u)^+ e_{d+1},$$

where $\pi^u$ is the uniform measure giving the same weight $1/k$ to all actions. Now, $V(\pi^u)$ is the classical covariance matrix associated with the design points $a_x \in \mathcal{A}$, so the condition number $CN$ of this design is given by

$$CN = \frac{\lambda_{\max}(V(\pi^u))}{\lambda_{\min}(V(\pi^u))}.$$

We see that $e_{d+1} V(\pi^u)^+ e_{d+1} \leq \lambda_{\min}(V(\pi^u))^{-1}$. When the actions $a_x$ are bounded (for example $\|a_x\| \leq M$), this implies that $\kappa_* \leq CN/M$.

We provide an example showing that $\kappa_*$ can be much smaller than the conditioning number. Consider the following example in dimension $d = 2$ with $k \geq 4$ actions, where $x_1 = (1, 0)$ and $x_2 = (-1, 0)$ belong to group 1, and $x_3, ..., x_k$ are identical, equal to $(0, 1)$, and in group $-1$. Then, Lemma 1 shows that the minimal variance for estimating the bias is indeed 1, and that the optimal design puts equal mass on $x_1$ and $x_2$. On the other hand, straightforward computations show that the conditioning number of the covariance matrix is $\frac{1+(k-2)^{-1}+\sqrt{1+(k-2)^{-2}}}{1+(k-2)^{-1}-\sqrt{1+(k-2)^{-2}}}$. Thus, on this example, $CN/\kappa_*$ is of order $k$.

### A.4    Comparison to the worst-case alignment constant

Lemma 5 also allows us to compare the bound in Theorem 1 with previous results on linear bandit with partial monitoring, expressed in terms of the worst-case alignment constant.

Previous work on linear bandit with partial linear monitoring measures the difficulty of the bandit game using the *worst-case alignment constant* $\alpha$, defined as

$$\alpha = \max_{u \in \mathbb{R}^d} \frac{\max_{x,x' \in \mathcal{X}}((x - x')^\top u)^2}{\max_{x \in \mathcal{X}}(z_x x^\top u + 1)^2}.$$

The following Lemma shows that this constant is essentially equivalent to the minimal variance of the bias estimator $\kappa_*$.

**Lemma 6.** $\frac{\kappa_*}{3} \le \alpha \le 16\kappa_*$.

On the one hand, Lemma 6 shows that $\kappa_*$ and $\alpha$ are essentially equivalent. In particular, Theorem 3 implies that the large $T$ regret is of order $\alpha^{1/3} \log(T)^{1/3} T^{2/3}$. This improves over previous known rates, obtained in [20], by a factor $d^{1/2} \log(T)^{1/6} (\log(kT)/\log(T))^{1/2}$.

On the other hand, as underlined, the constant $\kappa_*$ appears when considering the well-studied problem of $c$-optimal design. Therefore, classical results and algorithms for optimal design can be used to characterize and compute this constant.

### A.5 Optimal bias estimation against the worst parameter

The constant $\kappa_*$ also appears naturally when considering the related problem of optimal bias estimation against the worst parameter.

**Regret of $e_{d+1}$-optimal design**  Recall that $\kappa_*$ denotes the *minimal variance of the bias estimator*, i.e. the value of the solution of the $e_{d+1}$-optimal design problem

$$\kappa_* = \min_{\pi \in \mathcal{P}_{e_{d+1}}^{\mathcal{X}}} e_{d+1}^\top \left(V(\pi)\right)^+ e_{d+1},$$

The $e_{d+1}$-optimal design can be equivalently defined as the solution of the problem

$$\text{minimize} \sum_{x \in \mathcal{X}} \mu(x) \quad \text{such that } \mu \in \mathcal{M}_{e_{d+1}}^{\mathcal{X}} \text{ and } e_{d+1}^\top V(\mu)^+ e_{d+1} \le \kappa_*. \tag{6}$$

The characterization given in Equation (6) underlines that the $e_{d+1}$-optimal design provides (up to discretization issues) the minimal number of samples required for estimating $\omega^*$ with a variance $\kappa_*$. Let us denote by $\mu^*$ the optimal design for estimating $\omega^*$ with a variance 1, defined as

$$\mu^* = \operatorname*{argmin}_{\mu} \sum_{x \in \mathcal{X}} \mu(x) \quad \text{such that } \mu \in \mathcal{M}_{e_{d+1}}^{\mathcal{X}} \text{ and } e_{d+1}^\top V(\mu)^+ e_{d+1} \le 1.$$

Note that from the definition of $\kappa_*$, we have $\sum_x \mu^*(x) = \kappa_*$.

A first (naive) approach to obtain an estimate of the bias parameter $\omega^*$ with precision level $\epsilon > 0$ would consist in sampling actions according to $\epsilon^{-2}\mu^*$, rounded according to the procedure defined in Lemma 3. Let us denote by $\Delta_x$ the gap $\Delta_x = \max_{x' \in \mathcal{X}}(x' - x)^\top \gamma^*$ between the (non-observed) reward of the best action and the reward of the action $x$. The regret corresponding to this estimation phase would then be

$$\epsilon^{-2} \sum_{x \in \mathcal{X}} \mu^*(x) \Delta_x,$$

which can be as large as $\kappa_* \epsilon^{-2} \max_x \Delta_x$. Interestingly, we show that the regret corresponding to the $e_{d+1}$-optimal design is equivalent (up to a small multiplicative constant) to the minimax regret.

**Optimal worst-case estimation**  The minimax regret corresponds to the regret of the best sampling scheme against the worst admissible parameter $\gamma$. Note that, for a given design $\mu$, this worst-case regret is given by

$$\max_{x' \in \mathcal{X}, \gamma \in \mathcal{C}(\mathcal{X})} \sum_x \mu(x)(x' - x)^\top \gamma,$$

where we recall that $\mathcal{C}(\mathcal{X}) = \left\{\gamma \in \mathbb{R}^d : \forall x \in \mathcal{X}, |x^\top \gamma| \le 1\right\}$ is the set of admissible parameters. To achieve the lowest regret against the worst parameter, we must use the minimax optimal design $\widetilde{\mu}$ solution to the problem

$$\widetilde{\mu} = \operatorname*{argmin}_{\mu} \max_{x' \in \mathcal{X}, \gamma \in \mathcal{C}(\mathcal{X})} \sum_{x \in \mathcal{X}} \mu(x)(x' - x)^\top \gamma \quad \text{such that } \mu \in \mathcal{M}_{e_{d+1}}^{\mathcal{X}} \text{ and } e_{d+1}^\top V(\mu)^+ e_{d+1} \le 1.$$

Lemma 2 underlines that the regret corresponding to the $e_{d+1}$-optimal design is no larger than twice the minimax regret.

## A.6 Additionnal results the $\Delta$-optimal design

Recall that for a vector of gaps $\Delta = (\Delta_x)_{x \in \mathcal{X}}$, $\mu^\Delta$ denotes the $\Delta$-optimal design, defined as the solution of the following problem

$$\mu^\Delta = \underset{\mu}{\operatorname{argmin}} \sum_{x \in \mathcal{X}} \mu(x)\Delta_x \quad \text{such that } \mu \in \mathcal{M}^\mathcal{X}_{e_{d+1}} \text{ and } e^\top_{d+1}V(\mu)^+ e_{d+1} \leq 1. \quad \text{($\Delta$-optimal design)}$$

If we knew the gaps $\Delta_x$, we could sample the actions according to the $\Delta$-optimal design $\mu^\Delta$, and pay the regret $\epsilon^{-2}\kappa(\Delta)$ (up to rounding error) for estimating $\omega^*$ with an error smaller than $\epsilon$, where

$$\kappa(\Delta) = \sum_{x \in \mathcal{X}} \mu^\Delta(x)\Delta_x.$$

**Lemma 7.** *If $\gamma^* \in \mathcal{C}(\mathcal{X})$, then $\kappa(\Delta) \leq 2\kappa_*$*

*Proof.* Be definition of $\mathcal{C}(\mathcal{X})$, for all $\gamma^* \in \mathcal{C}(X)$, all $x, x' \in \mathcal{X}$, we have

$$(x - x')^\top \gamma^* \leq |x^\top \gamma^*| + |x'^\top \gamma^*| \leq 2.$$

Then,

$$\kappa(\Delta) \leq 2 \min_\mu \sum_{x \in \mathcal{X}} \mu(x) \quad \text{such that } \mu \in \mathcal{M}^\mathcal{X}_{e_{d+1}} \text{ and } e^\top_{d+1}V(\mu)^+ e_{d+1} \leq 1.$$

Let $\mu_*$ be the solution of the $e_{d+1}$-optimal design problem

$$\underset{\mu}{\operatorname{minimize}} \, e^\top_{d+1}V(\mu)^+ e_{d+1} \text{ such that } \mu \in \mathcal{P}^\mathcal{X}_{e_{d+1}}.$$

By definition of $\kappa_*$, we see that $e^\top_{d+1}V(\mu_*)^+ e_{d+1} = \kappa_*$. This implies that the measure $\kappa_* \times \mu_*$ verifies the constraints $e^\top_{d+1}V(\kappa_* \times \mu_*)^+ e_{d+1} \leq 1$ and $\kappa_* \mu_* \in \mathcal{M}^\mathcal{X}_{e_{d+1}}$. Thus,

$$\kappa(\Delta) \leq 2 \sum_{x \in \mathcal{X}} \kappa_* \mu_*(x) = 2\kappa_*.$$

$\square$

**On the regret $\kappa(\Delta)$**   The function $\kappa$ verifies the following properties.

**Lemma 8.** *For two vectors of gaps $\Delta$, $\Delta'$, denote by $\Delta \wedge \Delta'$ (respectively $\Delta \vee \Delta'$) the vector of gaps given by $(\Delta \wedge \Delta')_x = \Delta_x \wedge \Delta'_x$ (respectively $(\Delta \vee \Delta')_x = \Delta_x \vee \Delta'_x$) for all $x \in \mathcal{X}$. Moreover, denote $\Delta \leq \Delta'$ if $\Delta_x \leq \Delta'_x$ for all $x \in \mathcal{X}$. Then, the following properties hold :*

*i) for all $c > 0$, $\kappa(c\Delta) = c\kappa(\Delta)$;*

*ii) if $\Delta \leq \Delta'$, then $\kappa(\Delta) \leq \kappa(\Delta')$;*

*iii) $\kappa(\Delta \vee \Delta') \geq \kappa(\Delta) \vee \kappa(\Delta')$;*

*iv) the function $\epsilon \mapsto \kappa(\Delta \vee \epsilon)$ is continuous at $0$.*

## A.7 Computation of G- and $\Delta$-optimal design

Computing the optimal design is a convex problem, for which many algorithms have been proposed. The first method to compute G-optimal design is due to [12] and [44]; later, [39] proposed a multiplicative weight update algorithm. More recently, [40] suggested to use a Semi-Definite Programming approach to solve the G-optimal design problem. Linear programming was used in [17] to compute $c$-optimal design, while [34] studied a SDP formulation of this problem. Reducing

the G-optimal problem to a Mixed-Integer, Second Order Cone Programming, [37] proposed a new algorithm based on interior point methods. We refer the interested reader to the review in [36].

In practice, one can rely on the R package OptimalDesign or the Python Package PICOS [38] to compute G- and $c$-optimal design.

The following Lemma allows us to reduce the problem of finding a $\Delta$-optimal design to that of a $c$-optimal design for some rescaled features.

**Lemma 9.** *For any vector $\Delta \in (0, +\infty)^{\mathcal{X}}$, let $\pi^\Delta$ be the $e_{d+1}$-optimal design relative to the set $\mathcal{A}^\Delta = \left\{ \Delta_x^{-1/2} \begin{pmatrix} x \\ z_x \end{pmatrix} : x \in \mathcal{X} \right\}$ and let $\kappa^\Delta = e_{d+1}^\top V(\pi^\Delta)^+ e_{d+1}$ be the $e_{d+1}$-optimal variance relative to $\mathcal{A}^\Delta$. Then, the $\Delta$-optimal design $\mu^\Delta$ is given by $\mu^\Delta(x) = \kappa^\Delta \pi^\Delta(x)\Delta_x^{-1}$ for all $x \in \mathcal{X}$. In addition, the support of $\mu^\Delta$ can be chosen to be of cardinnality at most $d + 1$.*

Thus, Lemma 9 shows that to compute the $\Delta$-optimal design, one should follow these steps :

1. Compute the rescaled features $\mathcal{A}^\Delta$;

2. Compute the $e_{d+1}$-optimal design $\pi^\Delta$ on $\mathcal{A}^\Delta$, as well as the variance term $\kappa^\Delta = e_{d+1}^\top \left( \sum_{x \in \mathcal{X}} \frac{\pi^\Delta(x)}{\Delta_x} a_x a_x^\top \right)^+ e_{d+1}$;

3. Compute the $\Delta$-optimal design $\mu^\Delta$ given by $\mu^\Delta(x) = \kappa^\Delta \pi^\Delta(x)\Delta_x^{-1}$ for all $x \in \mathcal{X}$.

# B   Detailed Fair Phased Elimination algorithm

We present the notations used in Algorithm 4. The phases are indexed by $l \in \mathbb{N}^*$. The sets $\mathcal{X}_l^{(z)}$ for $z \in \{-1, +1\}$ corresponds to actions in group $z$ that are considered as potentially optimal in phase $l$. The variable $\widehat{z}_l^*$ encodes the group determined as optimal: it is 0 as long as this group has not been determined. The subscript $(z)$ refer to the group $z$ when $z \in \{-1, +1\}$, and otherwise to the estimation of the bias $\omega^*$: for example, the probability $\pi_l^{(z)}$ for $z \in \{-1, +1\}$ and $l > 1$ corresponds to the approximate G-optimal design on $\mathcal{X}_l^{(z)}$. Then, for $z \in \{-1, +1\}$, allocations $\mu^{(z)}$ (resp. $\mu^{(0)}$) correspond to allocation of samples in the exploration phase $\text{Exp}_l^{(z)}$ (resp. $\text{Exp}_l^{(0)}$). Similarly, $V_l^{(z)}$ (resp $V_l^{(0)}$) denotes the variance matrix of the estimator $\begin{pmatrix} \widehat{\gamma}_l^{(z)} \\ \widehat{\omega}_l^{(z)} \end{pmatrix}$ (resp. $\widehat{\omega}_l^{(0)}$) obtained from observations made during phase $\text{Exp}_l^{(z)}$ (resp. $\text{Exp}_l^{(0)}$). Finally, $\text{Explore}_l^{(z)}$ (resp. $\text{Explore}_l^{(0)}$) is a Boolean variable indicating whether the exploration at phase $l$ for group $z$ (resp. for the bias parameter) has been performed. It is used in the proofs to ensure that the corresponding estimators are well defined.

---

**Algorithm 4** Fair Phased Elimination (detailed version)

---

1: **Input:** $\delta$, $T$, $k = |\mathcal{X}|$

2: **Initialize:** Recovery $\leftarrow \emptyset$, $t \leftarrow 0$, $l \leftarrow 1$ $\widehat{z^*_1} \leftarrow 0$,

3:          $\mathcal{X}_1^{(+1)} \leftarrow \{x : z_x = 1\}$, $\mathcal{X}_1^{(-1)} \leftarrow \{x : z_x = -1\}$, $\widehat{\Delta}_x^1 \leftarrow 2$ for $x \in \mathcal{X}$

4: **while** $t < T$ **do**

5:      **Initialize:** $\epsilon_l \leftarrow 2^{2-l}$, $\widehat{z^*}_{l+1} \leftarrow \widehat{z^*}_l$, $\widehat{\Delta}^{l+1} \leftarrow \widehat{\Delta}^l$, $\text{Explore}_l^{(z)} \leftarrow$ False for $z \in \{-1, 0, +1\}$

6:      **for** $z \in \{-1, +1\}$ such that $z \neq -\widehat{z^*}_l$ **do**          ▷ G-optimal Exploration and Elimination

7:          $\pi_l^{(z)} \leftarrow \underset{\pi}{\arg\min} \left\{ \underset{x \in \mathcal{X}_l^{(z)}}{\max} a_x^\top V(\pi)^+ a_x : \pi \in \mathcal{P}_{\mathcal{X}_l^{(z)}}^{\mathcal{X}_l^{(z)}}, |\text{supp}(\pi)| \leq \frac{(d+1)(d+2)}{2} \right\}$

8:          $\mu_l^{(z)}(x) \leftarrow \left\lceil \frac{2(d+1)\pi_l^{(z)}(x)}{\epsilon_l^2} \log\left(\frac{kl(l+1)}{\delta}\right) \right\rceil$ for all $x \in \mathcal{X}_l^{(z)}$

9:          $n_l^{(z)} \leftarrow \underset{x \in \mathcal{X}_l^{(z)}}{\sum} \mu_l^{(z)}(x)$, $\text{Exp}_l^{(z)} \leftarrow \left\{ t+1, ..., T \wedge (t + n_l^{(z)}) \right\}$

10:          **if** $t + n_l^{(z)} \leq T$ **then**

11:             $\text{Explore}_l^{(z)} \leftarrow$ True, choose each action $x \in \mathcal{X}_l^{(z)}$ exactly $\mu_l^{(z)}(x)$ times

12:             $V_l^{(z)} \leftarrow \sum_{t \in \text{Exp}_l^{(z)}} a_{x_t} a_{x_t}^\top$, $\widehat{\theta}_l^{(z)} \leftarrow \left( V_l^{(z)} \right)^+ \sum_{t \in \text{Exp}_l^{(z)}} y_t a_{x_t}$

13:             $\mathcal{X}_{l+1}^{(z)} \leftarrow \left\{ x \in \mathcal{X}_l^{(z)} : \max_{x' \in \mathcal{X}_l^{(z)}} (a_{x'} - a_x)^\top \widehat{\theta}_l^{(z)} \leq 3\epsilon_l \right\}$

14:          **else** for $t \in \text{Exp}_l^{(z)}$, sample empirical best action in $\mathcal{X}_l^{(z)}$

15:          $t \leftarrow t + n_l^{(z)}$

16:      **if** $\widehat{z^*}_l = 0$ **then**

17:          compute the $\widehat{\Delta}^l$-optimal design $\widehat{\mu}_l$ and the corresponding regret $\kappa(\widehat{\Delta}^l)$

18:          **if** $\epsilon_l \leq \left( \kappa(\widehat{\Delta}^l) \log(T)/T \right)^{1/3}$ **then**          ▷ Recovery phase

19:             Recovery $\leftarrow \{t, ..., T\}$

20:             sample empirical best action in $\mathcal{X}_{l+1}^{(-1)} \cup \mathcal{X}_{l+1}^{(1)}$ until the end of the budget, $t \leftarrow T$

21:          **else**          ▷ $\widehat{\Delta}^l$-optimal Exploration and Elimination

22:             $\mu_l^{(0)}(x) \leftarrow \left\lceil \frac{2\widehat{\mu}_l(x)}{\epsilon_l^2} \log\left(\frac{l(l+1)}{\delta}\right) \right\rceil$ for all $x \in \mathcal{X}$

23:             $n_l^{(0)} \leftarrow \underset{x \in \mathcal{X}}{\sum} \mu_l^{(0)}(x)$, $\text{Exp}_l^{(0)} \leftarrow \left\{ t, ..., T \wedge (t + n_l^{(0)}) \right\}$

24:             **if** $t + n_l^{(0)} \leq T$ **then**

25:                 $\text{Explore}_l^{(0)} \leftarrow$ True, choose each action $x \in \mathcal{X}$ exactly $\mu_l^{(0)}(x)$ times

26:                 $V_l^{(0)} \leftarrow \sum_{t \in \text{Exp}_l^{(0)}} a_{x_t} a_{x_t}^\top$, $\widehat{\omega}_l^{(0)} \leftarrow e_{d+1}^\top \left( V_l^{(0)} \right)^+ \sum_{t \in \text{Exp}_l^{(0)}} y_t a_{x_t}$

27:                 **for** $x \in \mathcal{X}_{l+1}^{(-1)} \cup \mathcal{X}_{l+1}^{(1)}$ **do**

28:                     $\widehat{m}_{l,x} \leftarrow a_x^\top \widehat{\theta}_l^{(z_x)} - z_x \widehat{\omega}_l^{(0)}$

29:                     $\widehat{\Delta}_x^{l+1} \leftarrow \left( \max_{x' \in \mathcal{X}_{l+1}^{(-1)} \cup \mathcal{X}_{l+1}^{(1)}} \widehat{m}_{l,x'} - \widehat{m}_{l,x} + 4\epsilon_l \right) \wedge 2$

30:                 **for** $z \in \{-1, +1\}$ **do**

31:                     **if** $\underset{x \in \mathcal{X}_{l+1}^{(z)}}{\max} \widehat{m}_{l,x} - 2\epsilon_l \geq \underset{x \in \mathcal{X}_{l+1}^{(-z)}}{\max} \widehat{m}_{l,x} + 2\epsilon_l$ **then** $\widehat{z^*}_{l+1} \leftarrow z$

32:             **else** sample empirical best action in $\mathcal{X}_{l+1}^{(-1)} \cup \mathcal{X}_{l+1}^{(1)}$ until the end of the budget, $t \leftarrow T$

33:             $t \leftarrow t + n_l^{(0)}$

34:      $l \leftarrow l + 1$

---

# C Proofs

Before proving the main results our this paper, we begin by outlining in Section C.1 the main ideas used to obtain upper and lower bounds on the regret. Then, Theorem 1 is proved in Section C.2, Theorem 2 is proved in Section C.3, Theorem 3 is proved in Section C.4, and Theorem 4 is proved in Section C.5. Extension of Theorem 4 to $d = 2$ and $d = 3$ is discussed in Section C.6. Finally, auxiliary lemmas are proved in Appendix C.7.

For an event $\mathcal{F}$ such that $\mathbb{P}(\mathcal{F}) > 0$, we denote by $\mathbb{E}_{|\mathcal{F}}$ (resp. $\mathbb{P}_{|\mathcal{F}}$) the expectation (resp. the probability) conditionally on $\mathcal{F}$.

## C.1 Outline of the proofs

### C.1.1 Outline of the proof of Theorem 1

The proof of Theorem 1 can be found in Appendix C.2. We outline here the keys ingredients to this proofs. We begin by introducing some notations.

**Notations** We denote by $L_T$ the largest integer $l$ such that $\epsilon_l \geq \kappa_*^{1/3} T^{-1/3} \log(T)^{1/3}$. We denote by $L^{(0)}$ the last phase where $\widehat{\Delta}^l$-optimal Exploration and Elimination happens. We denote by $\text{Exp}_l^{(z)}$ the time indices where G-exploration is performed on $\mathcal{X}_l^{(z)}$ and by $\text{Exp}_l^{(0)}$ the time indices where $\Delta$-exploration is performed at phase $l$. We also denote by Recovery the time indices subsequent to the stopping criterion, this set being empty when the stopping criterion is not activated.

We define a "good" event $\overline{\mathcal{F}}$ such that the errors $\left| a_x^\top \left( \theta^* - \widehat{\theta}_l \right) \right|$ and $|\omega^* - \widehat{\omega}_l^{(0)}|$ are smaller than $\epsilon_l$ for all $l$ such that these quantities are defined, and all $x \in \mathcal{X}_l^{(-1)}$ and $\mathcal{X}_l^{(+1)}$. In the following, we use $c, c'$ to denote positive absolute constants, which may vary from line to line. With these notations, we decompose the regret as follows :

$$
\begin{aligned}
R_T \leq\ & 2T\mathbb{P}(\mathcal{F}) + \mathbb{E}_{|\overline{\mathcal{F}}}\left[ \underbrace{\sum_{l \leq L_T} \sum_{z \in \{-1,+1\}} \sum_{t \in \text{Exp}_l^{(z)}} (x^* - x_t)^\top \gamma^*}_{R_T^G} \right] + \mathbb{E}_{|\overline{\mathcal{F}}}\left[ \underbrace{\sum_{l \leq L^{(0)}} \sum_{t \in \text{Exp}_l^{(0)}} (x^* - x_t)^\top \gamma^*}_{R_T^\Delta} \right] \\
& + \mathbb{E}_{|\overline{\mathcal{F}}}\left[ \underbrace{\sum_{l \geq L_T+1} \sum_{z \in \{-1,+1\}} \sum_{t \in \text{Exp}_l^{(z)}} (x^* - x_t)^\top \gamma^* + \sum_{t \in \text{Recovery}} (x^* - x_t)^\top \gamma^*}_{R_T^{Rec}} \right].
\end{aligned}
$$

**Bound on $T\mathbb{P}(\mathcal{F})$.** Using arguments based on concentration of Gaussian variables, we show that $\mathbb{P}(\mathcal{F}) \leq 2T^{-1}$.

**Bound on $R_T^G$.** We show that on $\overline{\mathcal{F}}$, only actions with gaps smaller than $c\epsilon_l$ remain in the sets $\mathcal{X}_l^{(-1)}$ and $\mathcal{X}_l^{(+1)}$. The length of each G-optimal Exploration and Elimination phase is of the order $d\log(klT)/\epsilon_l^2$, so the regret of each phase is of the order $d\log(klT)/\epsilon_l$. Summing over the different phases, we find that

$$R_T^G \leq cd \log(kL_TT)/\epsilon_{L_T}. \tag{7}$$

Using the definition of $L_T$, we find that $R_T^G \leq cd \log(kL_TT)\kappa_*^{-1/3} \log(T)^{-1/3}T^{1/3}$.

**Bound on $R_T^\Delta$.** We show that on $\mathcal{F}$, $\widehat{\Delta}^l \geq \Delta$ for all $l \geq 1$. Then, our choice of design $\mu_l^{(0)}$ ensures that for $l \leq L^{(0)}$, on $\overline{\mathcal{F}}$,

$$\sum_{t \in \text{Exp}_l^{(0)}} (x^* - x_t)^\top \gamma^* \leq c \left( \frac{\log(l(l+1)T)}{\epsilon_l^2} \kappa(\widehat{\Delta}^l) + d + 1 \right)$$

for some constant $c > 0$. Summing over the different phases, we find that

$$R_T^\Delta \leq c\kappa(\widehat{\Delta}^{L^{(0)}}) \log(L^{(0)}T)/\epsilon_{L^{(0)}}^2. \tag{8}$$

Now, the algorithm does not enter the Recovery phase before phase $L^{(0)} + 1$, so we must have $\epsilon_{L^{(0)}}^{-2} \leq T^{2/3} \log(T)^{-2/3} \kappa(\widehat{\Delta}^{L^{(0)}})^{-2/3}$. This implies that $R_T^\Delta \leq c\kappa(\widehat{\Delta}^{L^{(0)}})^{1/3} \log(T)^{1/3} T^{2/3}$. Since $\kappa(\widehat{\Delta}^l) \leq 2\kappa_*$, we find that $R_T^\Delta \leq c'\kappa_*^{1/3} \log(T)^{1/3} T^{2/3}$.

**Bound on $\mathbf{R_T^{Rec}}$.** On the one hand, the actions selected during the Phases $\mathrm{Exp}_l^{(-1)}$ and $\mathrm{Exp}_l^{(+1)}$ for $l \geq L_T + 1$ are sub-optimal by a gap at most $c\epsilon_{L_T}$ on the event $\overline{\mathcal{F}}$. On the other hand, if the algorithm enters the Recovery phase at a phase $l$, then

$$\epsilon_l \leq \kappa(\widehat{\Delta}^{L^{(0)}})^{1/3} T^{-1/3} \log(T)^{1/3} \leq \kappa_*^{1/3} T^{-1/3} \log(T)^{1/3},$$

so we must have $l = L^{(0)} + 1 \geq L_T + 1$. Therefore, all actions selected during the Recovery phase are sub-optimal by a gap at most $c\epsilon_{L_T}$. Then, $R_T^{Rec}$ can be bounded as $R_T^{Rec} \leq c\epsilon_{L_T}T$. This implies in particular that $R_T^{Rec} \leq c'\kappa_*^{1/3} \log(T)^{1/3} T^{2/3}$.

When $T \geq T_{\kappa_*,d,k}$ for some $T_{\kappa,d,k}$ large enough, we find that $\mathbb{R}_T \leq c'\kappa_*^{1/3} \log(T)^{1/3} T^{2/3}$.

### C.1.2 Outline of the Proof of Theorem 2

The proof of Theorem 2 is close to that of Theorem 1, and we adopt the same notations as in the proof sketch above.

**Notations** We denote by $L^{(0)}$ the last phase where $\widehat{\Delta}^l$-optimal Exploration and Elimination happens. We denote $\overline{\mathcal{F}}$ some "good" event such that the errors $|a_x^\top(\theta^* - \widehat{\theta}_l^{(z_x)})|$ and $|\omega^* - \widehat{\omega}_l^{(0)}|$ are smaller than $\epsilon_l$ for all $l$ such that these quantities are defined, and all $x \in \mathcal{X}_l^{(-1)} \cup \mathcal{X}_l^{(+1)}$. We denote by $\mathrm{Exp}_l^{(z)}$ the time indices where G-exploration is performed on $\mathcal{X}_l^{(z)}$ and by $\mathrm{Exp}_l^{(0)}$ the time indices where $\Delta$-exploration is performed at phase $l$. We also denote by Recovery the time indices subsequent to the stopping criterion, this set being empty when the stopping criterion is not activated. In the following, we use $c, c'$ to denote positive absolute constants, which may vary from line to line.

**Fact 1** Let $l_{\Delta_{\min}}$ be the largest integer such that $\epsilon_{l_{\Delta_{\min}}} \geq C\Delta_{\min}$ for some well-chosen absolute constant $C > 0$. We show that on the good event $\overline{\mathcal{F}}$, no more than $l_{\Delta_{\min}}$ G-optimal Exploration and Elimination phases are needed to find the best action. For all phases $l \geq l_{\Delta_{\min}}$, the algorithm always chooses $x^*$, and suffers no regret.

**Fact 2** We show that on the good event $\overline{\mathcal{F}}$, for each phase $l$, $\widehat{\Delta}^l \leq c(\Delta \vee \epsilon_l)$ for some constant $c$. Lemma 8 then implies that for all $l \leq L^{(0)}$ and all $\tau > 0$, $\kappa(\widehat{\Delta}^l) \leq c\kappa(\Delta \vee \epsilon_l) \leq c(1 + \epsilon_l\tau^{-1})\kappa(\Delta \vee \tau)$.

**Fact 3** Let $l_{\Delta_{\neq}}$ be the largest integer such that $\epsilon_{l_{\Delta_{\neq}}} \geq C\Delta_{\neq}$ for some well-chosen absolute constant $C > 0$. On the good event $\overline{\mathcal{F}}$, if the algorithm enters the $\widehat{\Delta}^l$-optimal Exploration and Elimination phase at round $l \geq l_{\Delta_{\neq}}$, we show that the algorithm finds the best group at this phase. This implies that $L^{(0)} \leq l_{\Delta_{\neq}}$.

**Fact 4** We denote by $L_T$ the largest integer $l$ such that $\epsilon_l \geq (\kappa_* \log(T)/T)^{1/3}$. Since $\kappa_* \geq \kappa(\widehat{\Delta}^l)$ for all $l \geq 1$, we see that if the algorithm enters the Recovery phase, we must have $L_T \leq L^{(0)}$, and $\epsilon_{L^{(0)}} \leq \epsilon_{L_T} \approx \varepsilon_T$.

Using **Fact 1**, we find that the regret can be written as

$$
R_T \quad \leq \quad 2T\mathbb{P}\left(\mathcal{F}\right) + \mathbb{E}_{|\overline{\mathcal{F}}}\Bigg[\underbrace{\sum_{l \leq l_{\Delta_{\min}}}\sum_{z \in \{-1,+1\}}\sum_{t \in \mathrm{Exp}_l^{(z)}}(x^* - x_t)^\top \gamma^*}_{R_T^G}\Bigg]
$$

$$
+ \mathbb{E}_{|\overline{\mathcal{F}}}\Bigg[\underbrace{\sum_{l \leq L^{(0)}}\sum_{t \in \mathrm{Exp}_l^{(0)}}(x^* - x_t)^\top \gamma^*}_{R_T^\Delta}\Bigg] + \mathbb{E}_{|\overline{\mathcal{F}}}\Bigg[\underbrace{\sum_{t \in \mathrm{Recovery}}(x^* - x_t)^\top \gamma^*}_{R_T^{Rec}}\Bigg].
$$

**Bound on $\mathbf{R_T^G}$.** We rely on arguments similar to those used in Equation (7) to show that $R_T^G \leq c(d+1)\log(kl_{\Delta_{\min}}T)\epsilon_{l_{\Delta_{\min}}}^{-1}$. Since $\epsilon_{l_{\Delta_{\min}}} \geq C\Delta_{\min}$, this implies that

$$
R_T^G \leq \frac{c(d+1)\log(kl_{\Delta_{\min}}T)}{\Delta_{\min}} \leq \frac{c'd\log(T)}{\Delta_{\min}}
$$

if $T \geq k$.

**Bound on $\mathbf{R_T^\Delta} + \mathbf{R_T^{Rec}}$.** We begin by bounding $R_T^\Delta$. Recall that Equation (8) states that $R_T^\Delta \leq c\kappa(\widehat{\Delta}^{L^{(0)}})\log(l_{L^{(0)}}T)\epsilon_{L^{(0)}}^{-2}$. Using **Fact 2**, we find that for any $\tau > 0$,

$$
R_T^\Delta \leq c\kappa(\Delta \vee \tau)\log(l_{L^{(0)}}T)\left(\epsilon_{L^{(0)}}^{-2} + \epsilon_{L^{(0)}}^{-1}\tau^{-1}\right). \tag{9}
$$

Let us now consider two cases, corresponding to Recovery$= \emptyset$ and Recovery$\neq \emptyset$.

Case 1: Recovery$= \emptyset$. On the one hand, our case assumption implies that

$$
R_T^{Rec} = 0.
$$

On the other hand, by **Fact 3**, we know that on $\overline{\mathcal{F}}$, $L^{(0)} \leq l_{\Delta_{\neq}}$. Then, using the definition of $l_{\Delta_{\neq}}$ and Equation (9) with $\tau = \Delta_{\neq}$, we find that

$$
R_T^\Delta \leq c\kappa(\Delta \vee \Delta_{\neq})\log(L^{(0)}T)\Delta_{\neq}^{-2}.
$$

Case 2: Recovery$\neq \emptyset$. All actions selected during the Recovery phase belong to $\mathcal{X}_{L^{(0)}+1}^{(-1)} \cup \mathcal{X}_{L^{(0)}+1}^{(+1)}$, so on $\overline{\mathcal{F}}$ these actions are sub-optimal by a gap at most $c\epsilon_{L^{(0)}+1}$, so $R_T^{Rec} \leq cT\epsilon_{L^{(0)}+1}$. Now, since the algorithm enters the Recovery phase, we must have $\epsilon_{L^{(0)}+1} \leq \left(\kappa(\Delta^{L^{(0)}+1})\log(T)/T\right)^{1/3}$, which implies that

$$
R_T^{Rec} \leq \frac{c\kappa(\widehat{\Delta}^{L^{(0)}+1})\log(T)}{\epsilon_{L^{(0)}+1}^2}.
$$

Using **Fact 2** with $\tau = \epsilon_{L^{(0)}}$ together with Equation (9), we find that

$$
R_T^\Delta + R_T^{Rec} \leq \frac{c\kappa(\Delta \vee \epsilon_{L^{(0)}})\log(T)}{\epsilon_{L^{(0)}}^2}.
$$

On the one hand, **Fact 3** guarantees that, since we entered the Recovery phase before finding the best group, we must have $\epsilon_{L^{(0)}} \geq \epsilon_{l_{\Delta_{\neq}}}$. On the other hand, **Fact 4** ensures that $\epsilon_{L^{(0)}} \leq \varepsilon_T$. Thus,

$$
R_T^{Rec} \leq \frac{c\kappa(\Delta \vee \varepsilon_T)\log(T)}{\Delta_{\neq}^2}.
$$

Conclusion Combining these results, we find that

$$
R_T \leq c\left(\frac{d}{\Delta_{\min}} \vee \frac{\kappa(\Delta \vee \Delta_{\neq})}{\Delta_{\neq}^2} \vee \frac{\kappa(\Delta \vee \varepsilon_T)}{\Delta_{\neq}^2}\right)\log(T)
$$

when $T \geq k$. Using Lemma 8, we get that $\kappa(\Delta \vee \Delta_{\neq}) \vee \kappa(\Delta \vee \varepsilon_T) \leq \kappa(\Delta \vee \Delta_{\neq} \vee \varepsilon_T)$, which concludes the proof of the results.

### C.1.3 Outline of the Proof of Theorem 4

We outline the main ingredients used to prove Theorem 4. Theorem 3 relies on similar arguments.

To prove the lower bounds, we need to construct two close problem instances with optimal actions belonging to different groups - to obtain the part of the lower bound involving $\Delta_{\neq}$ - and in addition we must also create confusing instances with different optimal actions belonging to a same group - to obtain the part of the lower bound involving $\Delta_{\min}$. This is done by considering the following set of actions and of problems.

**Lemma 10.** *Set* $\mathcal{A} = \left\{ \begin{pmatrix} x_1 \\ z_{x_1} \end{pmatrix}, ..., \begin{pmatrix} x_{d+1} \\ z_{x_{d+1}} \end{pmatrix} \right\}$, *where* $\begin{pmatrix} x_i \\ z_{x_i} \end{pmatrix} = e_i + e_{d+1}$, *for* $i \in \{2, ..., \lfloor d/2 \rfloor\}$, $\begin{pmatrix} x_i \\ z_{x_i} \end{pmatrix} = e_i - e_{d+1}$ *for* $i \in \{\lfloor d/2 \rfloor + 1, ..., d\}$, *and* $\begin{pmatrix} x_{d+1} \\ z_{x_{d+1}} \end{pmatrix} = -\left(1 - \frac{2}{\sqrt{\kappa_*}+1}\right) e_1 - e_{d+1}$. *It holds that*

$$
\min_{\pi \in \mathcal{P}^{\mathcal{A}}_{e_{d+1}}} \left\{ e_{d+1}^\top \left( \sum_{\left(\begin{smallmatrix} x \\ z \end{smallmatrix}\right) \in \mathcal{A}} \pi(x) \begin{pmatrix} x \\ z_x \end{pmatrix} \begin{pmatrix} x \\ z_x \end{pmatrix}^\top \right)^+ e_{d+1} \right\} = \kappa_*.
$$

We also define the following parameters:

$$
\begin{aligned}
\gamma^{(1)} &= \frac{1 + \Delta_{\neq} - \Delta_{\min}}{2} \left( \sum_{1 \le j \le \lfloor d/2 \rfloor} e_j \right) + \frac{1 - \Delta_{\neq} - \Delta_{\min}}{2} \left( \sum_{\lfloor d/2 \rfloor + 1 \le j \le d} e_j \right) \\
&\quad + \Delta_{\min} e_1 + \Delta_{\min} e_{\lfloor d/2 \rfloor + 1} \\
\gamma^{(i)} &= \gamma^{(1)} + 2\Delta_{\min} e_i + 2\Delta_{\min} e_{\lfloor d/2 \rfloor + i} \quad \forall i \in \{2, ..., \lfloor d/2 \rfloor\} \\
\gamma^{(\lfloor d/2 \rfloor + 1)} &= \frac{1 - \Delta_{\neq} - \Delta_{\min}}{2} \left( \sum_{1 \le j \le \lfloor d/2 \rfloor} e_j \right) + \frac{1 + \Delta_{\neq} - \Delta_{\min}}{2} \left( \sum_{\lfloor d/2 \rfloor + 1 \le j \le d} e_j \right) \\
&\quad + \Delta_{\min} e_1 + \Delta_{\min} e_{\lfloor d/2 \rfloor + 1}.
\end{aligned}
$$

The bias parameters are given by $\omega^{(i)} = -\frac{\Delta_{\neq}}{2} \ \forall i \in \{1, ..., \lfloor d/2 \rfloor\}$, and $\omega^{(\lfloor d/2 \rfloor + 1)} = \frac{\Delta_{\neq}}{2}$. The parameters $\theta^{(i)} = \begin{pmatrix} \gamma^{(i)} \\ \omega^{(i)} \end{pmatrix}$ characterize $\lfloor d/2 \rfloor + 1$ problems, with noise distribution i.i.d. $\mathcal{N}(0, 1)$. We write **Problem i** for the problem characterized by $\theta^{(i)}$. Note that by construction and for any $i \in \{1, ..., \lfloor d/2 \rfloor + 1\}$, we have that $\theta^{(i)} \in \Theta^{\mathcal{A}}_{\Delta_{\min}, \Delta_{\neq}}$.

The following facts hold:

- For any $i \in \{1, ..., \lfloor d/2 \rfloor + 1\}$, action $x_i$ is the unique optimal action in **Problem i**. Since $1/2 \ge \Delta_{\neq} \ge \Delta_{\min}$, sampling any other (sub-optimal) action leads to an instantaneous regret of at least $\Delta_{\min}$. Moreover, choosing an action in the group $-z_i$ leads to an instantaneous regret of at least $\Delta_{\neq}$.

- In **Problem i** for any $i \in \{1, ..., \lfloor d/2 \rfloor + 1\}$, action $d+1$ is very sub-optimal and sampling it leads to an instantaneous regret higher than $(1 - 2/(\sqrt{\kappa_*} + 1))(1 - \Delta_{\neq} + \Delta_{\min}) + (1 + \Delta_{\neq} + \Delta_{\min})/2 \ge 1/2$, since $\kappa_* \ge 1$ and $1/2 \ge \Delta_{\neq} \ge \Delta_{\min}$. This action is the worst action in all problems.

- Many actions are such that their distributions are the same across problems. More specifically:

  - For any $i \in \{2, ..., \lfloor d/2 \rfloor\}$, between **Problem 1** and **Problem i**, the only actions that provide different evaluations when sampled are action $i$ and action $\lfloor d/2 \rfloor + i$, and the mean difference between the evaluations in both cases is $2\Delta_{\min}$.

  - Between **Problem 1** and **Problem** $\lfloor d/2 \rfloor + 1$, the only actions that provide different evaluations when sampled is action $d+1$, and the mean gap in this case is $\frac{2}{\sqrt{\kappa_*}+1} \Delta_{\neq} := \alpha \Delta_{\neq}$.

The proof is then divided in two parts, one part for proving the part of the bound depending on $\Delta_{\min}$ and one part for proving the part of the bound depending on $\Delta_{\neq}$.

**Part of the bound depending on $\Delta_{\min}$.** This part of the proof is obtained using classical arguments for $K$-armed bandit problems. For $i \in \{2, ..., \lfloor d/2 \rfloor\}$, all actions but $x_i$ and $x_{\lfloor d/2 \rfloor + i}$ have the same feedback under **Problem 1** and **Problem i**. On the other hand, the average feedback for actions $x_i$ and $x_{\lfloor d/2 \rfloor + i}$ differs by $2\Delta_{\min}$, so either action needs to be selected approximately $\frac{\log(T)}{\Delta_{\min}^2}$ times in order to identify the problem at hand with high enough probability. In **Problem 1**, the simple regret for choosing $x_i$ or $x_{\lfloor d/2 \rfloor + i}$ is larger than $\Delta_{\min}$, so the total regret obtained when doing this is at least of the order $\frac{\log(T)}{\Delta_{\min}}$. Summing over the different actions $i$ leads to a lower bound of the order $\frac{d \log(T)}{\Delta_{\min}}$.

**Part of the bound depending on $\Delta_{\neq}$.** To obtain the second part of the lower bound, we note that all actions but $x_{d+1}$ have the same feedback under **Problem 1** and **Problem** $\lfloor d/2 \rfloor + 1$. The average feedback for actions $x_{d+1}$ differs by $\alpha \Delta_{\neq}$ under these parameters, so action $x_{d+1}$ needs to be selected approximately $\frac{\log(T)}{\alpha^2 \Delta_{\neq}^2} \gtrsim \frac{\log(T) \kappa_*}{\Delta_{\neq}^2}$ times to identify the problem at hand with high enough probability. Since selecting action $x_{d+1}$ leads to an simple regret larger than $1/2$ under **Problem 1**, this implies that the regret must be at least of the order $\frac{\kappa_* \log(T)}{\Delta_{\neq}^2}$.

**Bounds on $\kappa(\Delta)$** Finally, the following lemma allows to express $\kappa(\Delta)$ as a function of $\kappa_*$.

**Lemma 11.** *For any $i \in \{1, ..., \lfloor d/2 \rfloor + 1\}$, the gap vector $\Delta$ verifies*

$$\kappa(\Delta) = \frac{(1 + \sqrt{\kappa_*})^2 \Delta_{d+1}}{4}$$

*where $\Delta_{d+1} = \max_i (x_i - x_{d+1})^\top \gamma^{(i)}$.*

On the one hand, since $\kappa_* \geq 1$, we see that $\kappa_* \leq (1 + \sqrt{\kappa_*})^2 \leq 4\kappa_*$. On the other hand, $1/2 \leq \Delta_{d+1} \leq 2$, so $\kappa(\Delta) \in \left[\frac{\kappa_*}{8}, 2\kappa_*\right]$.

### C.2 Proof of Theorem 1

We begin by defining for $z \in \{-1, 0, +1\}$

$$L^{(z)} = \max \left\{ l \geq 1 : \text{Explore}_l^{(z)} = \text{True} \right\}$$

the largest integer $l$ such that $\text{Explore}_l^{(z)} = \text{True}$. Recall that $\kappa_*$ is the $e_{d+1}$-optimal variance. By definition of the algorithm, for all $l \leq L^{(0)} + 1$, $\widehat{\Delta}^l \leq 2$, so $\kappa(\widehat{\Delta}^l) \leq 2\kappa_*$. Now, let us also define

$$L_T = \max \left\{ l \geq 1 : \epsilon_l > \left( \frac{2\kappa_* \log(T)}{T} \right)^{1/3} \right\}.$$

Then, if Recovery$\neq \emptyset$, we must have $L^{(0)} \geq L_T$. Moreover, we see that since $\epsilon_{L_T} = 2^{2-L_T}$, we have $L_T \leq 2 + \frac{\log_2(T/(2\kappa_* \log(T)))}{3} \leq 3\log_2(T)$ when $T > 1$.

We define a "bad" event $\mathcal{F}$, such that, on $\overline{\mathcal{F}}$, our estimators $\widehat{\gamma}_l^{(z)}$ and $\widehat{\omega}_l^{(z)}$ are close to the true parameters $\gamma^*$ and $\omega^*$ for all rounds $l$. More precisely, let

$$\mathcal{F} = \bigcup_{l \geq 1} \mathcal{F}_l, \tag{10}$$

where for $l \geq 1$

$$\mathcal{F}_l = \left\{ \exists z \in \{-1, 1\} \text{ such that Explore}_l^{(z)} = \text{True, and } x \in \mathcal{X}_l^{(z)} \text{ such that } \left| \begin{pmatrix} \widehat{\gamma}_l^{(z)} - \gamma^* \\ \widehat{\omega}_l^{(z)} - \omega^* \end{pmatrix}^\top \begin{pmatrix} x \\ z_x \end{pmatrix} \right| \geq \epsilon_l \right\}$$

$$\bigcup \left\{ \text{Explore}_l^{(0)} = \text{True and } \left| \widehat{\omega}_l^{(0)} - \omega^* \right| \geq \epsilon_l \right\}.$$

Then, the regret decomposes as

$$R_T \leq \sum_{t \leq T} \mathbb{E}_{|\overline{\mathcal{F}}} \left[ (x^* - x_t)^\top \gamma^* \right] + 2T\mathbb{P}[\mathcal{F}]. \tag{11}$$

The following lemma relies on concentration of Gaussian variables to bound the probability of the event $\mathcal{F}$.

**Lemma 12.** $\mathbb{P}(\mathcal{F}) \leq 2\delta$.

Now, the first term of (11) can be decomposed as

$$\sum_{t \leq T}(x^* - x_t)^\top \gamma^* \leq \sum_{z \in \{-1,0,+1\}} \sum_{l=1}^{L^{(z)}+1} \sum_{t \in \mathrm{Exp}_l^{(z)}}(x^* - x_t)^\top \gamma^* + \sum_{t \in \mathrm{Recovery}}(x^* - x_t)^\top \gamma^*,$$

where we use as convention that the sum over an empty set is null. Note that for $z \in \{-1, +1\}$, during the phase $\mathrm{Exp}_l^{(z)}$ the algorithm only samples actions from $\mathcal{X}_l^{(z)}$. By contrast, during the phase $\mathrm{Exp}_l^{(0)}$, even actions eliminated from the sets $\mathcal{X}_l^{(z)}$ can be sampled. Finally, if the algorithm stops during phase $\mathrm{Exp}_{L^{(0)}+1}^{(0)}$, but does not have enough budget to complete the last $\widehat{\Delta}^l$-optimal Exploration and Elimination Phase, it samples the remaining actions in the set $\mathcal{X}_{L^{(0)}+2}^{(-1)} \cup \mathcal{X}_{L^{(0)}+2}^{(+1)}$. Hence, the first term of (11) can be upper-bounded by

$$\sum_{t \leq T}(x^* - x_t)^\top \gamma^* \leq \sum_{z \in \{-1,+1\}} \sum_{l=1}^{L_T} \left( \sum_{x \in \mathcal{X}_l^{(z)}} \mu_l^{(z)}(x) \right) \max_{x \in \mathcal{X}_l^{(z)}}(x^* - x)^\top \gamma^* \qquad (12)$$

$$+ \sum_{z \in \{-1,+1\}} \sum_{l=L_T+1}^{L^{(z)}+1} \sum_{t \in \mathrm{Exp}_l^{(z)}}(x^* - x_t)^\top \gamma^* + \sum_{t \in \mathrm{Recovery}}(x^* - x_t)^\top \gamma^*$$

$$+ \sum_{l=1}^{L^{(0)}} \sum_{x \in \mathcal{X}} \mu_l^{(0)}(x)\Delta_x + \mathbb{1}\left\{\mathrm{Explore}_{L^{(0)}+1}^{(0)} = \mathrm{False}\right\} \sum_{t \in \mathrm{Exp}_{L^{(0)}+1}^{(0)}} \max_{x \in \mathcal{X}_{L^{(0)}+2}^{(-1)} \cup \mathcal{X}_{L^{(0)}+2}^{(+1)}}(x^* - x)^\top \gamma^*.$$

We begin by bounding the sum of the regret corresponding to the Recovery phase and to the phases $\mathrm{Exp}_L^{(z)}$ for $z \in \{-1, +1\}$ and $l > L_T$ on the event $\overline{\mathcal{F}}$.

**Bound on** $\sum_{z \in \{-1,+1\}} \sum_{l=L_T+1}^{L^{(z)}+1} \sum_{t \in \mathrm{Exp}_l^{(z)}}(x^* - x_t)^\top \gamma^* + \sum_{t \in \mathrm{Recovery}}(x^* - x_t)^\top \gamma^*.$

**Lemma 13.** *Let* $x^* \in \mathrm{argmax}_{x \in \mathcal{X}} x^\top \gamma^*$ *be an optimal action. Then, on the event* $\overline{\mathcal{F}}$ *defined in Equation* (10)*, for* $l \geq 1$ *such that* $\mathrm{Explore}_l^{(z_{x^*})} = \mathit{True}$,

$$\mathcal{X}_{l+1}^{(z_{x^*})} \subset \left\{ x \in \mathcal{X}_1^{(z_{x^*})} : (x^* - x)^\top \gamma^* < 10\epsilon_{l+1} \right\}. \qquad (13)$$

*Moreover, for* $l \geq 1$ *such that* $\mathrm{Explore}_l^{(-z_{x^*})} = \mathit{True}$,

$$\mathcal{X}_{l+1}^{(-z_{x^*})} \subset \left\{ x \in \mathcal{X}_1^{(-z_{x^*})} : (x^* - x)^\top \gamma^* < 42\epsilon_{l+1} \right\}.$$

Recall that if $\mathrm{Recovery} \neq \emptyset$, $L^{(0)} \geq L_T$. Then, all actions sampled during the Recovery phase belong to $\mathcal{X}_{l+1}^{(-1)} \cup \mathcal{X}_{l+1}^{(+1)}$ for some $l \geq L_T$. Lemma 13 shows that, on $\overline{\mathcal{F}}$, for $l \geq L_T$, the actions in $\mathcal{X}_{l+1}^{(z)}$ are sub-optimal by at most $42\epsilon_{L_T+1}$. Then, we get that on the event $\overline{\mathcal{F}}$,

$$\sum_{z \in \{-1,+1\}} \sum_{l=L_T+1}^{L^{(z)}+1} \sum_{t \in \mathrm{Exp}_l^{(z)}}(x^* - x_t)^\top \gamma^* + \sum_{t \in \mathrm{Recovery}}(x^* - x_t)^\top \gamma^* \leq T \times 42\epsilon_{L_T+1}$$

$$\leq 53\kappa_*^{1/3} T^{2/3} \log(T)^{1/3} \quad (14)$$

**Bound on** $\sum_{l=1}^{L^{(0)}} \sum_{x\in\mathcal{X}} \mu_l^{(0)}(x)\Delta_x + \mathbb{1}\left\{\text{Explore}_{L^{(0)}+1}^{(0)} = \text{False}\right\} \sum_{t\in\text{Exp}_{L^{(0)}+1}^{(0)}} \max_{x\in\mathcal{X}_{L^{(0)}+2}^{(-1)}\cup\mathcal{X}_{L^{(0)}+2}^{(+1)}} (x^* - x)^\top\gamma^*$.

We begin by bounding $\mathbb{1}\left\{\text{Explore}_{L^{(0)}+1}^{(0)} = \text{False}\right\} \sum_{t\in\text{Exp}_{L^{(0)}+1}^{(0)}} \max_{x\in\mathcal{X}_{L^{(0)}+2}^{(-1)}\cup\mathcal{X}_{L^{(0)}+2}^{(+1)}} (x^* - x)^\top\gamma^*$. Recall

that $n_{L^{(0)}+1}^{(0)} = \sum_{x\in\mathcal{X}} \mu_{L^{(0)}+1}^{(0)}(x)$ is the budget that would be necessary to complete the $\widehat{\Delta}^l$-optimal

Exploration and Elimination phase at phase $L^{(0)} + 1$. On the one hand, Lemma 13 implies that on the event $\overline{\mathcal{F}}$,

$$\mathbb{1}\left\{\text{Explore}_{L^{(0)}+1}^{(0)} = \text{False}\right\} \sum_{t\in\text{Exp}_{L^{(0)}+1}^{(0)}} \max_{x\in\mathcal{X}_{L^{(0)}+2}^{(-1)}\cup\mathcal{X}_{L^{(0)}+2}^{(+1)}} (x^* - x)^\top\gamma^* \leq 42 n_{L^{(0)}+1}^{(0)}\epsilon_{L^{(0)}+2} \leq 21 n_{L^{(0)}+1}^{(0)}\epsilon_{L^{(0)}+1}.$$

On the other hand, for all $l \leq L^{(0)} + 1$, the definition of $\widehat{\Delta}^l$ implies that $\widehat{\Delta}_x^l \geq \epsilon_l$ for all $x \in \mathcal{X}$. Therefore, $21 n_{L^{(0)}+1}^{(0)}\epsilon_{L^{(0)}+1} \leq 21 n_{L^{(0)}+1}^{(0)} \min_x \widehat{\Delta}_x^{L^{(0)}+1}$. This implies that on $\overline{\mathcal{F}}$,

$$\mathbb{1}\left\{\text{Explore}_{L^{(0)}+1}^{(0)} = \text{False}\right\} \sum_{t\in\text{Exp}_{L^{(0)}+1}^{(0)}} \max_{x\in\mathcal{X}_{L^{(0)}+2}^{(-1)}\cup\mathcal{X}_{L^{(0)}+2}^{(+1)}} (x^* - x)^\top\gamma^* \leq 21 \sum_{x\in\mathcal{X}} \mu_{L^{(0)}+1}^{(0)}(x)\widehat{\Delta}_x^{L^{(0)}+1}. \tag{15}$$

Next, to bound the remaining terms of Equation (12), we bound the regret $\sum_{x\in\mathcal{X}} \mu_l^{(0)}(x)\Delta_x$ of exploration phase $\text{Exp}_l^{(0)}$ using the following lemma.

**Lemma 14.** *For all $l > 0$, and $z \in \{-1, +1\}$, we have*

$$\sum_{x\in\mathcal{X}_l^{(z)}} \mu_l^{(z)}(x) \leq \frac{2(d+1)}{\epsilon_l^2} \log\left(\frac{kl(l+1)}{\delta}\right) + \frac{(d+1)(d+2)}{2}.$$

*and on $\overline{\mathcal{F}}$, we have*

$$\sum_{x\in\mathcal{X}} \mu_l^{(0)}(x)\Delta_x \leq \sum_{x\in\mathcal{X}} \mu_l^{(0)}(x)\widehat{\Delta}_x^l \leq \frac{2\kappa(\widehat{\Delta}^l)}{\epsilon_l^2} \log\left(\frac{l(l+1)}{\delta}\right) + 2(d+1).$$

Then, Equation (15) and Lemma 14 imply that on $\overline{\mathcal{F}}$

$$\sum_{l=1}^{L^{(0)}} \sum_{x\in\mathcal{X}} \mu_l^{(0)}(x)\Delta_x + \mathbb{1}\left\{\text{Explore}_{L^{(0)}+1}^{(0)} = \text{False}\right\} \sum_{t\in\text{Exp}_{L^{(0)}+1}^{(0)}} \max_{x\in\mathcal{X}_{L^{(0)}+2}^{(-1)}\cup\mathcal{X}_{L^{(0)}+2}^{(+1)}} (x^* - x)^\top\gamma^*$$

$$\leq 21 \sum_{l=1}^{L^{(0)}+1} \sum_{x\in\mathcal{X}} \mu_l^{(0)}(x)\widehat{\Delta}_x^l$$

$$\leq 42 \sum_{l=1}^{L^{(0)}+1} \frac{\kappa(\widehat{\Delta}^l)}{\epsilon_l^2} \log\left(\frac{l(l+1)}{\delta}\right) + 42(d+1)(L^{(0)} + 1)$$

$$\tag{16}$$

We rely on the following Lemma to bound $\kappa(\widehat{\Delta}^l)$.

**Lemma 15.** *On $\overline{\mathcal{F}}$, we have for any $l \geq 1$ and any $\tau > 0$*

$$\kappa(\widehat{\Delta}^l) \leq 513\left(1 + \frac{\epsilon_l}{\tau}\right)\kappa(\Delta \vee \tau).$$

*and*

$$\kappa(\widehat{\Delta}^l) \geq \kappa(\Delta \vee \epsilon_l).$$

Lemma 14 and Lemma 15 with $\tau = \epsilon_{L^{(0)}}$ imply that on $\overline{\mathcal{F}}$,

$$\sum_{l=1}^{L^{(0)}+1} \frac{\kappa(\widehat{\Delta}^l)}{\epsilon_l^2} \log\left(\frac{l(l+1)}{\delta}\right) \leq 513\kappa(\Delta \vee \epsilon_{L^{(0)}}) \log\left(\frac{(L^{(0)}+1)(L^{(0)}+2)}{\delta}\right) \left(\sum_{l=1}^{L^{(0)}+1} \frac{1}{\epsilon_l^2} + \sum_{l=1}^{L^{(0)}+1} \frac{1}{\epsilon_l \epsilon_{L^{(0)}}}\right)$$

$$\leq 513\kappa(\Delta \vee \epsilon_{L^{(0)}}) \log\left(\frac{6L^{(0)}}{\delta}\right) \left(\frac{16}{\epsilon_{L^{(0)}}^2} + \frac{4}{\epsilon_{L^{(0)}}^2}\right)$$

$$\leq 10260 \log\left(\frac{6L^{(0)}}{\delta}\right) \frac{\kappa(\widehat{\Delta}^{L^{(0)}})}{\epsilon_{L^{(0)}}^2} \tag{17}$$

where the last line follows from the second claim of Lemma 15. Now, by definition of $L^{(0)}$, $\epsilon_{L^{(0)}} \geq \left(\kappa(\widehat{\Delta}^{L^{(0)}}) \log(T)/T\right)^{1/3}$. Then, Equation (17) implies that

$$\sum_{l=1}^{L^{(0)}+1} \frac{\kappa(\widehat{\Delta}^l)}{\epsilon_l^2} \log\left(\frac{l(l+1)}{\delta}\right) \leq 10260 \log\left(\frac{6L^{(0)}}{\delta}\right) \kappa(\widehat{\Delta}^{L^{(0)}})^{1/3} \log(T)^{-2/3} T^{2/3}. \tag{18}$$

Moreover, we observe that during each phase $l$, but the last one, we sample at least

$$\max_{z \in \{-1,1\}} \sum_{x \in \mathcal{X}_l^{(z)}} \tau_{l,x}^{(z)} \geq \frac{2(d+1)}{\delta_l^2} \log(kl(l+1)/\delta)$$

actions during the G-optimal explorations, so the number of phases $L^{(0)}$ is never larger than

$$\ell_T = 1 \vee \log_4(T).$$

Using this remark, together with Equations (16) and (18), we find that on $\overline{\mathcal{F}}$

$$\sum_{l=1}^{L^{(0)}} \sum_{x \in \mathcal{X}} \mu_l^{(0)}(x)\widehat{\Delta}_x^l \quad + \quad \mathbb{1}\left\{\text{Explore}_{L^{(0)}+1}^{(0)} = \text{False}\right\} \sum_{t \in \text{Exp}_{L^{(0)}+1}^{(0)}} \max_{x \in \mathcal{X}_{L^{(0)}+2}^{(-1)} \cup \mathcal{X}_{L^{(0)}+2}^{(+1)}} (x^* - x)^\top \gamma^*$$

$$\leq 2^{19} \log\left(\frac{6L^{(0)}}{\delta}\right) \kappa(\widehat{\Delta}^{L^{(0)}}) T^{2/3} \log(T)^{-2/3} + 42\ell_T. \tag{19}$$

**Bound on** $\sum_{z \in \{-1,+1\}} \sum_{l=1}^{L_T} \left(\sum_{x \in \mathcal{X}_l^{(z)}} \mu_l^{(z)}(x)\right) \max_{x \in \mathcal{X}_l^{(z)}} (x^* - x)^\top \gamma^*$. We bound the remaining term in Equation (12) using the first claim in Lemma 14 and Lemma 13. On $\overline{\mathcal{F}}$,

$$\sum_{z \in \{-1,+1\}} \sum_{l=1}^{L_T} \left(\sum_{x \in \mathcal{X}_l^{(z)}} \mu_l^{(z)}(x)\right) \max_{x \in \mathcal{X}_l^{(z)}} (x^* - x)^\top \gamma^* \leq 2\sum_{l=1}^{L_T} \left(\frac{2(d+1)}{\epsilon_l^2} \log\left(\frac{kl(l+1)}{\delta}\right) + \frac{(d+1)(d+2)}{2}\right) 42\epsilon_l$$

$$\leq \frac{336(d+1)}{\epsilon_{L_T}} \log\left(\frac{kL_T(1+L_T)}{\delta}\right) + 168(d+1)(d+2)$$

$$\leq 267(d+1)\kappa_*^{-1/3} T^{1/3} \log(T)^{-1/3} \log\left(\frac{kL_T(1+L_T)}{\delta}\right)$$

$$+ 168(d+1)(d+2). \tag{20}$$

Combing Equations (11), (12), (14), (19), and (20), and using $\delta = T^{-1}$, $\kappa(\widehat{\Delta}^{L^{(0)}}) \leq \kappa_*$ and $L_T \leq 4T/\log(2)$, we get for all $T \geq 1$

$$R_T \leq C \left(\kappa_*^{1/3} T^{2/3} \log(T)^{1/3} + (d \vee \kappa_*) \log(T) + d^2 + d\kappa_*^{-1/3} T^{1/3} \log(kT) \log(T)^{-1/3}\right)$$

for some absolute constant $C > 0$. Finally, for

$$T \geq \frac{((d \vee \kappa_*)^{3/2} \log(T)) \vee d^3}{\sqrt{\kappa_*}} \vee \frac{(d \log(kT))^3}{(\kappa_* \log(T))^2},$$

we get

$$R_T \leq C' \kappa_*^{1/3} T^{2/3} \log(T)^{1/3}.$$

## C.3 Proof of Theorem 2

The beginning of the proof of Theorem 2 follows the same lines as the proof of Theorem 1. We begin by decomposing the regret as

$$R_T \leq \sum_{t \leq T} \mathbb{E}_{|\overline{\mathcal{F}}} \left[ (x^* - x_t)^\top \gamma^* \right] + 2T\mathbb{P}\left[\mathcal{F}\right]. \tag{21}$$

where $\mathcal{F}$ is defined in Equation (10). On the one hand, Lemma 12 implies $T\mathbb{P}\left[\mathcal{F}\right] \leq 2\delta T$. Then, Equation (21) implies

$$R_T \leq 4\delta T + \mathbb{E}_{|\overline{\mathcal{F}}} \left[ \sum_{z \in \{-1,+1\}} \sum_{l \geq 1}^{L^{(z)}+1} \sum_{t \in \mathrm{Exp}_l^{(z)}} (x^* - x_t)^\top \gamma^* \right] + \mathbb{E}_{|\overline{\mathcal{F}}} \left[ \sum_{t \in \mathrm{Recovery}} (x^* - x_t)^\top \gamma^* \right] \tag{22}$$

$$+ \mathbb{E}_{|\overline{\mathcal{F}}} \left[ \sum_{l=1}^{L^{(0)}} \sum_{x \in \mathcal{X}} \mu_l^{(0)}(x)\Delta_x \right] + \mathbb{E}_{|\overline{\mathcal{F}}} \left[ \mathbb{1}\left\{\mathrm{Explore}_{L^{(0)}+1}^{(0)} = \mathrm{False}\right\} \sum_{t \in \mathrm{Exp}_{L^{(0)}+1}^{(0)}} \max_{x \in \mathcal{X}_{L^{(0)}+2}^{(-1)} \cup \mathcal{X}_{L^{(0)}+2}^{(+1)}} (x^* - x)^\top \gamma^* \right]$$

where $\mathcal{F}$ is defined in Equation (10), and where we used the convention that the sum over an empty set is null.

**Bound on** $\mathbb{1}\left\{\mathrm{Explore}_{L^{(0)}+1}^{(0)} = \mathrm{False}\right\} \sum_{t \in \mathrm{Exp}_{L^{(0)}+1}^{(0)}} \max_{x \in \mathcal{X}_{L^{(z)}+1}} (x^* - x)^\top \gamma^*$.

Similarly to the proof of Theorem 1, we use Lemma 13 and Lemma 15 to show that on $\overline{\mathcal{F}}$

$$\mathbb{1}\left\{\mathrm{Explore}_{L^{(0)}+1}^{(0)} = \mathrm{False}\right\} \sum_{t \in \mathrm{Exp}_{L^{(0)}+1}^{(0)}} \max_{x \in \mathcal{X}_{L^{(z)}+1}} (x^* - x)^\top \gamma^* \leq 21 \sum_{x \in \mathcal{X}} \mu_{L^{(0)}+1}^{(0)}(x)\widehat{\Delta}_x^{L^{(0)}+1}. \tag{23}$$

**Bound on** $\sum_{z \in \{-1,+1\}} \sum_{l \geq 1}^{L^{(z)}+1} \sum_{t \in \mathrm{Exp}_l^{(z)}} (x^* - x_t)^\top \gamma^*$.

Lemma 13 shows that for $l \leq L^{(z)}$, the actions in $\mathcal{X}_{l+1}^{(z)}$ are sub-optimal by at most an additional factor at most $21\epsilon_l$. Let us set $l_{\Delta_{\min}} = \lceil -\log_2(\Delta_{\min}/21)\rceil$, so that

$$\frac{\Delta_{\min}}{42} \leq \epsilon_{l_{\Delta_{\min}}} \leq \frac{\Delta_{\min}}{21}.$$

For $l \geq l_{\Delta_{\min}}$, we have $\mathcal{X}_{l+1}^{(-1)} \cup \mathcal{X}_{l+1}^{(+1)} = \{x_{z^*}\}$. Thus, $l^{(-z_{x^*})} \leq l_{\Delta_{\min}}$, and for $l \geq l_{\Delta_{\min}}$, the algorithm selects only $x^*$ during the phase $\mathrm{Exp}_l^{(z^*)}$. Then, combining Lemmas 14 and 13, and the fact that $L^{(z)} + 1 \leq \ell_T$, we find that, on $\overline{\mathcal{F}}$,

$$\sum_{z \in \{-1,+1\}} \sum_{l=1}^{L^{(z)}+1} \sum_{t \in \mathrm{Exp}_l^{(z)}} (x^* - x_t)^\top \gamma^* \leq \sum_{z \in \{-1,+1\}} \sum_{l=1}^{l_{\Delta_{\min}}+1 \wedge \ell_T} \left( \sum_{x \in \mathcal{X}_l^{(z)}} \mu_l^{(z)}(x) \right) \max_{x \in \mathcal{X}_l^{(z)}} (x^* - x)^\top \gamma^*$$

$$\leq 2 \sum_{l=1}^{l_{\Delta_{\min}}+1 \wedge \ell_T} \left( \frac{2(d+1)}{\epsilon_l^2} \log\left( \frac{kl(l+1)}{\delta} \right) + \frac{(d+1)(d+2)}{2} \right) 42\epsilon_l$$

$$\leq 84(d+1)(d+2) + \epsilon_{l_{\Delta_{\min}}}^{-1} \times 672(d+1)\log\left( \frac{k(1+\ell_T)(2+\ell_T)}{\delta} \right)$$

$$\leq 84(d+1)(d+2) + \frac{28224(d+1)}{\Delta_{\min}}\log\left( \frac{k(1+\ell_T))(2+\ell_T))}{\delta} \right) \tag{24}$$

**Bound on** $\sum_{t \in \mathrm{Recovery}} (x^* - x_t)^\top \gamma^* + \sum_{l=1}^{L^{(0)}} \sum_{x \in \mathcal{X}} \mu_l^{(0)}(x)\Delta_x + \sum_{x \in \mathcal{X}} \mu_{L^{(0)}+1}^{(0)}(x)\widehat{\Delta}_x^{L^{(0)}+1}$.

We use the following lemma to bound the number of phases necessary to eliminate the sub-optimal group.

**Lemma 16.** *On the event $\overline{\mathcal{F}}$ defined in Equation (10), for $l \geq 1$ such that $\epsilon_l \leq \frac{\Delta_{\neq}}{8}$ and Explore$_L^{(0)} = $ True, $\widehat{z^*}_{l+1} = z_{x^*}$.*

Let $l_{\Delta_{\neq}} = \lceil -\log(\Delta_{\neq}/8)/\log(2) \rceil$ be such that

$$\frac{\Delta_{\neq}}{16} \leq \epsilon_{l_{\Delta_{\neq}}} \leq \frac{\Delta_{\neq}}{8}. \tag{25}$$

Lemma 16 implies that on $\overline{\mathcal{F}}$, $L^{(0)} \leq l_{\Delta_{\neq}}$.

To bound the remaining terms, we consider two cases, corresponding to Recovery$= \emptyset$ and Recovery$\neq \emptyset$.

**Case 1:** Recovery$= \emptyset$. Our case assumption implies that

$$\sum_{t \in \text{Recovery}} (x^* - x_t)^\top \gamma^* = 0. \tag{26}$$

Lemma 15 implies that

$$\sum_{l=1}^{L^{(0)}} \sum_{x \in \mathcal{X}} \mu_l^{(0)}(x) \Delta_x + \sum_{x \in \mathcal{X}} \mu_{L^{(0)}+1}^{(0)}(x) \widehat{\Delta}_x^{L^{(0)}+1} \leq \sum_{l=1}^{L^{(0)}+1} \sum_{x \in \mathcal{X}} \mu_l^{(0)}(x) \widehat{\Delta}_x^l.$$

Moreover, $L^{(0)} \leq l_{\Delta_{\neq}} \wedge \ell_T$, so on $\overline{\mathcal{F}}$

$$\sum_{l=1}^{L^{(0)}+1} \sum_{x \in \mathcal{X}} \mu_l^{(0)}(x) \widehat{\Delta}_x^l \leq \sum_{l=1}^{(l_{\Delta_{\neq}} \wedge \ell_T)+1} \sum_{x \in \mathcal{X}} \mu_l^{(0)}(x) \widehat{\Delta}_x^l.$$

Using Lemma 14, we find that on $\overline{\mathcal{F}}$

$$\sum_{l=1}^{(l_{\Delta_{\neq}} \wedge \ell_T)+1} \sum_{x \in \mathcal{X}} \mu_l^{(0)}(x) \widehat{\Delta}_x^l \leq \sum_{l=1}^{(l_{\Delta_{\neq}} \wedge \ell_T)+1} \frac{2\kappa(\widehat{\Delta}^l)}{\epsilon_l^2} \log\left(\frac{l(l+1)}{\delta}\right) + 2(d+1)(\ell_T+1)$$

$$\leq 2\log\left(\frac{(\ell_T+1)(\ell_T+2)}{\delta}\right) \sum_{l=1}^{l_{\Delta_{\neq}}+1} \frac{\kappa(\widehat{\Delta}^l)}{\epsilon_l^2} + 2(d+1)(\ell_T+1).$$

Using Lemma 15 with $\tau = \Delta_{\neq}$ and (25), we have on $\overline{\mathcal{F}}$

$$\sum_{l=1}^{l_{\Delta_{\neq}}+1} \frac{\kappa(\widehat{\Delta}^l)}{\epsilon_l^2} \leq 513\kappa(\Delta \vee \Delta_{\neq}) \sum_{l=1}^{l_{\Delta_{\neq}}+1} \left(\epsilon_l^{-2} + \epsilon_l^{-1}/\Delta_{\neq}\right)$$

$$\leq \frac{2^{18}\kappa(\Delta \vee \Delta_{\neq})}{\Delta_{\neq}^2}.$$

We obtain on $\overline{\mathcal{F}}$

$$\sum_{l=1}^{L^{(0)}+1} \sum_{x \in \mathcal{X}} \mu_l^{(0)}(x) \widehat{\Delta}_x^l \leq 2^{19} \log\left(\frac{(\ell_T+1)(\ell_T+2)}{\delta}\right) \frac{\kappa(\Delta \vee \Delta_{\neq})}{\Delta_{\neq}^2} + 2(d+1)(\ell_T+1) \tag{27}$$

Combining Equations (24), (23), (26), and (27), we find that on $\overline{\mathcal{F}}$, when Recovery= $\emptyset$, there exsists an absolute constant $c > 0$ such that for $\delta = T^{-1}$,

$$
\sum_{z \in \{-1,+1\}} \sum_{l \geq 1}^{L^{(z)}+1} \sum_{t \in \mathrm{Exp}_l^{(z)}} (x^* - x_t)^\top \gamma^* + \sum_{t \in \mathrm{Recovery}} (x^* - x_t)^\top \gamma^* + \sum_{l=1}^{L^{(0)}} \sum_{x \in \mathcal{X}} \mu_l^{(0)}(x) \Delta_x \tag{28}
$$

$$
+ \mathbb{1}\left\{\mathrm{Explore}_{L^{(0)}+1}^{(0)} = \mathrm{False}\right\} \sum_{t \in \mathrm{Exp}_{L^{(0)}+1}^{(0)}} \max_{x \in \mathcal{X}_{L^{(0)}+2}^{(-1)} \cup \mathcal{X}_{L^{(0)}+2}^{(+1)}} (x^* - x)^\top \gamma^*
$$

$$
\leq c\left(d^2 + \left(\frac{d}{\Delta_{\min}} \vee \frac{\kappa(\Delta \vee \Delta_{\neq})}{\Delta_{\neq}^2}\right) \log(T) + \frac{d}{\Delta_{\min}} \log(k)\right).
$$

**Case 2: Recovery $\neq \emptyset$.** In this case, the algorithm enters Recovery at phase $L^{(0)}$, so $\mathrm{Explore}_{L^{(0)}+1}^{(0)}$ =False and $\mathrm{Exp}_{L^{(0)}+1}^{(0)} = \emptyset$, and

$$
\mathbb{1}\left\{\mathrm{Explore}_{L^{(0)}+1}^{(0)} = \mathrm{False}\right\} \sum_{t \in \mathrm{Exp}_{L^{(0)}+1}^{(0)}} \max_{x \in \mathcal{X}_{L^{(0)}+2}^{(-1)} \cup \mathcal{X}_{L^{(0)}+2}^{(+1)}} (x^* - x)^\top \gamma^* = 0. \tag{29}
$$

Using Lemma 13, we see that

$$
\sum_{t \in \mathrm{Recovery}} (x^* - x_t)^\top \gamma^* \leq 21 T \epsilon_{L^{(0)}+1}.
$$

On the other hand, in the Recovery phase, $\epsilon_{L^{(0)}+1} \leq \left(\kappa(\widehat{\Delta}^{L^{(0)}+1}) \log(T)/T\right)^{1/3}$. Thus,

$$
\sum_{t \in \mathrm{Recovery}} (x^* - x_t)^\top \gamma^* \leq \frac{21 \kappa(\widehat{\Delta}^{L^{(0)}+1}) \log(T)}{\epsilon_{L^{(0)}+1}^2}.
$$

Now, Lemma 14 show that

$$
\sum_{l=1}^{L^{(0)}} \sum_{x \in \mathcal{X}} \mu_l^{(0)}(x) \Delta_x \leq 4 \log(2 L^{(0)} \delta^{-1}) \sum_{l=1}^{L^{(0)}} \frac{\kappa(\widehat{\Delta}^l)}{\epsilon_l^2} + 4 d L^{(0)}.
$$

Combining these results, and using $L^{(0)} \leq \ell_T$, we see that

$$
\sum_{t \in \mathrm{Recovery}} (x^* - x_t)^\top \gamma^* + \sum_{l=1}^{L^{(0)}} \sum_{x \in \mathcal{X}} \mu_l^{(0)}(x) \Delta_x \leq 4 d L^{(0)} + \left(4 \log(2 \ell_T \delta^{-1}) \vee 21 \log(T)\right) \sum_{l=1}^{L^{(0)}+1} \frac{\kappa(\widehat{\Delta}^l)}{\epsilon_l^2}. \tag{30}
$$

Using Lemma 15 with $\tau = \epsilon_{L^{(0)}}$, we see that

$$
\sum_{l=1}^{L^{(0)}+1} \frac{\kappa(\widehat{\Delta}^l)}{\epsilon_l^2} \leq 513 \sum_{l=1}^{L^{(0)}+1} \frac{\kappa(\Delta \vee \epsilon_{L^{(0)}})}{\epsilon_l^2} + 513 \sum_{l=1}^{L^{(0)}+1} \frac{\kappa(\Delta \vee \epsilon_{L^{(0)}})}{\epsilon_{L^{(0)}} \epsilon_l}
$$

$$
\leq 10260 \frac{\kappa(\Delta \vee \epsilon_{L^{(0)}})}{\epsilon_{L^{(0)}}^2}.
$$

Now, the algorithm enters the Recovery phase before finding the best group, so we must have $L^{(0)} \leq l_{\Delta_{\neq}}$. This implies that

$$
\sum_{l=1}^{L^{(0)}+1} \frac{\kappa(\widehat{\Delta}^l)}{\epsilon_l^2} \leq 2^{18} \frac{\kappa(\Delta \vee \epsilon_{L^{(0)}})}{\Delta_{\neq}^2}.
$$

Finally, note that $L^{(0)} \geq L_T$, so $\epsilon_{L^{(0)}} \leq \epsilon_{L_T} = \varepsilon_T$, and

$$\sum_{l=1}^{L^{(0)}+1} \frac{\kappa(\widehat{\Delta}^l)}{\epsilon_l^2} \leq 2^{18} \frac{\kappa(\Delta \vee \varepsilon_T)}{\Delta_{\neq}^2}. \tag{31}$$

Combining Equations (24), (29), (30), and (31), we find that on $\overline{\mathcal{F}}$, when Recovery$\neq \emptyset$, there exists an absolute constant $c > 0$ such that for $\delta = T^{-1}$,

$$\sum_{z \in \{-1,+1\}} \sum_{l \geq 1}^{L^{(z)}+1} \sum_{t \in \mathrm{Exp}_l^{(z)}} (x^* - x_t)^\top \gamma^* + \sum_{t \in \mathrm{Recovery}} (x^* - x_t)^\top \gamma^* + \sum_{l=1}^{L^{(0)}} \sum_{x \in \mathcal{X}} \mu_l^{(0)}(x)\Delta_x \tag{32}$$

$$+ \mathbb{1}\{\mathrm{Explore}_{L^{(0)}+1}^{(0)} = \mathrm{False}\} \sum_{t \in \mathrm{Exp}_{L^{(0)}+1}^{(0)}} \max_{x \in \mathcal{X}_{L^{(0)}+2}^{(-1)} \cup \mathcal{X}_{L^{(0)}+2}^{(+1)}} (x^* - x)^\top \gamma^*$$

$$\leq c \left( d^2 + \left( \frac{d}{\Delta_{\min}} \vee \frac{\kappa(\Delta \vee \varepsilon_T)}{\Delta_{\neq}^2} \right) \log(T) + \frac{d \log(k)}{\Delta_{\min}} \right).$$

**Conclusion**  We conclude the proof of Theorem 2 by combining Equations (22), (28) and (32).

## C.4  Proof of Theorem 3

Consider the actions $\mathcal{A}$ defined in the following lemma.

**Lemma 17.** *Let the action set be given by* $\mathcal{A} = \left\{ \begin{pmatrix} x_1 \\ z_{x_1} \end{pmatrix}, ..., \begin{pmatrix} x_{d+1} \\ z_{x_{d+1}} \end{pmatrix} \right\}$, *where* $\begin{pmatrix} x_1 \\ z_{x_1} \end{pmatrix} = e_1 + e_{d+1}$, $\begin{pmatrix} x_i \\ z_{x_i} \end{pmatrix} = e_i - e_{d+1}$ *for* $i \in \{2, ..., d\}$, *and* $\begin{pmatrix} x_{d+1} \\ z_{x_{d+1}} \end{pmatrix} = -\left( 1 - \frac{2}{\sqrt{\kappa_*}+1} \right) e_1 - e_{d+1}$. *It holds that*

$$\min_{\pi \in \mathcal{P}^{\mathcal{A}}} \left\{ e_{d+1}^\top \left( \sum_{\left( \begin{smallmatrix} x \\ z \end{smallmatrix} \right) \in \mathcal{A}} \pi_x \begin{pmatrix} x \\ z_x \end{pmatrix} \begin{pmatrix} x \\ z_x \end{pmatrix}^\top \right)^+ e_{d+1} \right\} = \kappa.$$

By Lemma 17, $\mathcal{A} \in \mathbf{A}_{\kappa_*,d}$. We will introduce two bandit problems characterized by two parameters $\theta_T^{(1)}$ and $\theta_T^{(2)}$ - assuming that the noise $\xi_t$ is Gaussian and i.i.d. - and we prove that for any algorithm, the regret for one of those two problems must be of larger order than $\kappa_*^{1/3} T^{2/3}$.

We also consider the following two alternative problems. For a small $1/4 > \rho_T > 0$ where $\rho_T = T^{-1/3}\kappa_*^{1/3}$ (satisfied since $T > 4^3\kappa_*$), the two alternative action parameters are defined as:

$$\gamma_T^{(1)} = \frac{1+\rho_T}{2}e_1 + \frac{1-\rho_T}{2}e_2 - \frac{\rho_T}{2}\left( \sum_{3 \leq j \leq d} e_j \right)$$

$$\gamma_T^{(2)} = \frac{1-\rho_T}{2}e_1 + \frac{1+\rho_T}{2}e_2 + \frac{\rho_T}{2}\left( \sum_{3 \leq j \leq d} e_j \right).$$

On top of this, two bias parameters are defined as $\omega_T^{(1)} = -\frac{\rho_T}{2}$ and $\omega_T^{(2)} = \frac{\rho_T}{2}$. Through this, we define the two bandit problems of the sketch of proof of Lemma 17 characterized by $\theta_T^{(1)} = \begin{pmatrix} \gamma_T^{(1)} \\ \omega_T^{(1)} \end{pmatrix}$ and $\theta_T^{(2)} = \begin{pmatrix} \gamma_T^{(2)} \\ \omega_T^{(2)} \end{pmatrix}$ - and where the distribution of the noise $\xi_t$ is supposed to be Gaussian and i.i.d. We refer to these two problems respectively as **Problem 1** and **Problem 2**. We write $R_T^{(1)}$, $\mathbb{P}^{(1)}$ and

$\mathbb{E}^{(1)}$ (respectively $R_T^{(2)}$, $\mathbb{P}^{(2)}$ and $\mathbb{E}^{(2)}$) for the regret, probability and expectation for the first bandit problem, when the parameter is $\theta_T^{(1)}$ (respectively the second bandit problem with $\theta_T^{(2)}$). We also write $\mathbb{P}_j^{(i)}$ for the distribution of a sample received in **Problem i** when sampling action $x_j$ at any given time $t$ - note that by definition of the bandit problems, this distribution does not depend on $t$ and on the past samples given that action $x_j$ is sampled.

The three following facts hold on these two bandit problems:

**Fact 1** The parameters $\gamma_T^{(1)}$ and $\gamma_T^{(2)}$ are chosen so that $x_1$ is the unique best action for **Problem 1**, and $x_2$ is the unique best action for **Problem 2**. Choosing any sub-optimal action induces an instantaneous regret of at least $\rho_T$, and choosing the very sub-optimal action $x_{d+1}$ induces an instantaneous regret of at least $1/2$.

**Fact 2** Because of the chosen bias parameters, the distributions of the evaluations of all actions but $x_{d+1}$ are exactly the same under the two bandit problems characterized by $\theta^{(1)}$ and $\theta_T^{(2)}$ - i.e. exactly the same data is observed under the two alternative bandit problems defined by the two alternative parameters for all actions but $x_{d+1}$. More precisely, for $i \in \{1, 2\}$, in **Problem i** and at any time $t$, when sampling action $x_i$ where $i \leq 2$, we observe a sample distributed according to $\mathcal{N}(1/2, 1)$ - i.e. $\mathbb{P}_j^{(i)}$ is $\mathcal{N}(1/2, 1)$ - and when sampling action $x_i$ where $2 < i \geq d+1$, we observe a sample distributed according to $\mathcal{N}(0, 1)$ - i.e. $\mathbb{P}_j^{(i)}$ is $\mathcal{N}(0, 1)$.

**Fact 3** The distributions of the outcomes of the evaluation of action $x_{d+1}$ differs in the two bandit problems. Set $\alpha = 2/(\sqrt{\kappa_*} + 1)$. In **Problem 1**, $\mathbb{P}_{d+1}^{(1)}$ is $\mathcal{N}(-\frac{1-\alpha-\rho_T\alpha}{2}, 1)$. In **Problem 2**, $\mathbb{P}_{d+1}^{(2)}$ is $\mathcal{N}(-\frac{1-\alpha+\rho_T\alpha}{2}, 1)$. So that the difference between the means of the evaluations of action $x_{d+1}$ in the two bandit problems is $\bar{\Delta} = \rho_T\alpha = \frac{2\rho_T}{\sqrt{\kappa_*}+1} \leq \frac{2\rho_T}{\sqrt{\kappa_*}}$.

For $i \leq d+1$, we write $N_i(T)$ for the number of times that action $x_i$ has been selected before time $T$. In **Problem 1**, choosing the action $x_{d+1}$ leads to an instantaneous regret larger than $\frac{1}{2}$ (**Fact 1**), so that

$$R_T^{(1)} \geq \frac{\mathbb{E}^{(1)}\left[N_{x_{d+1}}(T)\right]}{2}.$$

If $\mathbb{E}^{(1)}\left[N_{d+1}(T)\right] \geq \frac{T^{2/3}\kappa_*^{1/3}}{2}$, then Theorem 1 follows immediately; we therefore consider from now on the case when

$$\mathbb{E}^{(1)}\left[N_{d+1}(T)\right] \leq \frac{T^{2/3}\kappa_*^{1/3}}{2}. \tag{33}$$

Now, let us define the event

$$F = \left\{ N_1(T) \geq \frac{T}{2}\kappa_*^{1/3} \right\}.$$

Note that action $x_1$ is optimal for **Problem 1** and that action $x_2$ is optimal for **Problem 2** (**Fact 1**). Since choosing an action that is sub-optimal leads to an instantaneous regret larger than $\rho_T$ (**Fact 1**), we also have

$$R_T^{(1)} \geq \frac{T\rho_T}{2}\mathbb{P}^{(1)}\left(\overline{F}\right)$$

and

$$R_T^{(2)} \geq \frac{T\rho_T}{2}\mathbb{P}^{(2)}\left(F\right).$$

Then, Bretagnolle-Huber inequality (see, e.g., Theorem 14.2 in [24]) implies that

$$R_T^{(1)} + R_T^{(2)} \geq \frac{T\rho_T}{4}\exp\left(-KL\left(\mathbb{P}^{(1)}, \mathbb{P}^{(2)}\right)\right).$$

For the choice $\rho_T = T^{-1/3}\kappa_*^{1/3}$, this implies that

$$R_T^{(1)} + R_T^{(2)} \geq \frac{T^{2/3}\kappa_*^{1/3}}{4}\exp\left(-KL\left(\mathbb{P}^{(1)}, \mathbb{P}^{(2)}\right)\right). \tag{34}$$

Now, the Kullback-Leibler divergence between $\mathbb{P}^{(1)}$ and $\mathbb{P}^{(2)}$ can be rewritten as follows (see, e.g., Lemma 15.1 in [24]) :

$$KL(\mathbb{P}^{(1)}, \mathbb{P}^{(2)}) \quad = \quad \frac{1}{2} \sum_{j \leq d+1} \mathbb{E}^{(1)} \left[ N_j(T) \right] KL(\mathbb{P}_j^{(1)}, \mathbb{P}_j^{(2)}).$$

By **Fact 2**, we have that for any $j \leq d$, $\mathbb{P}_j^{(1)} = \mathbb{P}_j^{(2)}$. So that

$$KL(\mathbb{P}^{(1)}, \mathbb{P}^{(2)}) \quad = \quad \frac{1}{2} \mathbb{E}^{(1)} \left[ N_{d+1}(T) \right] KL(\mathbb{P}_{d+1}^{(1)}, \mathbb{P}_{d+1}^{(2)}).$$

By the characterization of $\mathbb{P}_{d+1}^{(1)}, \mathbb{P}_{d+1}^{(2)}$ in **Fact 3**, and recalling that the Kullback-Leibler divergence between two normalized Gaussian distributions is given by the squared distance between their means, we find that

$$KL(\mathbb{P}^{(1)}, \mathbb{P}^{(2)}) \quad = \quad \frac{1}{2} \mathbb{E}^{(1)} \left[ N_{d+1}(T) \right] \bar{\Delta}^2.$$

Thus, by the definition of $\bar{\Delta}$ in **Fact 3** and by Equation (33)

$$KL\left( \mathbb{P}^{(1)}, \mathbb{P}^{(2)} \right) = \frac{1}{2} \mathbb{E}^{(1)} \left[ N_{d+1}(T) \right] \left( \frac{2\rho_T}{\sqrt{\kappa_* + 1}} \right)^2 \leq \frac{T^{2/3} \kappa_*^{1/3}}{4} \times \frac{4\rho_T^2}{\kappa_*} = 1, \qquad (35)$$

reminding that $\rho_T = T^{-1/3} \kappa_*^{1/3}$.

Combining Equations (34) and (35) implies that

$$\max \left\{ R_T^{(1)}, R_T^{(2)} \right\} \geq \frac{T^{2/3} \kappa_*^{1/3}}{8} \exp(-1),$$

which concludes the proof of Theorem 3.

### C.5   Proof of Theorems 4

Theorems 4 follows directly from the next Theorem.

**Theorem 6.** *For all $\kappa_* \geq 1$ and all $d \geq 4$, there exists an action set $\mathcal{A} \in \mathbf{A}_{\kappa_*, d}$, such that for all bandit algorithms, for all $(\Delta_{\min}, \Delta_{\neq}) \in (0, 1/8)^2$ with $\Delta_{\min} \leq \Delta_{\neq}$, and for all budget $T \geq 2$, there exists a problem characterized by $\theta \in \Theta_{\Delta_{\min}, \Delta_{\neq}}^{\mathcal{A}}$ such that the regret of the algorithm on the problem satisfies*

$$R_T^\theta \quad \geq \quad \left[ \frac{d}{10\Delta_{\min}} \log\left(T\right) \left[ 1 - \frac{\log\left( \frac{8d \log(T)}{\Delta_{\min}^2} \right)}{\log\left(T\right)} \right] \right] \vee \left[ \frac{\kappa_* + 1}{4\Delta_{\neq}^2} \log\left(T\right) \left[ 1 - \frac{\log\left( \frac{8\kappa_* \log(T)}{\Delta_{\neq}^3} \right)}{\log\left(T\right)} \right] \right]$$

$$\vee \left[ \frac{\kappa_*}{4\Delta_{\neq}^2} \left[ 1 \wedge \log\left( \frac{T\Delta_{\neq}^3}{8\kappa_*} \right) \right] \right]. \qquad (36)$$

*Moreover, on this problem, $\kappa(\Delta) \in [\kappa_*/8, 2\kappa_*]$.*

**Remark 1.** *Note that Theorem 6 allows us to recover a lower bound similar to that of Theorem 3 by choosing $\Delta_{\neq}$ and $\Delta_{\min}$ of the order $\kappa_*^{1/3} T^{-1/3}$, however this bound only holds for $d$ larger than 4.*

We prove Theorem 6 for the following set of actions $\mathcal{A}$: $\mathcal{A} = \left\{ \begin{pmatrix} x_1 \\ z_{x_1} \end{pmatrix}, ..., \begin{pmatrix} x_{d+1} \\ z_{x_{d+1}} \end{pmatrix} \right\}$, where $\begin{pmatrix} x_i \\ z_{x_i} \end{pmatrix} = e_i + e_{d+1}$, for $i \in \{2, ..., \lfloor d/2 \rfloor\}$, $\begin{pmatrix} x_i \\ z_{x_i} \end{pmatrix} = e_i - e_{d+1}$ for $i \in \{\lfloor d/2 \rfloor + 1, ..., d\}$, and $\begin{pmatrix} x_{d+1} \\ z_{x_{d+1}} \end{pmatrix} = -\left( 1 - \frac{2}{\sqrt{\kappa_* + 1}} \right) e_1 - e_{d+1}$. Then, by Lemma 10, for this choice of action set, we have $\mathcal{A} \in \mathbf{A}_{\kappa_*, d}$.

We consider the following set of bandit problems: for $i \in \{1, ..., \lfloor d/2 \rfloor + 1\}$ **Problem i** is characterized by the parameter $\theta^{(i)}$, where $\theta^{(i)} = \begin{pmatrix} \gamma^{(i)} \\ \omega^{(i)} \end{pmatrix}$ is defined as:

$$\gamma^{(1)} = \frac{1 + \Delta_{\neq} - \Delta_{\min}}{2} \left( \sum_{1 \leq j \leq \lfloor d/2 \rfloor} e_j \right) + \frac{1 - \Delta_{\neq} - \Delta_{\min}}{2} \left( \sum_{\lfloor d/2 \rfloor + 1 \leq j \leq d} e_j \right) + \Delta_{\min} e_1 + \Delta_{\min} e_{\lfloor d/2 \rfloor + 1}$$

$$\gamma^{(i)} = \gamma^{(1)} + 2\Delta_{\min} e_i + 2\Delta_{\min} e_{\lfloor d/2 \rfloor + i} \quad \forall i \in \{2, ..., \lfloor d/2 \rfloor\}$$

$$\gamma^{(\lfloor d/2 \rfloor + 1)} = \frac{1 - \Delta_{\neq} - \Delta_{\min}}{2} \left( \sum_{1 \leq j \leq \lfloor d/2 \rfloor} e_j \right) + \frac{1 + \Delta_{\neq} - \Delta_{\min}}{2} \left( \sum_{\lfloor d/2 \rfloor + 1 \leq j \leq d} e_j \right) + \Delta_{\min} e_1 + \Delta_{\min} e_{\lfloor d/2 \rfloor + 1},$$

and the bias parameters are defined as $\omega^{(i)} = -\frac{\Delta_{\neq}}{2} \ \forall i \in \{1, ..., \lfloor d/2 \rfloor\}$, and otherwise $\omega^{(\lfloor d/2 \rfloor + 1)} = \frac{\Delta_{\neq}}{2}$. We write $\mathbb{E}^{(i)}, \mathbb{P}^{(i)}, R_T^{(i)}$ for resp. the probability, expectation, and regret, in **Problem i**. Note that this choice of parameters ensures that $\forall i \in \{1, ..., \lfloor d/2 \rfloor + 1\}, \theta^{(i)} \in \Theta_{\Delta_{\min}, \Delta_{\neq}}^{\mathcal{A}}$.

Set $\mathcal{A} = \left\{ \begin{pmatrix} x_1 \\ z_{x_1} \end{pmatrix}, ..., \begin{pmatrix} x_{d+1} \\ z_{x_{d+1}} \end{pmatrix} \right\}$, where $\begin{pmatrix} x_i \\ z_{x_i} \end{pmatrix} = e_i + e_{d+1}$, for $i \in \{2, ..., \lfloor d/2 \rfloor\}$, $\begin{pmatrix} x_i \\ z_{x_i} \end{pmatrix} = e_i - e_{d+1}$ for $i \in \{\lfloor d/2 \rfloor + 1, ..., d\}$, and $\begin{pmatrix} x_{d+1} \\ z_{x_{d+1}} \end{pmatrix} = -\left( 1 - \frac{2}{\sqrt{\kappa_*} + 1} \right) e_1 - e_{d+1}$. Then, Lemma 10 shows that $\mathcal{A} \in \mathbf{A}_{\kappa_*, d}$.

The following facts hold:

**Fact 1** For any $i \in \{1, ..., \lfloor d/2 \rfloor + 1\}$, action $x_i$ is the unique optimal action in **Problem i**. Since $1/2 \geq \Delta_{\neq} \geq \Delta_{\min}$, sampling any other (sub-optimal) action leads to an instantaneous regret of at least $\Delta_{\min}$. Moreover, choosing an action in the group $-z_i$ leads to an instantaneous regret of at least $\Delta_{\neq}$.

**Fact 2** In **Problem i** for any $i \in \{1, ..., \lfloor d/2 \rfloor + 1\}$, action $d + 1$ is very sub-optimal and sampling it leads to an instantaneous regret higher than $(1 - 2/(\sqrt{\kappa_*} + 1))(1 - \Delta_{\neq} + \Delta_{\min}) + (1 + \Delta_{\neq} + \Delta_{\min})/2 \geq 1/2$, since $\kappa_* \geq 1$ and $1/2 \geq \Delta_{\neq} \geq \Delta_{\min}$.

**Fact 3** In **Problem i**, for $i \in \{1, ..., \lfloor d/2 \rfloor + 1\}$, when sampling action $x_j$ at time, $t$ the distribution of the observation does not depend on $t$ or on the past (except through the choice of $x_j$) and is $\mathbb{P}_j^{(i)}$. It is characterized as:

$$\forall i \in \{1, ..., \lfloor d/2 \rfloor + 1\}, \mathbb{P}_1^{(i)}, \mathbb{P}_{\lfloor d/2 \rfloor + 1}^{(i)} \text{ are } \mathcal{N}((1 + \Delta_{\min})/2, 1)$$

$$\forall i \in \{1, ..., \lfloor d/2 \rfloor + 1\}, \forall j \in \{2, ..., d\} \setminus \{\lfloor d/2 \rfloor + 1, i, \lfloor d/2 \rfloor + i\}, \mathbb{P}_j^{(i)} \text{ is } \mathcal{N}((1 - \Delta_{\min})/2, 1),$$

$$\forall i \in \{2, \lfloor d/2 \rfloor\}, \mathbb{P}_i^{(i)} \text{ is } \mathcal{N}((1 + 3\Delta_{\min})/2, 1) \quad \mathbb{P}_{\lfloor d/2 \rfloor + i}^{(i)} \text{ is } \mathcal{N}((1 + 3\Delta_{\min})/2, 1)$$

$$\forall i \in \{1, \lfloor d/2 \rfloor\}, \mathbb{P}_{d+1}^{(i)} \text{ is } \mathcal{N}(-(1 - \alpha)(1 + \Delta_{\neq} + \Delta_{\min})/2 + \Delta_{\neq}/2, 1),$$

$$\mathbb{P}_{d+1}^{(\lfloor d/2 \rfloor + 1)} \text{ is } \mathcal{N}(-(1 - \alpha)(1 - \Delta_{\neq} + \Delta_{\min})/2 - \Delta_{\neq}/2, 1) \text{ where } \alpha = 2/(\sqrt{\kappa_*} + 1).$$

So that:

**Fact 3.1** For any $i \in \{2, ..., \lfloor d/2 \rfloor\}$, between **Problem 1** and **Problem i**, the only actions that provide different evaluations when sampled are action $i$ and action $\lfloor d/2 \rfloor + i$, and the mean gaps in both cases is $2\Delta_{\min}$.

**Fact 3.2** Between **Problem 1** and **Problem** $\lfloor d/2 \rfloor + 1$, the only action that provide different evaluation when sampled is action $d + 1$, and the mean gap in this case is $\alpha\Delta_{\neq}$.

For $j \leq d + 1$, we write $N_j(T)$ for the total number of times action $x_j$ has been selected before time $T$. Then, for $j \in \{1, ..., \lfloor d/2 \rfloor\}$, let $E^{(j)} = \{N_i(T) \leq T/2\}$. Note that for $i \in \{1, ..., \lfloor d/2 \rfloor\}$, in **Problem i** the action $x_i$ is the optimal action. Therefore, for any efficient algorithm, for all $i \in \{1, ..., \lfloor d/2 \rfloor\}$ the event $E^{(i)}$ should have a low probability under $\mathbb{P}^{(i)}$. Indeed, for $i \in \{1, ..., \lfloor d/2 \rfloor\}$, the regret of the algorithm under **Problem i** can be lower-bounded as follows - see **Facts 1 and 2**:

$$R_T^{(i)} \geq \sum_{j \leq \lfloor d/2 \rfloor, \ j \neq i} \mathbb{E}^{(i)} [N_j(T)] \Delta_{\min} + \sum_{\lfloor d/2 \rfloor + 1 \leq j \leq d} \mathbb{E}^{(i)} [N_j(T)] \Delta_{\neq} + \frac{\mathbb{E}^{(i)} [N_{d+1}(T)]}{2} \quad (37)$$

Since $\sum_j \mathbb{E}^{(i)}[N_j(T)] = T$ and $\Delta_{\min} \leq \Delta_{\neq} \leq \frac{1}{2}$, this implies together with **Facts 1**:

$$R_T^{(i)} \geq \left(T - \mathbb{E}^{(i)}[N_i(T)]\right) \Delta_{\min}$$

Using the definition of $E^{(i)}$, we find that

$$R_T^{(i)} \geq \frac{T\Delta_{\min}}{2} \mathbb{P}^{(i)}\left(E^{(i)}\right). \tag{38}$$

In particular for **Problem 1**, for any $i \in \{1, ..., \lfloor d/2 \rfloor\}$,

$$R_T^{(1)} \geq \frac{T\Delta_{\min}}{2} \mathbb{P}^{(1)}\left(\overline{E^{(i)}}\right). \tag{39}$$

since $E^{(1)} \supset \overline{E^{(i)}}$.

Similarly, let us also define the event $F = \left\{\sum_{i \leq \lfloor d/2 \rfloor} N_i(T) \geq T/2\right\}$. Then, in **Problem 1**, the group 1 contains the optimal action, and so for any efficient algorithm, the event $F$ should have a low probability under $\mathbb{P}^{(1)}$. Indeed, Equation (37) also implies

$$R_T^{(1)} \geq \left(T - \mathbb{E}^{(1)}\left[\sum_{i \leq \lfloor d/2 \rfloor} N_i(T)\right]\right) \Delta_{\neq} \geq \frac{T\Delta_{\neq}}{2} \mathbb{P}^{(1)}\left(\overline{F}\right). \tag{40}$$

On the other hand, for any efficient algorithm, the event $F$ should have high probability under $\mathbb{P}^{(\lfloor d/2 \rfloor + 1)}$. Indeed, under problem **Problem $\lfloor d/2 \rfloor + 1$**, the regret can be lower-bounded as follows - see **Facts 1 and 2**:

$$R_T^{(\lfloor d/2 \rfloor + 1)} \geq \sum_{j \leq \lfloor d/2 \rfloor} \mathbb{E}^{(\lfloor d/2 \rfloor + 1)}[N_j(T)] \Delta_{\neq} + \sum_{\lfloor d/2 \rfloor + 2 \leq j \leq d} \mathbb{E}^{(\lfloor d/2 \rfloor + 1)}[N_j(T)] \Delta_{\min} + \frac{\mathbb{E}^{(\lfloor d/2 \rfloor + 1)}[N_{d+1}(T)]}{2}.$$

which implies that

$$R_T^{(\lfloor d/2 \rfloor + 1)} \geq \sum_{j \leq \lfloor d/2 \rfloor} \mathbb{E}^{(\lfloor d/2 \rfloor + 1)}[N_j(T)] \Delta_{\neq} \geq \frac{T\Delta_{\neq}}{2} \mathbb{P}^{(\lfloor d/2 \rfloor + 1)}(F). \tag{41}$$

Now, Bretagnolle-Huber inequality (see, e.g., Theorem 14.2 in [24]) implies that for all $i \in \{2, ..., \lfloor d/2 \rfloor\}$,

$$\frac{1}{2} \exp\left(-KL\left(\mathbb{P}^{(1)}, \mathbb{P}^{(i)}\right)\right) \leq \mathbb{P}^{(i)}\left(E^{(i)}\right) + \mathbb{P}^{(1)}\left(\overline{E^{(i)}}\right) \tag{42}$$

and that

$$\frac{1}{2} \exp\left(-KL\left(\mathbb{P}^{(1)}, \mathbb{P}^{(\lfloor d/2 \rfloor + 1)}\right)\right) \leq \mathbb{P}^{(\lfloor d/2 \rfloor + 1)}(F) + \mathbb{P}^{(1)}\left(\overline{F}\right). \tag{43}$$

On the one hand, Equation (42) implies that for any $i \in \{2, ..., \lfloor d/2 \rfloor\}$,

$$KL\left(\mathbb{P}^{(1)}, \mathbb{P}^{(i)}\right) \geq -\log\left(2\mathbb{P}^{(i)}\left(E^{(i)}\right) + 2\mathbb{P}^{(1)}\left(\overline{E^{(i)}}\right)\right)$$

$$\geq \log(T) - \log\left(2T\mathbb{P}^{(i)}\left(E^{(i)}\right) + 2T\mathbb{P}^{(1)}\left(\overline{E^{(i)}}\right)\right). \tag{44}$$

Combining Equations (38), (39), and (44), we find that

$$KL\left(\mathbb{P}^{(1)}, \mathbb{P}^{(i)}\right) \geq \log(T) - \log\left(\frac{4(R_T^{(i)} + R_T^{(1)})}{\Delta_{\min}}\right). \tag{45}$$

On the other hand, Equation (43) implies that

$$KL\left(\mathbb{P}^{(1)}, \mathbb{P}^{(\lfloor d/2 \rfloor + 1)}\right) \geq -\log\left(2\mathbb{P}^{(\lfloor d/2 \rfloor + 1)}(F) + 2\mathbb{P}^{(1)}\left(\overline{F}\right)\right)$$

$$\geq \log(T) - \log\left(2T\mathbb{P}^{(\lfloor d/2 \rfloor + 1)}(F) + 2T\mathbb{P}^{(1)}\left(\overline{F}\right)\right). \tag{46}$$

Combining Equations (38), (39), and (46), we find that

$$KL\left(\mathbb{P}^{(1)}, \mathbb{P}^{(\lfloor d/2\rfloor+1)}\right) \geq \log(T) - \log\left(\frac{4(R_T^{(\lfloor d/2\rfloor+1)} + R_T^{(1)})}{\Delta_{\neq}}\right). \tag{47}$$

Also, note that for all $i \in \{2, ..., \lfloor d/2\rfloor + 1\}$, the Kullback-Leibler divergence between $\mathbb{P}^{(1)}$ and $\mathbb{P}^{(i)}$ can be decomposed as follows (see, e.g., Lemma 15.1 in [24]) :

$$KL(\mathbb{P}^{(1)}, \mathbb{P}^{(i)}) = \sum_{j \leq d+1} \mathbb{E}^{(1)}\left[N_j(T)\right] KL(\mathbb{P}_j^{(1)}, \mathbb{P}_j^{(i)}). \tag{48}$$

**Lower bound in** $d\Delta_{\min}^{-1}\log T$. By design, for $i \in \{2, ..., \lfloor d/2\rfloor\}$, all actions but $x_i$ and $x_{\lfloor d\rfloor + i}$ have the same distribution under $\mathbb{P}^{(1)}$ and $\mathbb{P}^{(i)}$ - see **Fact 3.1**. Then, Equation (48) becomes from **Fact 3.1** and from the expression of KL divergence between standard Gaussian distributions:

$$KL(\mathbb{P}^{(1)}, \mathbb{P}^{(i)}) = \frac{4\Delta_{\min}^2}{2}\mathbb{E}^{(1)}\left[N_i(T)\right] + \frac{4\Delta_{\min}^2}{2}\mathbb{E}^{(1)}\left[N_{\lfloor d\rfloor + i}(T)\right].$$

So that, summing over $i \in \{2, ..., \lfloor d/2\rfloor\}$, and by **Fact 1**:

$$\sum_{i \in \{2, ..., \lfloor d/2\rfloor\}} KL(\mathbb{P}^{(1)}, \mathbb{P}^{(i)}) \leq 2\Delta_{\min} R_T^{(1)}.$$

So that by Equation (45) (summing over $i \in \{2, ..., \lfloor d/2\rfloor\}$):

$$2\Delta_{\min} R_T^{(1)} \geq \sum_{i \in \{2, ..., \lfloor d/2\rfloor\}} \left[\log(T) - \log\left(\frac{4(R_T^{(i)} + R_T^{(1)})}{\Delta_{\min}}\right)\right]$$

$$= (\lfloor d/2\rfloor - 1)\log(T) - \sum_{i \in \{2, ..., \lfloor d/2\rfloor\}} \log\left(\frac{4(R_T^{(i)} + R_T^{(1)})}{\Delta_{\min}}\right).$$

Let us assume that our algorithm satisfies $\max_{i \leq \lfloor d/2\rfloor} R_T^{(i)} \leq \frac{d\log(T)}{\Delta_{\min}}$ - otherwise the bound immediately follows for this algorithm. Then

$$R_T^{(1)} \geq \frac{1}{2\Delta_{\min}}(\lfloor d/2\rfloor - 1)\log(T) - \frac{1}{2\Delta_{\min}}\sum_{i \in \{2, ..., \lfloor d/2\rfloor\}} \log\left(\frac{8d\log T}{\Delta_{\min}^2}\right)$$

$$\geq \frac{1}{2\Delta_{\min}}(\lfloor d/2\rfloor - 1)\left[\log(T) - \log\left(\frac{8d\log(T)}{\Delta_{\min}^2}\right)\right]. \tag{49}$$

Sine $d \geq 4$, we note that $\lfloor d/2\rfloor - 1 \geq d/5$. This concludes the proof for this part of the bound.

**Lower bound in** $\kappa_* \Delta_{\neq}^{-2}\log T$. By design, all actions but $x_{d+1}$ have the same evaluation under **Problem 1** and **Problem** $\lfloor d/2\rfloor + 1$ - see **Fact 3.2**. Then, by **Fact 3.2** and the expression between the KL divergence of standard Gaussians, Equation (48) becomes

$$KL(\mathbb{P}^{(1)}, \mathbb{P}^{(\lfloor d/2\rfloor+1)}) = \mathbb{E}^{(1)}\left[N_{d+1}(T)\right]\frac{(\alpha\Delta_{\neq})^2}{2} = \frac{1}{2}\mathbb{E}^{(1)}\left[N_{d+1}(T)\right]\left(\frac{2\Delta_{\neq}}{\sqrt{\kappa_*} + 1}\right)^2.$$

Combined with equation (47), this implies that

$$\frac{1}{2}\mathbb{E}^{(1)}\left[N_{d+1}(T)\right]\left(\frac{2\Delta_{\neq}}{\sqrt{\kappa_*} + 1}\right)^2 \geq \log(T) - \log\left(\frac{4(R_T^{(\lfloor d/2\rfloor+1)} + R_T^{(1)})}{\Delta_{\neq}}\right). \tag{50}$$

Let us assume that our algorithm satisfies $\max_{i \leq \lfloor d/2\rfloor + 1} R_T^{(i)} \leq \frac{\kappa_* \log(T)}{\Delta_{\neq}^2}$ - otherwise the bound immediately follows for this algorithm. We then have

$$\frac{1}{2}\mathbb{E}^{(1)}\left[N_{d+1}(T)\right]\left(\frac{2\Delta_{\neq}}{\sqrt{\kappa_*} + 1}\right)^2 \geq \log(T) - \log\left(\frac{8\kappa_* \log(T)}{\Delta_{\neq}^3}\right).$$

Using Equation (37), we find that

$$R_T^{(1)} \geq \frac{\kappa_* + 1}{4\Delta_{\neq}^2}\left[\log(T) - \log\left(\frac{8\kappa_* \log(T)}{\Delta_{\neq}^3}\right)\right]. \tag{51}$$

**Lower bound in $\kappa_* \Delta_{\neq}^{-2}$.** Let us assume that our algorithm satisfies $\max_{i \leq \lfloor d/2 \rfloor + 1} R_T^{(i)} \leq \frac{\kappa_*}{\Delta_{\neq}^2}$ - otherwise the bound immediately follows for this algorithm. Then, Equation (50) implies

$$\frac{1}{2}\mathbb{E}^{(1)}\left[N_{d+1}(T)\right]\left(\frac{2\Delta_{\neq}}{\sqrt{\kappa_*}}\right)^2 \geq \log(T) - \log\left(\frac{8\kappa_*}{\Delta_{\neq}^3}\right).$$

Using again Equation (37), we find that

$$R_T^{(1)} \geq \frac{\kappa_* + 1}{4\Delta_{\neq}^2}\log\left(\frac{T\Delta_{\neq}^3}{8\kappa_*}\right). \tag{52}$$

We conclude the proof of Theorem 6 by combining Equations (49), (51) and (52).

**Bounds on $\kappa(\Delta)$** Finally, Lemma 11 allows us to express $\kappa(\Delta)$ as a function of $\kappa_*$. On the one hand, since $\kappa_* \geq 1$, we see that $\kappa_* \leq (1 + \sqrt{\kappa_*})^2 \leq 4\kappa_*$. On the other hand, $1/2 \leq \Delta_{d+1} \leq 2$, so $\kappa(\Delta) \in \left[\frac{\kappa_*}{8}, 2\kappa_*\right]$.

### C.6 Extension of the gap-dependent lower bounds to $d = 2, 3$

Theorem 4 can be extended to $d \in \{2, 3\}$ by considering separately the cases $\frac{d}{\Delta_{\min}} \geq \frac{\kappa}{\Delta_{\neq}^2}$ and $\frac{d}{\Delta_{\min}} < \frac{\kappa}{\Delta_{\neq}^2}$.

**Case 1 :** $\frac{d}{\Delta_{\min}} \geq \frac{\kappa}{\Delta_{\neq}^2}$ Let us consider the set of actions defined by $\mathcal{A} = \left\{\begin{pmatrix} x_1 \\ z_{x_1} \end{pmatrix}, ..., \begin{pmatrix} x_{d+1} \\ z_{x_{d+1}} \end{pmatrix}\right\}$, where $\begin{pmatrix} x_i \\ z_{x_i} \end{pmatrix} = e_1 + e_{d+1}$ for $i \in \{1, ..., d\}$, and $\begin{pmatrix} x_{d+1} \\ z_{x_{d+1}} \end{pmatrix} = -\left(1 - \frac{2}{\sqrt{\kappa_*}+1}\right)e_1 - e_{d+1}$. Using the same proof as in Lemma 17, we see that

$$\min_{\pi \in \mathcal{P}^{\mathcal{A}}}\left\{e_{d+1}^{\top}\left(\sum_{\left(\begin{smallmatrix}x\\z\end{smallmatrix}\right)\in\mathcal{A}} \pi_x \begin{pmatrix} x \\ z_x \end{pmatrix}\begin{pmatrix} x \\ z_x \end{pmatrix}^{\top}\right)^{+} e_{d+1}\right\} = \kappa.$$

Then, we consider the following problems : for $i \leq d$, **Problem i** is characterized by the parameter $\theta^{(i)}$, where $\theta^{(i)} = \begin{pmatrix} \gamma^{(i)} \\ \omega^{(i)} \end{pmatrix}$ is defined as:

$$\gamma^{(1)} = \frac{1 - \Delta_{\min}}{2}\sum_{i \leq d}e_i + \Delta_{\min}e_1$$

$$\gamma^{(i)} = \frac{1 - \Delta_{\min}}{2}\sum_{i \leq d}e_i + \Delta_{\min}e_1 + \Delta_{\min}e_i \quad \text{for i} > 1$$

and the bias parameters are defined as $\omega^{(i)} = 0$ for $i \leq d$. The following facts hold:

**Fact 1** For any $i \in \{1, ..., d\}$, action $x_i$ is the unique optimal action in **Problem i**. Sampling any other (sub-optimal) action leads to an instantaneous regret of at least $\Delta_{\min}$.

**Fact 2** In **Problem i**, for $i \in \{1, ..., d\}$, when sampling action $x_j$ at time, $t$ the distribution of the observation does not depend on $t$ or on the past (except through the choice of $x_j$) and is $\mathbb{P}_j^{(i)}$. It is characterized as:

$$\forall i \in \{1, ..., d\}, \mathbb{P}_1^{(i)} \text{ is } \mathcal{N}((1 + \Delta_{\min})/2, 1)$$

$$\forall i \in \{1, ..., d\}, \mathbb{P}_{d+1}^{(1)} \text{ is } \mathcal{N}(-(1 - \frac{2}{\sqrt{\kappa_*}+1})(1 + \Delta_{\min})/2, 1)$$

$$\forall i \in \{2, ..., d\}, \mathbb{P}_i^{(i)} \text{ is } \mathcal{N}((1 + 3\Delta_{\min})/2, 1)$$

$$\forall i, j \in \{2, ..., d\}, i \neq j : \mathbb{P}_j^{(i)} \text{ is } \mathcal{N}((1 - \Delta_{\min})/2, 1)$$

So that for any $i \in \{2, ..., d\}$, between **Problem 1** and **Problem i**, the only action that provides different evaluations when sampled is action $i$, and the mean gap is $2\Delta_{\min}$.

Since $\Delta_{\neq} \leq \frac{1}{8}$, this choice of parameters ensures that $\forall i \in \{1, ..., d\}, \theta^{(i)} \in \Theta^{\mathcal{A}}_{\Delta_{\min}, \Delta_{\neq}, \kappa_*}$. Adapting the proof of Lemma 17, we note that the minimal variance of bias estimation is at least $\kappa_*$. This proves that $\mathcal{A} \in \Theta^{\mathcal{A}}_{\Delta_{\min}, \Delta_{\neq}, \kappa_*}$. Now, the lower bound

$$R_T \geq \frac{d-1}{2\Delta_{\min}} \left[ \log(T) - \log\left( \frac{8d \log(T)}{\Delta_{\min}^2} \right) \right]$$

follows directly using arguments from the proof of Theorem 6.

**Case 2 :** $\frac{d}{\Delta_{\min}} > \frac{\kappa}{\Delta_{\neq}^2}$ Let the action set be given by $\mathcal{A} = \left\{ \begin{pmatrix} x_1 \\ z_{x_1} \end{pmatrix}, ..., \begin{pmatrix} x_{d+1} \\ z_{x_{d+1}} \end{pmatrix} \right\}$, where $\begin{pmatrix} x_1 \\ z_{x_1} \end{pmatrix} = e_1 + e_{d+1}$, $\begin{pmatrix} x_i \\ z_{x_i} \end{pmatrix} = e_i - e_{d+1}$ for $i \in \{2, ..., d\}$, and $\begin{pmatrix} x_{d+1} \\ z_{x_{d+1}} \end{pmatrix} = -\left( 1 - \frac{2}{\sqrt{\kappa_*}+1} \right) e_1 - e_{d+1}$. By Lemma 17, $\mathcal{A} \in \mathbf{A}_{\kappa_*, d}$. We consider two bandit problems characterized by two parameters $\theta^{(1)}$ and $\theta^{(2)}$, defined as:

$$\gamma^{(1)} = \frac{1+\Delta_{\neq}}{2} e_1 + \frac{1-\Delta_{\neq}}{2} e_2 - \frac{\Delta_{\neq}}{2} e_3$$
$$\gamma^{(2)} = \frac{1-\Delta_{\neq}}{2} e_1 + \frac{1+\Delta_{\neq}}{2} e_2 + \frac{\Delta_{\neq}}{2} e_3.$$

On top of this, two bias parameters are defined as $\omega^{(1)} = -\frac{\Delta_{\neq}}{2}$ and $\omega^{(2)} = \frac{\Delta_{\neq}}{2}$.

The following facts hold:

**Fact 1** For any $i \in \{1, 2\}$, action $x_i$ is the unique optimal action in **Problem i**. Since $1/2 \geq \Delta_{\neq}$, sampling any other (sub-optimal) action leads to an instantaneous regret of at least $\Delta_{\neq}$.

**Fact 2** In **Problem i**, for $i \in \{1, ..., d\}$, when sampling action $x_j$ at time, $t$ the distribution of the observation does not depend on $t$ or on the past (except through the choice of $x_j$) and is $\mathbb{P}_j^{(i)}$. It is characterized as:

$$\forall i \in \{1, 2\}, \forall j \in \{1, 2\}, \mathbb{P}_j^{(i)} \text{ is } \mathcal{N}(1/2, 1)$$
$$\forall i \in \{1, 2\}, \mathbb{P}_3^{(1)} \text{ is } \mathcal{N}(0, 1)$$
$$\mathbb{P}_{d+1}^{(1)} \text{ is } \mathcal{N}\left( \left( 1 - \frac{2}{\sqrt{\kappa_*}+1} \right) \left( \frac{1+\Delta_{\neq}}{2} \right) + \frac{\Delta_{\neq}}{2}, 1 \right)$$
$$\mathbb{P}_{d+1}^{(2)} \text{ is } \mathcal{N}\left( \left( 1 - \frac{2}{\sqrt{\kappa_*}+1} \right) \left( \frac{1-\Delta_{\neq}}{2} \right) - \frac{\Delta_{\neq}}{2}, 1 \right)$$

So that, between **Problem 1** and **Problem 2**, the only action that provides different evaluations when sampled is action 1, and the mean gaps in both cases is $\frac{2\Delta_{\neq}}{\sqrt{\kappa_*}+1}$.

Note that the minimum gap for these parameters is $\Delta_{\neq} \geq \Delta_{\min}$. Thus, this choice of parameters ensures that $\forall i \in \{1, ..., d\}, \theta^{(i)} \in \Theta^{\mathcal{A}}_{\Delta_{\min}, \Delta_{\neq}, \kappa_*}$. Adapting the proof of Lemma 17, we note that the minimal variance of bias estimation is at least $\kappa_*$. This proves that $\mathcal{A} \in \Theta^{\mathcal{A}}_{\Delta_{\min}, \Delta_{\neq}, \kappa_*}$. Then, the lower bound

$$R_T \geq \frac{\kappa_* + 1}{4\Delta_{\neq}^2} \left[ \log(T) - \log\left( \frac{8\kappa_* \log(T)}{\Delta_{\neq}^3} \right) \right].$$

follows directly using arguments from the proof of Theorem 6.

## C.7 Auxiliary Lemmas

### C.7.1 Proof of Lemma 1

Lemma 1 follows from the characterization of $\kappa_*$ given in Lemma 5. We begin by proving the first statement. Assume that $\kappa_* > 1$ (otherwise the first statement is void). Note that for all $u \in \mathbb{R}^d$, $\lim_{\lambda \to +\infty} (\max_{x \in \mathcal{X}} (x^\top(\lambda u) + z_x)^2)^{-1} = 0$, so the minimum over $u \in \mathbb{R}^d$ of $(\max_{x \in \mathcal{X}} (x^\top(\lambda u) + z_x)^2)^{-1}$ is attained for some vector $\tilde{u} \in \mathbb{R}^d$. Since $\kappa_* > 1$, $\tilde{u}$ is not null.

Moreover, $\max_{x \in \mathcal{X}} (1 + z_x x^\top \tilde{u})^2 < 1$, so $\max_{x \in \mathcal{X}} z_x x^\top \tilde{u} < 0$. Thus, for all $x \in \mathcal{X}$, $x^\top \tilde{u}$ and $z_x$ are of opposite sign, and $x^\top \tilde{u} \neq 0$. This implies that the hyperplane containing $0$ with normal vector $\tilde{u}$ contains no action, and separates the two groups. Moreover,

$$\kappa_*^{-1/2} = \max_{x \in \mathcal{X}} |z_x x^\top \tilde{u} + 1|.$$

We denote $x^{(1)} \in \operatorname{argmax}_{x \in \mathcal{X}} z_z x^\top \tilde{u}$, and $x^{(2)} \in \operatorname{argmin}_{x \in \mathcal{X}} z_z x^\top \tilde{u}$. Let us show that $(z_{x^{(1)}} x^{(1)^\top} \tilde{u} + 1) = -\left(1 + z_{x^{(2)}} x^{(2)^\top} \tilde{u}\right)$, i.e that $z_{x^{(1)}} x^{(1)^\top} \tilde{u} + z_{x^{(2)}} x^{(2)^\top} \tilde{u} = -2$. Indeed, note that

$$\kappa_*^{-1/2} = (z_{x^{(1)}} x^{(1)^\top} \tilde{u} + 1) \vee -(1 + z_{x^{(2)}} x^{(2)^\top} \tilde{u}).$$

Then, for $u' = \dfrac{-2}{\left(z_{x^{(1)}} x^{(1)} + z_{x^{(2)}} x^{(2)}\right)^\top \tilde{u}} \tilde{u}$, we see that

$$z_{x^{(1)}} x^{(1)^\top} u' + 1 = -\left(1 + z_{x^{(2)}} x^{(2)^\top} u'\right) = \max_{x \in \mathcal{X}} |z_x x^\top u' + 1|.$$

By contradiction, let us first assume that $z_{x^{(1)}} x^{(1)^\top} \tilde{u} + z_{x^{(2)}} x^{(2)^\top} \tilde{u} < -2$. Then,

$$\max_{x \in \mathcal{X}} |z_x x^\top u' + 1| = z_{x^{(1)}} x^{(1)^\top} u' + 1 < z_{x^{(1)}} x^{(1)^\top} \tilde{u} + 1 = \kappa_*^{-1/2}$$

which contradicts the definition of $\kappa_*$.

Similarly, if we assume that $z_{x^{(1)}} x^{(1)^\top} \tilde{u} + z_{x^{(2)}} x^{(2)^\top} \tilde{u} > -2$, then

$$\max_{x \in \mathcal{X}} |z_x x^\top u' + 1| = -(z_{x^{(2)}} x^{(2)^\top} u' + 1) < -(z_{x^{(2)}} x^{(2)^\top} \tilde{u} + 1) = \kappa_*^{-1/2}$$

which contradicts again the definition of $\kappa_*$. Therefore,

$$(z_{x^{(1)}} x^{(1)^\top} \tilde{u} + 1) = -\left(1 + z_{x^{(2)}} x^{(2)^\top} \tilde{u}\right) = \kappa_*^{-1/2}.$$

Then, the hyperplane containing $0$ with normal vector $\tilde{u}$ separates the actions of the two groups. Moreover, the margin is $-z_{x^{(1)}} x^{(1)^\top} \tilde{u} = 1 - \kappa_*^{-1/2}$, while the maximum distance of all points is $-z_{x^{(2)}} x^{(2)^\top} \tilde{u} = 1 + \kappa_*^{-1/2}$. Thus, there exists $\tilde{u}$ such that the hyperplane containing $0$ with normal vector $\tilde{u}$ separates the actions of the two groups, with margin equal to $\frac{\sqrt{\kappa_*}-1}{\sqrt{\kappa_*}+1}$ times the maximum distance of all points to the hyperplane.

Conversely, assume that there exists $\kappa > \kappa_*$ such that there exists $u \in \mathbb{R}^d$ such that the hyperplane containing $0$ with normal vector $u$ separates the actions of the two groups, with margin equal to $\frac{\sqrt{\kappa}-1}{\sqrt{\kappa}+1} = \frac{1-\kappa^{-1/2}}{1+\kappa^{-1/2}}$ times the maximum distance of all points to the hyperplane, denoted hereafter $d$. Since the hyperplane separates the points, we can assume without loss of generality that for all $x \in \mathcal{X}$, $z_x x^\top u < 0$. Similarly, up to a renormalization, we can assume without loss of generality that $d = 1 + \kappa^{-1/2}$. Then,

$$\begin{aligned}
\max_{x \in \mathcal{X}} |z_x x^\top u + 1| &= (\max_{x \in \mathcal{X}} z_x x^\top u + 1) \vee -(\min_{x \in \mathcal{X}} z_x x^\top u + 1) \\
&= \left(-\frac{1 - \kappa^{-1/2}}{1 + \kappa^{-1/2}} \times (1 + \kappa^{-1/2}) + 1\right) \vee -(1 - \kappa^{-1/2} - 1) = \kappa^{-1/2} < \kappa_*^{-1/2}
\end{aligned}$$

which contradicts the definition of $\kappa_*$. This concludes the proof of the first statement.

To prove the second statement, let us assume that no separating hyperplane containing zero exists. Then, for all $u \in \mathbb{R}^d$, there exists $x \in \mathcal{X}$ such that $z_x x^\top u \geq 0$. This implies that $\min_{u \in \mathbb{R}^d} \max_{x \in \mathcal{X}} (z_x x^\top u + 1) \geq 1$, so $\kappa_* \leq 1$. Choosing $u = 0$, we see that $\kappa_* \geq 1$, which implies that $\kappa_* = 1$.

### C.7.2 Proof of Lemma 2

Since for all $\gamma \in \mathcal{X}$ and all $x \in \mathcal{X}$, $|x^\top \gamma| \leq 1$, it is easy to see that the gaps are bounded by 2, and that $\widetilde{\kappa} \leq 2\kappa_*$.

Let us now show that $\widetilde{\kappa} \geq \kappa_*/2$.

$$\left(x^{(1)}, x^{(2)}, \widetilde{\gamma}\right) \in \operatorname*{argmax}_{(x,x')\in\mathcal{X}, \gamma\in\mathcal{C}(\mathcal{X})} (x - x')^\top \gamma$$

$$\overline{x} = \frac{1}{2}(x^{(1)} + x^{(2)})$$

$$\widetilde{n} = \sum_{x\in\mathcal{X}} \widetilde{\mu}(x)$$

$$\text{and} \quad \widetilde{x} = \frac{1}{\widetilde{n}} \sum_{x\in\mathcal{X}} \widetilde{\mu}(x)x.$$

Recall that $\kappa_*$ can equivalently be defined as the budget necessary to estimate the bias with a variance smaller than 1. Therefore, we have

$$\widetilde{n} \geq \kappa_*. \tag{53}$$

Let us define $\Delta_{\max}$ as $\Delta_{\max} = (x^{(1)} - x^{(2)})^\top \widetilde{\gamma} = \max_{(x,x')\in\mathcal{X}, \gamma\in\mathcal{C}(\mathcal{X})} (x - x')^\top \gamma$. By definition of $\widetilde{\kappa}$ and $\widetilde{\mu}$,

$$\widetilde{\kappa} \geq \sum_{x\in\mathcal{X}} \widetilde{\mu}(x)(x^{(1)} - x)^\top \widetilde{\gamma}$$

$$= \widetilde{n}(x^{(1)} - \widetilde{x})^\top \widetilde{\gamma}.$$

Using Equation (53), we find that

$$\frac{\widetilde{\kappa}}{\kappa_*} \geq (x^{(1)} - \overline{x})^\top \widetilde{\gamma} + (\overline{x} - \widetilde{x})^\top \widetilde{\gamma}$$

$$= \frac{\Delta_{\max}}{2} + (\overline{x} - \widetilde{x})^\top \widetilde{\gamma}. \tag{54}$$

Now, since $\widetilde{\gamma} \in \mathcal{C}(\mathcal{X})$, we also have $-\widetilde{\gamma} \in \mathcal{C}(\mathcal{X})$, and therefore

$$\widetilde{\kappa} \geq \sum_{x\in\mathcal{X}} \widetilde{\mu}(x)(x^{(2)} - x)^\top (-\widetilde{\gamma})$$

$$= \widetilde{n}(\widetilde{x} - x^{(2)})^\top \widetilde{\gamma}$$

Using again Equation (53), we find that

$$\frac{\widetilde{\kappa}}{\kappa_*} \geq (\widetilde{x} - \overline{x})^\top \widetilde{\gamma} + (\overline{x} - x^{(2)})^\top \widetilde{\gamma}$$

$$= (\widetilde{x} - \overline{x})^\top \widetilde{\gamma} + \frac{\Delta_{\max}}{2}. \tag{55}$$

Combining Equations (54) and (55), we find that

$$\frac{\widetilde{\kappa}}{\kappa_*} \geq \frac{\Delta_{\max}}{2} + |(\overline{x} - \widetilde{x})^\top \widetilde{\gamma}|.$$

This implies in particular that $\widetilde{\kappa} \geq \frac{\Delta_{\max}\kappa_*}{2}$.

To conclude the proof of the Lemma, we show that $\Delta_{\max} \geq 1$. By contradiction, assume that $\Delta_{\max} < 1$.

For all non-zero vector $u \in \mathbb{R}^d$, let us denote $x_u = \operatorname{argmax}_{x\in\mathcal{X}} |x^\top u|$. Since $\mathcal{X}$ spans $\mathbb{R}^d$, we necessarily have $|x_u^\top u| > 0$, so we can define the normalized vector $\tilde{u} = u/|x_u^\top u|$ such that $\tilde{u}$ belongs to the set $\mathcal{C}(\mathcal{X})$. Finally, denote $x_u^{(1)}, x_u^{(2)} \in \operatorname{argmax}_{x,x'\in\mathcal{X}}(x_u^{(1)} - x_u^{(2)})^\top \tilde{u}$. Note that by definition of $\Delta_{\max}$, we always have $(x_u^{(1)} - x_u^{(2)})^\top \tilde{u} \leq \Delta_{\max} < 1$.

Case 1 : $x_u^\top \tilde{u} > 0$ Then, by definition of $x_u$ and $x_u^{(1)}$, we see that $x_u^{(1)^\top} \tilde{u} = x_u^\top \tilde{u} = 1$. Then, $(x_u^{(1)} - x_u^{(2)})^\top \tilde{u} < 1$ implies that $1 - x_u^{(2)^\top} \tilde{u} < 1$, so $x_u^{(2)^\top} \tilde{u} > 0$, and in particular $x_u^{(2)^\top} u > 0$.

Case 2 : $x_u^\top u < 0$ Then, by definition of $x_u$ and $x_u^{(2)}$, we see that $x_u^{(2)^\top} \tilde{u} = x_u^\top u = -1$. Then $(x_u^{(1)} - x_u^{(2)})^\top \tilde{u} < 1$ implies that $x_u^{(1)^\top} \tilde{u} + 1 < 1$, so $x_u^{(1)^\top} \tilde{u} < 0$, and in particular $x_u^{(1)^\top} u < 0$.

Putting together Case 1 and Case 2, we see that $x_u^{(1)^\top} u$ and $x_u^{(2)^\top} u$ are of the same sign and are not null. By definition of $x_u^{(1)}$ and $x_u^{(2)}$, we conclude that for all $x \in \mathcal{X}$, the sign of $x^\top u$ is the same, and that $x^\top u$ is not 0. Since this is true for all non-zero vector $u$, this implies in particular that no hyperplane containing the origin can separate the actions, which contradicts the assumption that $\mathcal{X}$ spans $\mathbb{R}^d$.

### C.7.3 Proof of Lemmas 3 and 4

We begin by proving Lemma 4. Recall that $\pi$ is a G-optimal design for the set $\{a_x : x \in \mathcal{X}\}$, and that $\mu$ is defined as $\mu(x) = \lceil m\pi(x) \rceil$ for all $x \in \mathcal{X}$.

We first observe that $V(\pi) = A_\pi^\top A_\pi$, where $A_\pi$ is the matrix with lines given by $[\sqrt{\pi(x)} a_x^\top]_{x \in \mathcal{X}}$. Since the supports of $\mu$ and $\pi$ are the same, we get that $\mathrm{Range}(A_\pi^\top) = \mathrm{Range}(A_\mu^\top)$. As a consequence

$$\mathrm{Range}(V(\pi)) = \mathrm{Range}(A_\pi^\top) = \mathrm{Range}(A_\mu^\top) = \mathrm{Range}(V(\mu)),$$

and $x \in \mathrm{Range}(V(\mu))$ for all $x \in \mathcal{X}$. This ensures that $a_x^\top \widehat{\theta}_\mu$ is an unbiased estimator of $a_x^\top \theta^*$.

Furthermore $V(\mu) \succcurlyeq mV(\pi)$, so the variance $a_x^\top V(\mu)^+ a_x$ of $a_x^\top \widehat{\theta}_\mu$ is upper-bounded by $a_x^\top V(\mu)^+ a_x \le m^{-1} a_x^\top V(\pi)^+ a_x$. Now, the General Equivalence Theorem of Kiefer and Pukelshein shows that $\max_{x \in \mathcal{X}} a_x^\top V(\pi)^+ a_x \le d+1$. Thus, $a_x^\top V(\pi)^+ a_x \le m^{-1}(d+1)$.

We now prove Lemma 3. Recall that $\pi \in \mathcal{M}_{e_{d+1}}^\mathcal{X}$ is such that $e_{d+1} \in \mathrm{Range}\, V(\pi)$, and that $\mu$ is defined as $\mu(x) = \lceil m\pi(x) \rceil$ for all $x \in \mathcal{X}$. Using similar arguments, we can show that $e_{d+1} \in \mathrm{Range}(V(\mu))$, which ensures that $e_{d+1}^\top \widehat{\theta}_\mu$ is an unbiased estimator of $e_{d+1}^\top \theta^*$. The second part of the Lemma follows directly using that $V(\mu) \succcurlyeq mV(\pi)$.

### C.7.4 Proof of Lemma 5

Elfving's set $\mathcal{S}$ for estimating the bias in the biased linear bandit problem is given by

$$\mathcal{S} = convex\ hull \left\{ \begin{pmatrix} x \\ z_x \end{pmatrix}, \begin{pmatrix} -x \\ -z_x \end{pmatrix} : x \in \mathcal{X} \right\},$$

or equivalently by

$$\mathcal{S} = convex\ hull \left\{ \pm \begin{pmatrix} z_x x \\ 1 \end{pmatrix} : x \in \mathcal{X} \right\}.$$

Now, Theorem 5 indicates that $\kappa_*^{-1/2} e_{d+1}$ belongs to a supporting hyperplane of $\mathcal{S}$. We first show that when $\mathcal{A}$ spans $\mathbb{R}^{d+1}$, any normal vector $w \in \mathbb{R}^{d+1}$ to this hyperplane is such that $w^\top e_{d+1} \ne 0$.

By contradiction, let us assume that $\kappa_*^{-1/2} e_{d+1}$ belongs to some supporting hyperplane $\mathcal{H}$ of $\mathcal{S}$ parametrized as $\mathcal{H} = \{a \in \mathbb{R}^{d+1} : a^\top w = b\}$, where the normal vector $w$ is of the form $w = \binom{u}{0}$. Then, $\kappa_*^{-1/2} e_{d+1} \in \mathcal{H}$, so $\kappa_*^{-1/2} e_{d+1}^\top w = b$, and thus $b = 0$. Now, $\mathcal{H}$ is a supporting hyperplane of $\mathcal{S}$, so for all $a \in \mathcal{S}$ we see that $a^\top w \le b$. In particular, for all $x \in \mathcal{X}$, $x^\top u \le 0$ and $-x^\top u \le 0$, so $x^\top u = 0$. This implies that $\mathcal{X}$ is supported by an hyperplane in $\mathbb{R}^d$ with normal vector $u$, which contradicts our assumption that $\mathcal{A}$ spans $\mathbb{R}^{d+1}$. Thus, the supporting hyperplane of $\mathcal{S}$ containing $\kappa_*^{-1/2} e_{d+1}$ has a normal vector $w \in \mathbb{R}^{d+1}$ such that $w^\top e_{d+1} \ne 0$. In particular, we can parameterize this hyperplane as $\mathcal{H}_{u,b} = \{a \in \mathbb{R}^{d+1} : a^\top \binom{u}{1} = b\}$ for some $b \in \mathbb{R}$ and $u \in \mathbb{R}^d$.

Now, if $\mathcal{H}_{u,b}$ is a supporting hyperplane of $\mathcal{S}$, then, by definition, $\mathcal{S}$ is contained in the half space $\{a \in \mathbb{R}^{d+1} : a^\top \binom{u}{1} \le b\}$. In particular, for all $x \in \mathcal{X}$, one must have $z_x x^\top u + 1 \le b$ and

$-z_x x^\top u - 1 \le b$ : therefore, for all $x \in \mathcal{X}$, $|z_x x^\top u + 1| \le b$. Moreover, $\mathcal{H}_{u,b}$ is a supporting hyperplane of $\mathcal{S}$, so there exists an extreme point $a \in \mathcal{S}$ such that $a \in \mathcal{H}_{u,b}$. Note that $\mathcal{S}$ is the convex hull of $\left\{\pm \binom{z_x x}{1} : x \in \mathcal{X}\right\}$, so the extreme points of $\mathcal{S}$ are in $\left\{\pm \binom{z_x x}{1} : x \in \mathcal{X}\right\}$. In particular, this implies that $b = \max\left\{|z_x x^\top u + 1| : x \in \mathcal{X}\right\}$. Thus, the supporting hyperplane of $\mathcal{S}$ containing $\kappa_*^{-1/2} e_{d+1}$ is necessarily of the form $\mathcal{H}_{u, \max\{|z_x x^\top u + 1| : x \in \mathcal{X}\}}$.

On the one hand, $\kappa_*^{-1/2}$ belongs to the boundary of $\mathcal{S}$ and therefore to a supporting hyperplane $\mathcal{H}_{u, \max\{|z_x x^\top u + 1| : x \in \mathcal{X}\}}$ of $\mathcal{S}$. Then, there exists $u \in \mathbb{R}^d$ such that $\kappa_*^{-1/2} = \max\left\{|z_x x^\top u + 1| : x \in \mathcal{X}\right\}$.

On the other hand, it is easy to verify that for all $u \in \mathbb{R}^d$, $\mathcal{H}_{u, \max\{|z_x x^\top u + 1| : x \in \mathcal{X}\}}$ is a supporting hyperplane of $\mathcal{S}$. Now, $\kappa_*^{-1/2} e_{d+1}$ belongs to $\mathcal{S}$, so $\kappa_*^{-1/2} e_{d+1}^\top \binom{u}{1} \le \max\left\{|z_x x^\top u + 1| : x \in \mathcal{X}\right\}$. These two results imply that

$$\kappa_*^{-1/2} = \min_{u \in \mathbb{R}^d} \max_{x \in \mathcal{X}} |z_x x^\top u + 1|$$

which proves the Lemma.

### C.7.5 Proof of Lemma 6

We prove that $2(\sqrt{\kappa_*} - 1)^2 \vee 1 \le \alpha \le 8(\kappa_* + 1)$. Lemma 6 follows directly by noticing that $\alpha \ge 1$ and $\kappa_* \ge 1$.

Let us begin by proving that $2(\sqrt{\kappa_*} - 1)^2 \le \alpha$ for $\kappa_* > 1$ (otherwise this inequality is automatically verified). Note that for all $u \in \mathbb{R}^d$, $\lim_{\lambda \to +\infty} \frac{1}{\max_{x \in \mathcal{X}} (x^\top (\lambda u) + z_x)^2} = 0$, so the minimum over $u \in \mathbb{R}^d$ of $\frac{1}{\max_{x \in \mathcal{X}} (x^\top u + z_x)^2} = 0$ is attained for some vector $\tilde{u} \in \mathbb{R}^d$. Let us also denote $\tilde{x} \in \arg\max_{x \in \mathcal{X}} (z_x x^\top \tilde{u} + 1)^2$, such that

$$\kappa_* = \frac{1}{\left(z_{\tilde{x}} \tilde{x}^\top \tilde{u} + 1\right)^2}.$$

With these notations, we see that for all $x \in \mathcal{X}$,

$$(z_x x^\top \tilde{u} + 1)^2 \le (z_{\tilde{x}} \tilde{x}^\top \tilde{u} + 1)^2 = \kappa_*^{-1} < 1.$$

This implies that for all $x \in \mathcal{X}$,

$$z_x x^\top \tilde{u} \le -1 + \kappa_*^{-1/2} < 0.$$

Now, let us denote $x^{(1)}, x^{(2)} \in \arg\max_{x, x' \in \mathcal{X}} (x - x')^\top \tilde{u}$. By definition of $\alpha$, we see that

$$\alpha \ge \frac{\left((x^{(1)} - x^{(2)})^\top \tilde{u}\right)^2}{(z_{\tilde{x}} \tilde{x}^\top \tilde{u} + 1)^2} = \left((x^{(1)} - x^{(2)})^\top \tilde{u}\right)^2 \times \kappa_*.$$

Since $z_x x^\top \tilde{u} < 0$ for all $x \in \mathcal{X}$, and since no group is empty, we can conclude that there exists $x, x' \in \mathcal{X}$ such that $x^\top \tilde{u} > 0$ and $x'^\top \tilde{u} < 0$. In particular, by definition of $x^{(1)}$ and $x^{(2)}$, we see that $(x^{(1)})^\top \tilde{u} > 0$ and $(x^{(2)})^\top \tilde{u} < 0$. Then,

$$\left((x^{(1)} - x^{(2)})^\top \tilde{u}\right)^2 \ge \left((x^{(1)})^\top \tilde{u}\right)^2 + \left((x^{(2)})^\top \tilde{u}\right)^2 \ge 2(1 - \kappa_*^{-1/2})^2.$$

This implies that

$$\alpha \ge 2(1 - \kappa_*^{-1/2})^2 \times \kappa_* = 2(\sqrt{\kappa_*} - 1)^2.$$

Let us now prove that $\alpha \ge 1$. Note that by assumption, $\mathcal{X}$ spans $\mathbb{R}^d$, and in particular there exists $\tilde{u} \in \mathbb{R}^d$ and $x, x' \in \mathcal{X}$ such that $\max_{x \in \mathcal{X}} x^\top \tilde{u} > 0$ and $\min_{x \in \mathcal{X}} x^\top \tilde{u} \le 0$. Thus, $\max_{x, x' \in \mathcal{X}} ((x - x')^\top \tilde{u})^2 \ge \max_{x \in \mathcal{X}} (x^\top \tilde{u})^2$. For any $\lambda > 0$, choosing $u = \lambda \tilde{u}$ in the definition of $\alpha$ implies that

$$\alpha \ge \frac{\lambda^2 \max_{x \in \mathcal{X}} (x^\top u)^2}{\max_{x \in \mathcal{X}} (\lambda z_x x^\top u + 1)^2}.$$

Letting $\lambda$ go to infinity, we find that $\alpha \geq 1$.

Finally, we prove that $\alpha \leq 8(\kappa_* + 1)$. For all $u \in \mathbb{R}^d$, we see that

$$\frac{\max_{x,x' \in \mathcal{X}}((x - x')^\top u)^2}{\max_{x \in \mathcal{X}}(z_x x^\top u + 1)^2} \leq \frac{4\max_{x \in \mathcal{X}}(z_x x^\top u)^2}{\max_{x \in \mathcal{X}}(z_x x^\top u + 1)^2}.$$

Now, we see that

$$\frac{\max_{x \in \mathcal{X}}(z_x x^\top u)^2}{\max_{x \in \mathcal{X}}(z_x x^\top u + 1)^2} \leq \frac{2\max_{x \in \mathcal{X}}(z_x x^\top u + 1)^2 + 2}{\max_{x \in \mathcal{X}}(z_x x^\top u + 1)^2} \leq 2 + \frac{2}{\max_{x \in \mathcal{X}}(z_x x^\top u + 1)^2}.$$

This in turn implies that for all $u \in \mathbb{R}^d$,

$$\frac{\max_{x,x' \in \mathcal{X}}((x - x')^\top u)^2}{\max_{x \in \mathcal{X}}(z_x x^\top u + 1)^2} \leq 8(1 + \kappa_*),$$

which finally implies that $\alpha \leq 8(1 + \kappa_*)$.

### C.7.6 Proof of Lemma 8

**Proof of Claim i)** The proof of the first claim is immediate by definition of $\kappa$. Indeed, let $\widetilde{\mathcal{M}} = \left\{ \mu \in \mathcal{M}_{e_{d+1}}^{\mathcal{X}} : e_{d+1}^\top V(\mu)^+ e_{d+1} \leq 1 \right\}$ be the set of measures $\mu$ admissible for estimating $\omega^*$ with a precision level 1. Then,

$$\kappa(c\Delta) \quad = \quad \min_{\mu \in \widetilde{\mathcal{M}}} \sum_x \mu(x) c\Delta_x = c \min_{\mu \in \widetilde{\mathcal{M}}} \sum_x \mu(x)\Delta_x = c\kappa(\Delta).$$

**Proof of Claim ii)** The proof of the second claim is also straightforward. If $\Delta \leq \Delta'$, then for all $\mu \in \widetilde{\mathcal{M}}$, $\sum_x \mu(x)\Delta_x \leq \sum_x \mu(x)\Delta'_x$. Recall that $\mu^{\Delta'} = \operatorname{argmin}_{\mu \in \widetilde{\mathcal{M}}} \sum_x \mu(x)\Delta'_x$. Then,

$$\kappa(\Delta') \quad = \quad \sum_x \mu^{\Delta'}(x)\Delta'_x \geq \sum_x \mu^{\Delta'}(x)\Delta_x \geq \min_{\mu \in \widetilde{\mathcal{M}}} \sum_x \mu(x)\Delta_x = \kappa(\Delta).$$

**Proof of Claim iii)** To prove the third claim, note that

$$\kappa(\Delta \vee \Delta') \quad = \quad \min_{\mu \in \widetilde{\mathcal{M}}} \sum_x \mu(x)\left(\Delta_x \vee \Delta_x\right)$$

$$\geq \quad \min_{\mu \in \widetilde{\mathcal{M}}} \left( \sum_x \mu(x)\Delta_x \vee \sum_x \mu(x)\Delta'_x \right)$$

$$\geq \quad \left( \min_{\mu \in \widetilde{\mathcal{M}}} \sum_x \mu(x)\Delta_x \right) \vee \left( \min_{\mu \in \widetilde{\mathcal{M}}} \sum_x \mu(x)\Delta'_x \right)$$

$$\geq \quad \kappa(\Delta) \vee \kappa(\Delta').$$

**Proof of Claim iv)** Recall that

$$\kappa(\Delta) = \min_{\mu \in \widetilde{\mathcal{M}}} \sum_x \mu(x)\Delta_x.$$

Let us define a sequence $(\mu_n)_{n \in \mathbb{N}} \in \widetilde{\mathcal{M}}^{\mathbb{N}}$ such that $\sum_x \mu_n(x)\Delta_x \xrightarrow[n \to \infty]{} \kappa(\Delta)$, and let us denote $\kappa_n = \sum_x \mu_n(x)\Delta_x$. According to Claim ii), we have

$$\kappa(\Delta) \leq \kappa(\Delta \vee \epsilon) = \min_{\mu \in \widetilde{\mathcal{M}}} \sum_x \mu(x)\left(\Delta_x \vee \epsilon\right) \leq \sum_x \mu_n(x)\Delta_x + \epsilon \sum_x \mu_n(x).$$

It follows that for all $n$,

$$\kappa(\Delta) \leq \liminf_{\epsilon \to 0^+} \kappa(\Delta \vee \epsilon) \leq \limsup_{\epsilon \to 0^+} \kappa(\Delta \vee \epsilon) \leq \kappa_n.$$

Letting $n$ go to infinity, we get that $\lim_{\epsilon \to 0^+} \kappa(\Delta \vee \epsilon) = \kappa(\Delta)$.

### C.7.7 Proof of Lemma 9

Setting $\mu \cdot \Delta = (\mu(x)\Delta_x)_{x \in \mathcal{X}}$ and

$$V_\Delta(\lambda) = \sum_{x \in \mathcal{X}} \lambda_x \begin{pmatrix} \Delta_x^{-1/2} x \\ \Delta_x^{-1/2} z_x \end{pmatrix} \begin{pmatrix} \Delta_x^{-1/2} x \\ \Delta_x^{-1/2} z_x \end{pmatrix}^\top,$$

we observe that $V_\Delta(\mu \cdot \Delta) = V(\mu)$. Hence,

$$\kappa(\Delta) = \min_{\substack{\mu \in \mathcal{M}^+ \\ e_{d+1}^\top V_\Delta(\mu \cdot \Delta)^+ e_{d+1} \leq 1}} \sum_{x \in \mathcal{X}} (\mu \cdot \Delta)_x.$$

We observe that $e_{d+1} \in \mathrm{Range}(V(\mu))$ is equivalent to $e_{d+1} \in \mathrm{Range}(V_\Delta(\mu \cdot \Delta))$. Hence, $\mu^\Delta \cdot \Delta = \lambda^\Delta$ where

$$\lambda^\Delta \in \underset{\substack{\lambda \in \mathbb{R}_+^{\mathcal{X}} \\ e_{d+1} \in \mathrm{Range}(V_\Delta(\lambda)) \\ e_{d+1}^\top V_\Delta(\lambda)^+ e_{d+1} \leq 1}}{\mathrm{argmin}} \sum_{x \in \mathcal{X}} \lambda_x.$$

The conclusion then follows by noticing that by homogeneity, $\lambda^\Delta = \kappa^\Delta \pi^\Delta$.

### C.7.8 Proof of Lemma 12

Lemma 12 follows directly from Lemmas 18 and 19.

**Lemma 18.**

$$\mathbb{P}\left( \exists l \geq 1, z \in \{-1,1\} \text{ such that } Explore_l^{(z)} = True, \text{ and } x \in \mathcal{X}_l^{(z)} \text{ such that } \left| \begin{pmatrix} \widehat{\gamma}_l^{(z)} - \gamma^* \\ \widehat{\omega}_l^{(z)} - \omega^* \end{pmatrix}^\top \begin{pmatrix} x \\ z_x \end{pmatrix} \right| \geq \epsilon_l \right) \leq \delta.$$

**Lemma 19.**

$$\mathbb{P}\left( \exists l \geq 1 \text{ such that } Explore_l^{(0)} = True \text{ and } \left| \widehat{\omega}_l^{(0)} - \omega^* \right| \geq \epsilon_l \right) \leq \delta.$$

### C.7.9 Proof of Lemma 13

To prove Lemma 13, we rely on the following key lemma. This lemma proves that on $\overline{\mathcal{F}}$, i.e. when the error bounds hold, the algorithm never eliminates the best action or the best group.

**Lemma 20.** *On the event $\overline{\mathcal{F}}$, for all $x^* \in \mathrm{argmax}_{x \in \mathcal{X}} x^\top \gamma^*$ and all $l$ such that $Explore_l^{(z_{x^*})} = True$, $x^* \in \mathcal{X}_{l+1}^{(z_{x^*})}$. Moreover, on the event $\overline{\mathcal{F}}$, for all $l$ such that $Explore_l^{(0)} = True$, there exists $x^* \in \mathrm{argmax}_{x \in \mathcal{X}} x^\top \gamma^*$ such that $\widehat{z^*}_{l+1} \neq -z_{x^*}$.*

Let $l \geq 1$ be such that $Explore_l^{(z_{x^*})} = True$. Then, on $\overline{\mathcal{F}}$, $x^* \in \mathcal{X}_{l+1}^{(z_{x^*})}$ by Lemma 20. Moreover, for all $x \in \mathcal{X}_{l+1}^{(z_{x^*})}$, by definition of $\mathcal{X}_{l+1}^{(z_{x^*})}$, we have that on $\overline{\mathcal{F}}$

$$\left( \begin{pmatrix} x^* \\ z_{x^*} \end{pmatrix} - \begin{pmatrix} x \\ z_{x^*} \end{pmatrix} \right)^\top \begin{pmatrix} \widehat{\gamma}_l^{(z)} \\ \widehat{\omega}_l^{(z)} \end{pmatrix} \leq 3\epsilon_l.$$

which implies that

$$\left( \begin{pmatrix} x^* \\ z_{x^*} \end{pmatrix} - \begin{pmatrix} x \\ z_{x^*} \end{pmatrix} \right)^\top \begin{pmatrix} \gamma^* \\ \omega^* \end{pmatrix} \leq 3\epsilon_l + \left| \begin{pmatrix} x^* \\ z_{x^*} \end{pmatrix}^\top \begin{pmatrix} \widehat{\gamma}_l^{(z)} - \gamma^* \\ \widehat{\omega}_l^{(z)} - \omega^* \end{pmatrix} \right| + \left| \begin{pmatrix} x \\ z_{x^*} \end{pmatrix}^\top \begin{pmatrix} \widehat{\gamma}_l^{(z)} - \gamma^* \\ \widehat{\omega}_l^{(z)} - \omega^* \end{pmatrix} \right|.$$

Thus, on the event $\overline{\mathcal{F}}$, for all $x \in \mathcal{X}_{l+1}^{(z_{x^*})}$

$$(x^* - x)^\top \gamma^* < 5\epsilon_l,$$

which proves Equation (13). To prove the second claim of Lemma 13, assume that for all $x' \in \mathrm{argmax}_{x \in \mathcal{X}} x^\top \gamma^*$, $z_{x'} = z_{x^*}$ (when this does not hold, the second claim follows from Equation

([13](#)). Now, let $l \geq 1$ be such that $\mathrm{Explore}_l^{(-z_{x^*})} = \mathrm{True}$. By Lemma [20], on $\overline{\mathcal{F}}$, $x^* \in \mathcal{X}_l^{(z_{x^*})}$ and $\widehat{z}_l^* = 0$. Then, the algorithm is unable to determine the group containing the best set during the phase $\mathrm{Exp}_{l-1}^{(0)}$, so there must exist $x' \in \mathcal{X}_l^{(-z_{x^*})}$ such that

$$
\begin{pmatrix} x^* \\ z_{x^*} \end{pmatrix}^\top \begin{pmatrix} \widehat{\gamma}_{l-1}^{(z_{x^*})} \\ \widehat{\omega}_{l-1}^{(z_{x^*})} \end{pmatrix} \leq \begin{pmatrix} x' \\ -z_{x^*} \end{pmatrix}^\top \begin{pmatrix} \widehat{\gamma}_{l-1}^{(-z_{x^*})} \\ \widehat{\omega}_{l-1}^{(-z_{x^*})} \end{pmatrix} + 2z_{x^*}\widehat{\omega}_{l-1}^{(0)} + 4\epsilon_{l-1}.
$$

It follows that

$$
\begin{pmatrix} x^* - x' \\ 2z_{x^*} \end{pmatrix}^\top \begin{pmatrix} \gamma^* \\ \omega^* \end{pmatrix} \leq \begin{pmatrix} x^* \\ z_{x^*} \end{pmatrix}^\top \begin{pmatrix} \gamma^* - \widehat{\gamma}_{l-1}^{(z_{x^*})} \\ \omega^* - \widehat{\omega}_{l-1}^{(z_{x^*})} \end{pmatrix} + \begin{pmatrix} x' \\ -z_{x^*} \end{pmatrix}^\top \begin{pmatrix} \widehat{\gamma}_{l-1}^{(-z_{x^*})} - \gamma^* \\ \widehat{\omega}_{l-1}^{(-z_{x^*})} - \omega^* \end{pmatrix} + 2z_{x^*}\widehat{\omega}_{l-1}^{(0)} + 4\epsilon_{l-1}.
$$

On $\overline{\mathcal{F}}$, this implies that

$$
\begin{pmatrix} x^* - x' \\ 2z_{x^*} \end{pmatrix}^\top \begin{pmatrix} \gamma^* \\ \omega^* \end{pmatrix} < 2z_{x^*}\widehat{\omega}_{l-1}^{(0)} + 6\epsilon_{l-1}
$$

so

$$
(x^* - x')^\top \gamma^* \leq 2z_{x^*}\left(\widehat{\omega}_{l-1}^{(0)} - \omega^*\right) + 6\epsilon_{l-1} < 8\epsilon_{l-1} = 16\epsilon_l. \tag{56}
$$

Moreover, for all $x \in \mathcal{X}_{l+1}^{(-z_{x^*})}$ we have $(a_{x'} - a_x)^\top \theta_l^{(-z_{x^*})} \leq 3\epsilon_l$, so following the same lines as for the first claim, we get $(x' - x)^\top \gamma^* < 5\epsilon_l$. Combining this bound with [(56)](#), we get

$$
\max_{x \in \mathcal{X}_{l+1}^{(-z_{x^*})}} (x^* - x)^\top \gamma^* < 21\epsilon_l.
$$

This concludes the proof of Lemma [13].

### C.7.10   Proof of Lemma [14]

For $z \in \{-1, +1\}$ and $l > 0$,

$$
\sum_x \mu_l^{(z)}(x) \leq \sum_x \frac{2(d+1)\pi_l^{(z)}(x)}{\epsilon_l^2} \log\left(\frac{kl(l+1)}{\delta}\right) + |\mathrm{supp}(\pi_l^{(z)})|.
$$

Now, $\mathrm{supp}(\pi_l^{(z)}) \leq \frac{(d+1)(d+2)}{2}$ and $\sum_x \pi_l^{(z)}(x) = 1$, so

$$
\sum_x \mu_l^{(z)}(x) \leq \frac{2(d+1)}{\epsilon_l^2} \log\left(\frac{kl(l+1)}{\delta}\right) + \frac{(d+1)(d+2)}{2}
$$

which proves the first claim of Lemma [14].

To prove the second claim, we bound the regret for bias estimation at stage $l$ as follows. On $\overline{\mathcal{F}}$, we have $\Delta_x \leq \widehat{\Delta}_x^l$ for all $x \in \mathcal{X}$ and $l \geq 1$, so

$$
\sum_{x \in \mathcal{X}} \mu_l^{(0)}(x)\Delta_x \leq \sum_{x \in \mathcal{X}} \mu_l^{(0)}(x)\widehat{\Delta}_x^l.
$$

Recall that $\hat{\mu}_l$ is the $\widehat{\Delta}^l$-optimal design, and that for all $x \in \mathcal{X}$, $\mu_l^{(0)}(x) = \lceil \frac{2\hat{\mu}_l(x)}{\epsilon_l^2} \log\left(\frac{l(l+1)}{\delta}\right) \rceil$. Since $\widehat{\Delta}_x^l \leq 2$ for all $x \in \mathcal{X}$, we have

$$
\sum_{x \in \mathcal{X}} \mu_l^{(0)}(x)\widehat{\Delta}_x^l \leq \sum_{x \in \mathcal{X}} \frac{2\hat{\mu}_l(x)}{\epsilon_l^2} \log\left(\frac{l(l+1)}{\delta}\right)\widehat{\Delta}_x^l + 2|\mathrm{supp}(\mu_l^{(0)})|
$$

and $|\mathrm{supp}(\mu_l^{(0)})| \leq d+1$, so

$$
\sum_x \mu_l^{(0)}(x)\Delta_x \leq \frac{2}{\epsilon_l^2} \log\left(\frac{l(l+1)}{\delta}\right) \sum_{x \in \mathcal{X}} \hat{\mu}_l(x)\widehat{\Delta}_x^l + 2(d+1).
$$

By definition of $\hat{\mu}_l(x)$, we have that

$$
\sum_{x \in \mathcal{X}} \hat{\mu}_l(x)\widehat{\Delta}_x^l = \kappa(\widehat{\Delta}^l).
$$

It follows that, on $\overline{\mathcal{F}}$,

$$
\sum_x \mu_l^{(0)}(x)\Delta_x \leq \sum_x \mu_l^{(0)}(x)\widehat{\Delta}_x^l \leq \frac{2}{\epsilon_l^2} \log\left(\frac{l(l+1)}{\delta}\right) \kappa(\widehat{\Delta}^l) + 2(d+1).
$$

### C.7.11 Proof of Lemma 15

For the first claim, we rely on the next lemma.

**Lemma 21.** *Let us set $\ell_x = \max\left\{l \geq 1 : x \in \mathcal{X}_l^{(-1)} \cup \mathcal{X}_l^{(1)}\right\}$. On $\overline{\mathcal{F}}$, we have for any $l \geq 1$*

1. $\widehat{\Delta}_x^l \leq \Delta_x + 16\epsilon_l$ *for all $x \in \mathcal{X}_l^{(-1)} \cup \mathcal{X}_l^{(1)}$ (i.e. for all $x$ such that $l \leq \ell_x$);*

2. *if $\Delta_x \geq 21\epsilon_l$ then $\ell_x \leq l$;*

3. $\epsilon_{\ell_x} < \Delta_x$ *for all $x \in \mathcal{X}$.*

Lemma 15 relies on the following remarks : if $\Delta, \Delta'$ are such that $\Delta_x \leq \Delta_x'$ for all $x \in \mathcal{X}$, then by Lemma 8 (ii), $\kappa(\Delta) \leq \kappa(\Delta')$. Let us now prove that for all $l \geq 1$ and all $x \in \mathcal{X}$, $\widehat{\Delta}_x^l \leq 513(\Delta \vee \epsilon_l)$.

**Case $\epsilon_l \geq \Delta_x$.** On $\overline{\mathcal{F}}$, we have $l \leq \ell_x - 1$ according to the third claim of Lemma 21. So, on $\overline{\mathcal{F}}$,

$$\widehat{\Delta}_x^l \leq \Delta_x + 16\epsilon_l \leq 17(\Delta_x \vee \epsilon_l).$$

**Case $\epsilon_l < \Delta_x$.** Then, on $\overline{\mathcal{F}}$, we have $32\epsilon_{l+5} < \Delta_x$ and so $l + 5 \geq \ell_x$ according to the second claim of Lemma 21. Hence, on $\overline{\mathcal{F}}$, according to Lemma 21, we have

$$\widehat{\Delta}_x^l \leq \max_{k=0,\ldots,5} \widehat{\Delta}_x^{\ell_x - k} \leq \Delta_x + 16\epsilon_{\ell_x - 5}$$

$$\leq \Delta_x + 512\epsilon_{\ell_x} \leq 513\Delta_x.$$

Thus, for all $l \geq 1$ and all $x \in \mathcal{X}$,

$$\widehat{\Delta}_x^l \leq 513(\Delta \vee \epsilon_l).$$

Now, let $\widetilde{\mathcal{M}} = \left\{\mu \in \mathcal{M}_{e_{d+1}}^{\mathcal{X}} : e_{d+1}^\top V(\mu)^+ e_{d+1} \geq 1\right\}$ the measures $\mu$ admissible for estimating $\omega^*$ with a precision level 1. Note that for all $a, b, c > 0$,

$$(1 + ab^{-1})(c \vee b) = (c + cab^{-1}) \vee (a + b) \geq c \vee (a + b) \geq c \vee a. \tag{57}$$

Using Equation (57) with $a = \Delta_x$, $b = \tau$ and $c = \epsilon$, we see that

$$\kappa(\Delta \vee \epsilon) = \min_{\mu \in \widetilde{\mathcal{M}}} \sum_x \mu(x)(\Delta_x \vee \epsilon) \leq (1 + \epsilon/\tau) \min_{\mu \in \widetilde{\mathcal{M}}} \sum_x \mu(x)(\Delta_x \vee \tau) = (1 + \epsilon/\tau)\kappa(\Delta \vee \tau).$$

Using Lemma 8 together with $\widehat{\Delta}_x^l \leq 513(\Delta \vee \epsilon_l)$, we find that

$$\kappa(\widehat{\Delta}_x^l) \leq 513\kappa(\Delta \vee \epsilon_l) \leq 513(1 + \epsilon_l/\tau)\kappa(\Delta \vee \tau).$$

This proves the first claim of Lemma 15.

To prove the second claim, we use Lemma 8 and the fact that for all $x$, $\widehat{\Delta}_x^l \geq \epsilon_l$. Moreover, on $\overline{\mathcal{F}}$, $\widehat{\Delta}_x^l \geq \Delta_x$ for all $x \in \mathcal{X}$. Then, $\kappa(\widehat{\Delta}) \geq \kappa(\epsilon_l \vee \Delta)$ by Lemma 8 (iii).

### C.7.12 Proof of Lemmas 16

To prove Lemma 16, let us consider $l$ such that $\epsilon_l \leq \frac{\Delta_{\neq}}{8}$. According to Lemma 20, on $\overline{\mathcal{F}}$ we know that $\widehat{z^*}_l \neq -z_{x^*}$. When $\widehat{z^*}_l = z_{x^*}$, then we also have $\widehat{z^*}_{l+1} = z_{x^*}$ and the conclusion follows immediately. Let us consider now the case where $\widehat{z^*}_l = 0$. By definition of $\Delta_{\neq}$, for all $x' \in \mathcal{X}_{l+1}^{(-z_{x^*})}$,

$$(x^* - x')^\top \gamma^* \geq \Delta_{\neq}.$$

This implies that

$$\begin{pmatrix} x^* \\ z_{x^*} \end{pmatrix}^\top \begin{pmatrix} \widehat{\gamma}_l^{(z_{x^*})} \\ \widehat{\omega}_l^{(z_{x^*})} \end{pmatrix} - z_{x^*}\widehat{\omega}_l^{(0)} \geq \max_{x \in \mathcal{X}_{l+1}^{(-z_{x^*})}} \begin{pmatrix} x \\ -z_{x^*} \end{pmatrix}^\top \begin{pmatrix} \widehat{\gamma}_l^{(-z_{x^*})} \\ \widehat{\omega}_l^{(-z_{x^*})} \end{pmatrix} + z_{x^*}\widehat{\omega}_l^{(0)}$$

$$+ \begin{pmatrix} x^* \\ z_{x^*} \end{pmatrix}^\top \begin{pmatrix} \widehat{\gamma}_l^{(z_{x^*})} - \gamma^* \\ \widehat{\omega}_l^{(z_{x^*})} - \omega^* \end{pmatrix} + \min_{x \in \mathcal{X}_{l+1}^{(-z_{x^*})}} \begin{pmatrix} x \\ -z_{x^*} \end{pmatrix}^\top \begin{pmatrix} \gamma^* - \widehat{\gamma}_l^{(-z_{x^*})} \\ \omega^* - \widehat{\omega}_l^{(-z_{x^*})} \end{pmatrix}$$

$$+ \Delta_{\neq} + 2z_{x^*}\left(\omega^* - \widehat{\omega}_l^{(0)}\right).$$

On $\overline{\mathcal{F}}$, it follows that

$$\begin{pmatrix} x^* \\ z_{x^*} \end{pmatrix}^\top \begin{pmatrix} \widehat{\gamma}_l^{(z_{x^*})} \\ \widehat{\omega}_l^{(z_{x^*})} \end{pmatrix} - z_{x^*}\widehat{\omega}_l^{(0)} - 2\epsilon_l \geq \max_{x \in \mathcal{X}_{l+1}^{(-z_{x^*})}} \begin{pmatrix} x \\ -z_{x^*} \end{pmatrix}^\top \begin{pmatrix} \widehat{\gamma}_l^{(-z_{x^*})} \\ \widehat{\omega}_l^{(-z_{x^*})} \end{pmatrix} + z_{x^*}\widehat{\omega}_l^{(0)} - 6\epsilon_l + \Delta_{\neq}.$$

When $\Delta_{\neq} \geq 8\epsilon_l$, this implies that $\widehat{z_{l+1}^*} = z_{x^*}$.

### C.7.13 Proof of Lemmas 10 and 17

We prove Lemma 10. The proof of Lemma 17 follows by noticing that the two actions sets are equal up to a permutation of the direction of some basis vectors. To prove Lemma 17, we rely on Elfving's characterization of $c$-optimal design, given in Theorem 5. Theorem 5 shows that for $\pi \in \mathcal{P}^{\{1,...,d+1\}}$ to be $e_{d+1}$-optimal, there must exist $t > 0$ and $\zeta \in \{-1,+1\}^{d+1}$ such that

$$\sum_{1 \leq i \leq d+1} \pi_i = 1$$

$$0 = \pi_1\zeta_1 - (1 - \frac{2}{\sqrt{\kappa_*}+1})\pi_{d+1}\zeta_{d+1}$$

$$\forall i \in \{2,...,d\},\ 0 = \pi_i\zeta_i$$

$$t = \sum_{1 \leq i \leq \lfloor d/2 \rfloor} \pi_i\zeta_i - \sum_{\lfloor d/2 \rfloor+1 \leq i \leq d+1} \pi_i\zeta_i.$$

Solving this system, we find that $t^{-2} = \kappa_*$. Note that the unicity of the solution for the corresponding probability measure $\pi$ guarantees that $te_{d+1}$ belongs to the boundary of $\mathcal{S}$.

### C.7.14 Proof of Lemma 11

For a given parameter $\gamma^*$, let us denote by $\Delta_i$ the gap corresponding to the action $i$. To compute $\kappa(\Delta)$, we could want to rely on Lemma 9 to find the $\Delta$-optimal design, corresponding to the $e_{d+1}$-optimal design on the rescaled features $\Delta_x^{-1/2}\begin{pmatrix} x \\ z_x \end{pmatrix}$. Theorem 5 indeed allows us to compute such a design, as seen in the proof of Lemma 10. Unfortunately, we cannot rescale the features using the true gaps, since $\Delta_{x^*} = 0$. To circumvent this problem, we rely on the following reasoning :

1. We use Lemma 9 and Theorem 5 to compute the design $\mu^{\Delta \vee \epsilon}$ for $\epsilon \in (0, \Delta_{\min})$; and the corresponding regret $\kappa(\Delta \vee \epsilon)$;
2. We find the value of $\kappa(\Delta)$ by noticing that $\epsilon \mapsto \kappa(\Delta \vee \epsilon)$ is continuous at 0.

For $\epsilon \in (0, \Delta_{\min})$, define $\overline{\Delta} = \Delta \vee \epsilon$, and $\overline{x} = \overline{\Delta}_x^{-1/2}x$. Let $\overline{\pi}$ denote the $e_{d+1}$-optimal design for the rescaled features $\overline{x}$, and let $\overline{\kappa_*}$ denote its variance. Then, Lemma 9 ensures that $\kappa(\overline{\Delta}) = \overline{\kappa_*}$.

Now, Theorem 5 shows that there exists $\zeta \in \{-1,+1\}^{d+1}$ such that

$$\sum_{1 \leq i \leq d+1} \overline{\pi}_i = 1$$

$$0 = \overline{\pi}_1\zeta_1\overline{\Delta}_1^{-1/2} - (1 - \frac{2}{\sqrt{\kappa_*}+1})\overline{\pi}_{d+1}\zeta_{d+1}\overline{\Delta}_{d+1}^{-1/2}$$

$$\forall i \in \{2,...,d\},\ 0 = \overline{\pi}_i\zeta_i\overline{\Delta}_i^{-1/2}$$

$$\overline{\kappa_*}^{-1/2} = \sum_{1 \leq i \leq \lfloor d/2 \rfloor} \overline{\pi}_i\zeta_i\overline{\Delta}_i^{-1/2} - \sum_{\lfloor d/2 \rfloor+1 \leq i \leq d+1} \overline{\pi}_i\zeta_i\overline{\Delta}_i^{-1/2}$$

and $\overline{\kappa_*}^{-1/2}e_{d+1}$ belongs to the boundary of $\mathcal{S}$. Solving this system, we find that

$$\kappa(\overline{\Delta})^{-1/2} = \overline{\kappa_*}^{-1/2} = \frac{\left(\frac{2}{\sqrt{\kappa_*}+1}\right)\overline{\Delta}_{d+1}^{-1/2}}{1 + \left(1 - \frac{2}{\sqrt{\kappa_*}+1}\right)\overline{\Delta}_{d+1}^{-1/2}\overline{\Delta}_1^{1/2}}.$$

As in Lemma 10, the unicity of the solution for the corresponding probability measure $\overline{\pi}$ guarantees that $\overline{\kappa}_*^{-1/2} e_{d+1}$ belongs to the boundary of the Elfving's set. Now, $\epsilon \leq \Delta_{\min}$, so

$$\kappa(\overline{\Delta})^{-1/2} = \kappa(\Delta \vee \epsilon)^{-1/2} = \frac{\left(\frac{2}{\sqrt{\kappa_*}+1}\right)\Delta_{d+1}^{-1/2}}{1 + \left(1 - \frac{2}{\sqrt{\kappa_*}+1}\right)\Delta_{d+1}^{-1/2}\epsilon^{1/2}}.$$

The fourth claim of Lemma 8 ensures that $\kappa(\Delta \vee \epsilon) \underset{\epsilon \to 0}{\to} \kappa(\Delta)$. Therefore,

$$\kappa(\Delta) = \lim_{\epsilon \to 0} \left(\frac{\left(\frac{2}{\sqrt{\kappa_*}+1}\right)\Delta_{d+1}^{-1/2}}{1 + \left(1 - \frac{2}{\sqrt{\kappa_*}+1}\right)\Delta_{d+1}^{-1/2}\epsilon^{1/2}}\right)^{-2} = \frac{(\sqrt{\kappa_*}+1)^2 \Delta_{d+1}}{4}.$$

### C.7.15 Proof of Lemma 18

Recall that $\xi_t = y_t - x_t^\top \gamma^* - z_{x_t}\omega^*$. For $l \geq 0$ and $z \in \{-1, +1\}$, when $\text{Explore}_l^{(z)} = \text{True}$, the least square estimator $\begin{pmatrix} \widehat{\gamma}_l^{(z)} \\ \widehat{\omega}_l^{(z)} \end{pmatrix}$ is given by

$$\begin{pmatrix} \widehat{\gamma}_l^{(z)} \\ \widehat{\omega}_l^{(z)} \end{pmatrix} = \left(V_l^{(z)}\right)^+ \sum_{t \in \text{Exp}_l^{(z)}} \left(\begin{pmatrix} x_t \\ z_{x_t} \end{pmatrix}^\top \begin{pmatrix} \gamma^* \\ \omega^* \end{pmatrix} + \xi_t\right) \begin{pmatrix} x_t \\ z_{x_t} \end{pmatrix}$$

$$= \left(V_l^{(z)}\right)^+ \left(V_l^{(z)}\right) \begin{pmatrix} \gamma^* \\ \omega^* \end{pmatrix} + \left(V_l^{(z)}\right)^+ \sum_{t \in \text{Exp}_l^{(z)}} \xi_t \begin{pmatrix} x_t \\ z_{x_t} \end{pmatrix},$$

where $\left(V_l^{(z)}\right)^+$ is a generalized inverse of $V_l^{(z)}$. Since $V_l^{(z)} \left(V_l^{(z)}\right)^+ V_l^{(z)} = V_l^{(z)}$, multiplying the left and right hand side of the last equation by $V_l^{(z)}$, we find that

$$V_l^{(z)} \begin{pmatrix} \widehat{\gamma}_l^{(z)} - \gamma^* \\ \widehat{\omega}_l^{(z)} - \omega^* \end{pmatrix} = V_l^{(z)} \left(V_l^{(z)}\right)^+ \sum_{t \in \text{Exp}_l^{(z)}} \xi_t \begin{pmatrix} x_t \\ z_{x_t} \end{pmatrix}. \tag{58}$$

By Lemma 4, for all $x \in \mathcal{X}_l^{(z)}$, $\begin{pmatrix} x \\ z_x \end{pmatrix} \in \text{Range}\left(V_l^{(z)}\right)$, so

$$V_l^{(z)} \left(V_l^{(z)}\right)^+ \begin{pmatrix} x \\ z_x \end{pmatrix} = \begin{pmatrix} x \\ z_x \end{pmatrix}. \tag{59}$$

Then,

$$\begin{pmatrix} \widehat{\gamma}_l^{(z)} - \gamma^* \\ \widehat{\omega}_l^{(z)} - \omega^* \end{pmatrix}^\top \begin{pmatrix} x \\ z_x \end{pmatrix} = \begin{pmatrix} \widehat{\gamma}_l^{(z)} - \gamma^* \\ \widehat{\omega}_l^{(z)} - \omega^* \end{pmatrix}^\top V_l^{(z)} \left(V_l^{(z)}\right)^+ \begin{pmatrix} x \\ z_x \end{pmatrix}$$

$$= \sum_{t \in \text{Exp}_l^{(z)}} \begin{pmatrix} x_t \\ z_{x_t} \end{pmatrix}^\top \left(V_l^{(z)}\right)^+ V_l^{(z)} \left(V_l^{(z)}\right)^+ \begin{pmatrix} x \\ z_x \end{pmatrix} \xi_t$$

$$= \sum_{t \in \text{Exp}_l^{(z)}} \begin{pmatrix} x_t \\ z_{x_t} \end{pmatrix}^\top \left(V_l^{(z)}\right)^+ \begin{pmatrix} x \\ z_x \end{pmatrix} \xi_t,$$

where the first and third lines follow from Equation (59), and the second line follows from Equation (58). By definition of our algorithm, conditionally on $\mathcal{X}_l^{(z)}$ and $\text{Explore}_l^{(z)} = \text{True}$, the variables $(\xi_t)_{t \in \text{Exp}_l^{(z)}}$ are independent centered normal gaussian variables. Then,

$$\mathbb{P}_{|\mathcal{X}_l^{(z)}, \text{Explore}_l^{(z)} = \text{True}} \left(\left|\begin{pmatrix} \widehat{\gamma}_l^{(z)} - \gamma^* \\ \widehat{\omega}_l^{(z)} - \omega^* \end{pmatrix}^\top \begin{pmatrix} x \\ z_x \end{pmatrix}\right| \geq \sqrt{2\sum_{t \in \text{Exp}_l^{(z)}} \left(\begin{pmatrix} x_t \\ z_{x_t} \end{pmatrix}^\top \left(V_l^{(z)}\right)^+ \begin{pmatrix} x \\ z_x \end{pmatrix}\right)^2 \log\left(\frac{kl(l+1)}{\delta}\right)}\right) \leq \frac{\delta}{kl(l+1)}.$$

Expanding $\left( \left( \begin{smallmatrix} x_t \\ z_{x_t} \end{smallmatrix} \right)^{\top} \left( V_l^{(z)} \right)^{+} \left( \begin{smallmatrix} x \\ z_x \end{smallmatrix} \right) \right)^2 = \left( \begin{smallmatrix} x \\ z_x \end{smallmatrix} \right)^{\top} \left( V_l^{(z)} \right)^{+} \left( \begin{smallmatrix} x_t \\ z_{x_t} \end{smallmatrix} \right) \left( \begin{smallmatrix} x_t \\ z_{x_t} \end{smallmatrix} \right)^{\top} \left( V_l^{(z)} \right)^{+} \left( \begin{smallmatrix} x \\ z_x \end{smallmatrix} \right)$, and using the definition of $V_l^{(z)}$, we find that

$$\mathbb{P}_{|\mathcal{X}_l^{(z)}, \, \text{Explore}_l^{(z)}=\text{True}} \left( \left| \left( \begin{smallmatrix} \widehat{\gamma}_l^{(z)} - \gamma^* \\ \widehat{\omega}_l^{(z)} - \omega^* \end{smallmatrix} \right)^{\top} \left( \begin{smallmatrix} x \\ z_x \end{smallmatrix} \right) \right| \geq \sqrt{2 \left( \begin{smallmatrix} x \\ z_x \end{smallmatrix} \right)^{\top} \left( V_l^{(z)} \right)^{+} V_l^{(z)} \left( V_l^{(z)} \right)^{+} \left( \begin{smallmatrix} x \\ z_x \end{smallmatrix} \right) \log \left( \frac{kl(l+1)}{\delta} \right)} \right) \leq \frac{\delta}{kl(l+1)}$$

which in turn implies (using Equation (59))

$$\mathbb{P}_{|\mathcal{X}_l^{(z)}, \, \text{Explore}_l^{(z)}=\text{True}} \left( \left| \left( \begin{smallmatrix} \widehat{\gamma}_l^{(z)} - \gamma^* \\ \widehat{\omega}_l^{(z)} - \omega^* \end{smallmatrix} \right)^{\top} \left( \begin{smallmatrix} x \\ z_x \end{smallmatrix} \right) \right| \geq \sqrt{2 \left\| \left( \begin{smallmatrix} x \\ z_x \end{smallmatrix} \right) \right\|^2_{(V_l^{(z)})^{+}} \log \left( \frac{kl(l+1)}{\delta} \right)} \right) \leq \frac{\delta}{kl(l+1)}$$

Now, using Lemma 4 and the definition of $\mu_l^z$, we see that for all $x \in \mathcal{X}_l^{(z)}$,

$$\left( \begin{matrix} x \\ z_x \end{matrix} \right)^{\top} \left( V_l^{(z)} \right)^{+} \left( \begin{matrix} x \\ z_x \end{matrix} \right) \leq \frac{\epsilon_l^2}{2 \log \left( kl(l+1)/\delta \right)}.$$

Finally, for all $x \in \mathcal{X}_l^{(z)}$,

$$\mathbb{P}_{|\mathcal{X}_l^{(z)}, \, \text{Explore}_l^{(z)}=\text{True}} \left( \left| \left( \begin{smallmatrix} \widehat{\gamma}_l^{(z)} - \gamma^* \\ \widehat{\omega}_l^{(z)} - \omega^* \end{smallmatrix} \right)^{\top} \left( \begin{smallmatrix} x \\ z_x \end{smallmatrix} \right) \right| \geq \epsilon_l \right)$$

$$\leq \mathbb{P}_{|\mathcal{X}_l^{(z)}, \, \text{Explore}_l^{(z)}=\text{True}} \left( \left| \left( \begin{smallmatrix} \widehat{\gamma}_l^{(z)} - \gamma^* \\ \widehat{\omega}_l^{(z)} - \omega^* \end{smallmatrix} \right)^{\top} \left( \begin{smallmatrix} x \\ z_x \end{smallmatrix} \right) \right| \geq \sqrt{2 \left\| \left( \begin{smallmatrix} x \\ z_x \end{smallmatrix} \right) \right\|^2_{(V_l^{(z)})^{+}} \log \left( \frac{kl(l+1)}{\delta} \right)} \right) \leq \frac{\delta}{kl(l+1)}.$$

Integrating out the conditioning on the value of $\mathcal{X}_l^{(z)}$ and $\text{Explore}_l^{(z)}$ and using a union bound yields the desire result.

### C.7.16 Proof of Lemma 19

The proof is similar to that of Lemma 18. If $\text{Explore}_l^{(0)} = \text{True}$, then $\widehat{\omega}_l$ is defined as

$$\widehat{\omega}_l^{(0)} = e_{d+1}^{\top} \left( V_l^{(0)} \right)^{+} \sum_{t \in \text{Exp}_l^{(0)}} \left( \left( \begin{matrix} x_t \\ z_{x_t} \end{matrix} \right)^{\top} \left( \begin{matrix} \gamma^* \\ \omega^* \end{matrix} \right) + \xi_t \right) \left( \begin{matrix} x_t \\ z_{x_t} \end{matrix} \right).$$

Since $\left( \begin{smallmatrix} x \\ z_x \end{smallmatrix} \right)_{x \in \mathcal{X}}$ spans $\mathbb{R}^{d+1}$, $\mu$ is finite and $e_{d+1} \in \text{Range} \left( V(\hat{\mu}_l) \right)$. Then, according to Lemma 3, for every round $l$, we have $e_{d+1} \in \text{Range} \left( V_l^{(0)} \right)$, so $V_l^{(0)} \left( V_l^{(0)} \right)^{+} e_{d+1} = e_{d+1}$. This implies that

$$\widehat{\omega}_l^{(0)} - \omega^* = \sum_{t \in \text{Exp}_l^{(0)}} e_{d+1}^{\top} \left( V_l^{(0)} \right)^{+} \left( \begin{matrix} x_t \\ z_{x_t} \end{matrix} \right) \xi_t.$$

By definition of our algorithm, conditionally on $\text{Explore}_l^{(0)} = \text{True}$, the variables $(\xi_t)_{t \in \text{Exp}_l^{(0)}}$ are independent centered normal gaussian variables. Then,

$$\mathbb{P}_{|\text{Explore}_l^{(0)}=\text{True}} \left( \left| \widehat{\omega}_l^{(0)} - \omega^* \right| \geq \sqrt{2 \sum_{t \in \text{Exp}_l^{(z)}} \left( e_{d+1}^{\top} \left( V_l^{(0)} \right)^{+} \left( \begin{matrix} x_t \\ z_{x_t} \end{matrix} \right) \right)^2 \log \left( \frac{l(l+1)}{\delta} \right)} \right) \leq \frac{\delta}{l(l+1)}.$$

Using again $V_l^{(0)} \left( V_l^{(0)} \right)^{+} e_{d+1} = e_{d+1}$ and the definition of $V_l^{(0)}$, we find that

$$\mathbb{P}_{|\text{Explore}_l^{(0)}=\text{True}} \left( \left| \widehat{\omega}_l^{(0)} - \omega^* \right| \geq \sqrt{2 e_{d+1}^{\top} \left( V_l^{(0)} \right)^{+} e_{d+1} \log \left( \frac{l(l+1)}{\delta} \right)} \right) \leq \frac{\delta}{l(l+1)}. \qquad (60)$$

Now, Lemma 3 and the definition of $\mu_l^{(0)}$ imply that

$$e_{d+1}^\top \left(V_l^{(0)}\right)^+ e_{d+1} \leq \frac{\epsilon_l^2}{2\log\left(l(l+1)/\delta\right)}.$$

Finally, Equation (60) implies that

$$\mathbb{P}_{|\text{Explore}_l^{(0)}=\text{True}}\left(\left|\widehat{\omega}_l^{(0)} - \omega^*\right| \geq \epsilon_l\right) \leq \frac{\delta}{l(l+1)}.$$

Using a union bound over the phases $\text{Exp}_l^{(0)}$ yields the result.

### C.7.17  Proof of Lemma 20

To prove Lemma 20, we begin by showing that it is enough to prove that for $l \geq 1$,

$$
\begin{aligned}
\mathcal{F}_l \supset \quad & \left\{\exists x^* \in \operatorname*{argmax}_{x\in\mathcal{X}} x^\top\gamma^* : \text{Explore}_l^{(z_{x^*})} = \text{True and } x^* \notin \mathcal{X}_{l+1}^{(z_{x^*})}\right\} \\
\cup \quad & \left\{\bigcap_{l'\leq l}\overline{\left\{\exists x^* \in \operatorname*{argmax}_{x\in\mathcal{X}} x^\top\gamma^* : \text{Explore}_{l'}^{(z_{x^*})} = \text{True and } x^* \notin \mathcal{X}_{l'+1}^{(z_{x^*})}\right\}} \right. \\
& \left. \bigcap\left\{\text{Explore}_l^{(0)} = \text{True and } \forall x^* \in \operatorname*{argmax}_{x\in\mathcal{X}} x^\top\gamma^*, \widehat{z^*}_{l+1} = -z_{x^*}\right\}\right\}.
\end{aligned}
\tag{61}
$$

Indeed, denoting $\mathcal{F}_l^{(1)} = \left\{\exists x^* \in \operatorname{argmax}_{x\in\mathcal{X}} x^\top\gamma^* : \text{Explore}_l^{(z_{x^*})} = \text{True and } x^* \notin \mathcal{X}_{l+1}^{(z_{x^*})}\right\}$ and $\mathcal{F}_l^{(2)} = \left\{\text{Explore}_l^{(0)} = \text{True and } \forall x^* \in \operatorname{argmax}_{x\in\mathcal{X}} x^\top\gamma^*, \widehat{z^*}_{l+1} = -z_{x^*}\right\}$, we see that Equation (61) would then be rewritten as

$$\mathcal{F}_l \supset \mathcal{F}_l^{(1)} \bigcup \left\{\bigcap_{l'\leq l}\overline{\mathcal{F}_{l'}^{(1)}}\bigcap\mathcal{F}_l^{(2)}\right\}$$

which implies

$$\bigcup_{l\geq 1}\mathcal{F}_l \supset \bigcup_{l\geq 1}\left\{\mathcal{F}_l^{(1)}\bigcup\left\{\left\{\bigcap_{l'\leq l}\overline{\mathcal{F}_{l'}^{(1)}}\bigcap\mathcal{F}_l^{(2)}\right\}\bigcup_{l'\leq l}\mathcal{F}_{l'}^{(1)}\right\}\right\} \supset \bigcup_{l\geq 1}\left\{\mathcal{F}_l^{(1)}\cup\mathcal{F}_l^{(2)}\right\}.$$

Then, Equation (61) would imply that

$$\overline{\mathcal{F}} = \overline{\bigcup_{l\geq 1}\mathcal{F}_l} \subset \overline{\bigcup_{l\geq 1}\left\{\mathcal{F}_l^{(1)}\bigcup\mathcal{F}_l^{(2)}\right\}} = \bigcap_{l\geq 1}\left\{\overline{\mathcal{F}_l^{(1)}}\bigcap\overline{\mathcal{F}_l^{(2)}}\right\},$$

thus proving Lemma 20. To prove Equation (61), we show that both $\mathcal{F}_l^{(1)}$ and $\bigcap_{l'\leq l}\overline{\mathcal{F}_{l'}^{(1)}}\bigcap\mathcal{F}_l^{(2)}$ imply $\mathcal{F}_l$.

**If $\mathcal{F}_l^{(1)}$ is true:** then $\exists x^* \in \operatorname{argmax}_{x\in\mathcal{X}} : \text{Explore}_l^{(z_{x^*})} = \text{True and } x^* \notin \mathcal{X}_{l+1}^{(z_{x^*})}$.
Without loss of generality, assume that $l > 1$ is the smallest integer such that $\text{Explore}_l^{(z_{x^*})} = \text{True}$ and $x^* \notin \mathcal{X}_{l+1}^{(z_{x^*})}$. Then, necessarily $x^* \in \mathcal{X}_l^{(z_{x^*})}$ (because either $l = 1$, or $\text{Explore}_{l-1}^{(z_{x^*})} = \text{True}$).
Now, because $x^* \in \mathcal{X}_l^{(z_{x^*})} \setminus \mathcal{X}_{l+1}^{(z_{x^*})}$, there exists $x \in \mathcal{X}_l^{(z_{x^*})}$ such that

$$(x - x^*)^\top\widehat{\gamma}_l^{(z_{x^*})} \geq 3\epsilon_l$$

and in particular

$$x^\top\widehat{\gamma}_l^{(z_{x^*})} - \epsilon_l > (x^*)^\top\widehat{\gamma}_l^{(z_{x^*})} + \epsilon_l.$$

Recall that by definition of $x^*$, $(\gamma^*)^\top(x^* - x) \geq 0$. This in turn implies that

$$\begin{pmatrix} x \\ z_{x^*} \end{pmatrix}^\top \begin{pmatrix} \widehat{\gamma}_l^{(z_{x^*})} - \gamma^* \\ \widehat{\omega}_l^{(z_{x^*})} - \omega^* \end{pmatrix} - \epsilon_l > \begin{pmatrix} x^* \\ z_{x^*} \end{pmatrix}^\top \begin{pmatrix} \widehat{\gamma}_l^{(z_{x^*})} - \gamma^* \\ \widehat{\omega}_l^{(z_{x^*})} - \omega^* \end{pmatrix} + \epsilon_l.$$

The last equation implies that either $\begin{pmatrix} x \\ z_x \end{pmatrix}^\top \begin{pmatrix} \gamma_l^{(z)} - \gamma^* \\ \widehat{\omega}_l^{(z)} - \omega^* \end{pmatrix} > \epsilon_l$ or $\begin{pmatrix} x^* \\ z_{x^*} \end{pmatrix}^\top \begin{pmatrix} \gamma_l^{(z)} - \gamma^* \\ \widehat{\omega}_l^{(z)} - \omega^* \end{pmatrix} < -\epsilon_l$, which in turn implies $\mathcal{F}_l$.

**If $\bigcap_{l' \leq l} \overline{\mathcal{F}_{l'}^{(1)}} \bigcap \mathcal{F}_l^{(2)}$ is true:** then $\text{Explore}_l^{(0)} = \text{True}$ and $\forall x^* \in \text{argmax}_{x \in \mathcal{X}} x^\top \gamma^*, \widehat{z_{l+1}^*} = -z_{x^*}$. Moreover, for all $l' \leq l$, $\text{Explore}_{l'}^{(z_{x^*})} = \text{False}$ or $x^* \in \mathcal{X}_{l'+1}^{(z_{x^*})}$.

Note that this case can only hold if all optimal actions $x^*$ belong to the same group $z_{x^*}$. Without loss of generality, assume that $l > 1$ is the smallest integer such that $\text{Explore}_l^{(0)} = \text{True}$ and $\widehat{z_{l+1}^*} = -z_{x^*}$, and for all $l' \leq l$, $\text{Explore}_{l'}^{(z_{x^*})} = \text{False}$ or $x^* \in \mathcal{X}_{l'+1}^{(z_{x^*})}$. Note that because $\text{Explore}_l^{(0)} = \text{True}$, necessarily $\text{Explore}_{l'}^{(z_{x^*})} = \text{True}$ for all $l' \leq l$, and in particular $x^* \in \mathcal{X}_{l+1}^{(z_{x^*})}$.

Then, there exists $x \in \mathcal{X}_{l+1}^{(-z_{x^*})}$ such that

$$\begin{pmatrix} x \\ -z_{x^*} \end{pmatrix}^\top \begin{pmatrix} \widehat{\gamma}_l^{(-z_{x^*})} \\ \widehat{\omega}_l^{(-z_{x^*})} \end{pmatrix} - \begin{pmatrix} x^* \\ z_{x^*} \end{pmatrix}^\top \begin{pmatrix} \widehat{\gamma}_l^{(z_{x^*})} \\ \widehat{\omega}_l^{(z_{x^*})} \end{pmatrix} + 2z_{x^*} \widehat{\omega}_l^{(0)} \geq 4\epsilon_l.$$

Recall that all optimal actions $x^*$ are in the same group $z_{x^*}$, so $(\gamma^*)^\top (x^* - x) > 0$. This in turn implies that

$$\begin{pmatrix} x \\ -z_{x^*} \end{pmatrix}^\top \begin{pmatrix} \widehat{\gamma}_l^{(-z_{x^*})} - \gamma^* \\ \widehat{\omega}_l^{(-z_{x^*})} - \omega^* \end{pmatrix} - \begin{pmatrix} x^* \\ z_{x^*} \end{pmatrix}^\top \begin{pmatrix} \widehat{\gamma}_l^{(z_{x^*})} - \gamma^* \\ \widehat{\omega}_l^{(z_{x^*})} - \omega^* \end{pmatrix} + 2z_{x^*} (\widehat{\omega}_l^{(0)} - \omega^*) \geq 4\epsilon_l.$$

The last equation implies that either $\begin{pmatrix} x \\ -z_{x^*} \end{pmatrix}^\top \begin{pmatrix} \widehat{\gamma}_l^{(-z_{x^*})} - \gamma^* \\ \widehat{\omega}_l^{(-z_{x^*})} - \omega^* \end{pmatrix} \geq \epsilon_l$, or $\begin{pmatrix} x^* \\ z_{x^*} \end{pmatrix}^\top \begin{pmatrix} \widehat{\gamma}_l^{(z_{x^*})} - \gamma^* \\ \widehat{\omega}_l^{(z_{x^*})} - \omega^* \end{pmatrix} \leq -\epsilon_l$, or $z_{x^*} (\widehat{\omega}_l^{(0)} - \omega^*) \geq \epsilon_l$, which in turn implies $\mathcal{F}_l$.

### C.7.18   Proof of Lemma 21

The first claim holds for $l = 1$. For $l \geq 1$, for any $x \in \mathcal{X}_{l+1}^{(-1)} \cup \mathcal{X}_{l+1}^{(1)}$, we have $\widehat{\Delta}_x^{l+1} \leq \Delta_x + 8\epsilon_l$ on $\overline{\mathcal{F}}$ according to the definition of $\widehat{\Delta}^{l+1}$ and $\mathcal{F}$. The first claim then follows.

For the second claim, Lemma 13 gives that, on $\overline{\mathcal{F}}$, $\Delta_x < 21\epsilon_l$ for any $x \in \mathcal{X}_{l+1}^{(-1)} \cup \mathcal{X}_{l+1}^{(1)}$. So $\Delta_x \geq 21\epsilon_l$ implies $x \notin \mathcal{X}_{l+1}^{(-1)} \cup \mathcal{X}_{l+1}^{(1)}$ and hence $l \geq \ell_x$ on $\overline{\mathcal{F}}$.

For the third claim, we notice that

$$\max_{x' \in \mathcal{X}_{\ell_x}^{(z_x)}} (a_{x'} - a_x)^\top \widehat{\theta}_{\ell_x}^{(z_x)} > 3\epsilon_{\ell_x},$$

since $x \notin \mathcal{X}_{\ell_x+1}$. Since the left-hand side is smaller than $\Delta_x + 2\epsilon_{\ell_x}$ on $\overline{\mathcal{F}}$, we get $\Delta_x > \epsilon_{\ell_x}$.

## D   Extension to $M$ groups

**Model**   We extend the biased linear bandit to $Z$ groups, denoted $\mathcal{Z} = \{1, ..., Z\}$. The evaluations are given by

$$y_t = x_t^\top \gamma + Z_{x_t}^\top \omega + \xi_t,$$

where $Z_x$ is the $z_x$-th vector of the canonical basis in $\mathbb{R}^Z$, and $\omega = \{\omega_1, ..., \omega_Z\} \in \mathbb{R}^Z$ is the vector of biases. Note that for the model to be identifiable, we must assume it does not contain an intercept. For $x \in \mathcal{X}$, we denote $a_x = \begin{pmatrix} x \\ Z_x \end{pmatrix}$. To ensure identifiability of the model, we further assume that the set $\mathcal{A} = \{a_x : x \in \mathcal{X}\}$ spans $\mathbb{R}^{d+Z}$.

**Estimation of the biased evaluations**   Adapting the G-EXP-ELIM routine to the multiple group framework is rather straightforward. Note that this routine can be used as is to eliminate within-group sub-optimal actions. The actions of each group span a sub-space of dimension $d+1$, so the G-optimal measure is still supported by $O(d^2)$ points. Moreover, the variance corresponding to this G-optimal design is still $d+1$.

**Estimation of the bias**  By contrast, the bias elimination routine must be modified in order to handle $Z$ groups. At each phase $l$, we denote by $\mathcal{Z}_l$ the set of groups that have not been eliminated yet. If more than one group remain in $\mathcal{Z}_l$, we compute the difference $\omega_1 - \omega_z$ for all group $z$ remaining in $\mathcal{Z}_l$ with precision $\epsilon_l/2$ using a modified $\Delta$-EXP-ELIM routine, which we call $\Delta$-MULT-EXP-ELIM, described in 5. This routine samples action according to the distribution $\mu_z$, where for any groups $z \neq 1$, we defined $\mu_z$ as the solution of the problem

$$\underset{\mu \in \mathcal{M}_{\mathcal{X}}^{e_{d+1}-e_{d+z}}}{\text{minimize}} \sum_x \mu(x)\Delta_x \quad \text{such that} \quad (e_{d+1} - e_{d+z})^\top V(\mu)^+ (e_{d+1} - e_{d+z}) \leq 1 \quad (62)$$

We also define $\widetilde{\kappa}_z(\Delta)$ as the corresponding regret :

$$\widetilde{\kappa}_z(\Delta) \quad = \quad \sum_x \mu_z(x)\Delta_x.$$

Note that the support of the distribution $\mu_z$ is at most of size $d + Z$. This two-by-two comparison allows us to compute, for each $z, z' \in \mathcal{Z}_l$, the difference of bias $\omega_z - \omega_{z'} = \omega_1 - \omega_{z'} - (\omega_1 - \omega_z)$ with precision level $\epsilon_l$. Then, we can use these bias estimates to eliminate groups that are sub-optimal by a gap larger than $4\epsilon_l$. Again, we rely on estimates of the biases and of the biased evaluations obtained during the previous round to update the estimate of the gap vector $\widehat{\Delta}^{l+1}$.

---

**Algorithm 5** $\Delta$-MULT-EXP-ELIM $(\mathcal{X}, \mathcal{Z}, (\mathcal{X}^{(z)}, \widehat{\theta}^{(z)})_{z \in \mathcal{Z}}, \widehat{\Delta}, n, \epsilon)$

1: **for** $z \in \mathcal{Z}$, $z \neq 1$ **do**
2:      Compute $\widehat{\Delta}$-optimal design $\hat{\mu}_z$ solution of (62) on $\mathcal{X}$, with $|\operatorname{supp}(\hat{\mu}_z)| \leq d + Z$
3:      Sample $\lceil n\hat{\mu}_z(x) \rceil$ times each action $a_x$ for $x \in \mathcal{X}$
4:      Compute $\widehat{\omega}_1 - \widehat{\omega}_z = (e_{d+1} - e_{d+z})^\top \widehat{\theta}$, where $\widehat{\theta}$ is the ordinary least square estimator
5: **for** $z \in \mathcal{Z}$ and $x \in \mathcal{X}^{(z)}$ **do** $\widehat{m}_x \leftarrow a_x^\top \widehat{\theta}^{(z)} + (\widehat{\omega}_1 - \widehat{\omega}_z)$
6: **for** $z \in \mathcal{Z}$ and $x \in \mathcal{X}^{(z)}$ **do** $\widehat{\Delta}_x \leftarrow 2 \wedge \left( \max_{z' \in \mathcal{Z}, x' \in \mathcal{X}^{(z')}} \widehat{m}_{x'} - \widehat{m}_x + 4\epsilon \right)$
7: **for** $z \in \mathcal{Z}$ **do**
8:      **if** $\max_{z' \in \mathcal{Z}} \max_{x \in \mathcal{X}^{(z')}} a_x^\top \widehat{\theta}^{(z')} + (\widehat{\omega}_1 - \widehat{\omega}_{z'}) \geq \max_{x \in \mathcal{X}^{(z)}} a_x^\top \widehat{\theta}^{(z)} + (\widehat{\omega}_1 - \widehat{\omega}_z) + 4\epsilon$ **then** $\mathcal{Z} \leftarrow \mathcal{Z} \setminus \{z\}$
9: **return** $\mathcal{Z}$ and $\widehat{\Delta}$

---

**Stopping criterion**  We denote by $\widetilde{\kappa}_{\mathcal{Z}_l}(\widehat{\Delta}^l) = \sum_{z \in \mathcal{Z}_l, z \neq 1} \widetilde{\kappa}_z(\widehat{\Delta}^l)$ the regret for estimating the biases at phase $l$. If $\epsilon_l \leq \left( \widetilde{\kappa}_{\mathcal{Z}_l}(\widehat{\Delta}^l) \log(T)/T \right)^{1/3}$, bias estimation becomes too costly, so we sample the empirical best action for the remaining time. The FAIR PHASED ELIMINATION FOR MULTIPLE GROUPS algorithm is presented in 6.

### D.1  Worst case regret

Before analyzing the worst case regret of Algorithm 6, we introduce a new quantity, $\widetilde{\kappa}_*$, defined as

$$\widetilde{\kappa}_* = \sum_{z \in \mathcal{Z}, z \neq 1} \min_{\pi \in \mathcal{P}_{e_{d+1}-e_{d+z}}^{\mathcal{X}}} (e_{d+1} - e_{d+z})^\top (V(\pi))^+ (e_{d+1} - e_{d+z}).$$

Note that for all $z \in \mathcal{Z}$, $z \neq 1$, and $l \geq 1$, we have $\widetilde{\kappa}_{\mathcal{Z}_l}(\widehat{\Delta}^l) \leq 2\widetilde{\kappa}_*$.

**Claim 1.** *For the choice $\delta = T^{-1}$, there exists an absolute constant $C > 0$ and a constant $T_{\widetilde{\kappa}_*, k, Z, d, k}$ depending on $\widetilde{\kappa}_*, k, Z, d$, and $k$ such that the following bound on the regret of the* FAIR PHASED ELIMINATION FOR MULTIPLE GROUPS *algorithm 6 holds*

$$R_T \leq CZ \left( \widetilde{\kappa}_* \log(T) \right)^{1/3} T^{2/3} \qquad for \quad T \geq T_{\widetilde{\kappa}_*, k, Z, d, k}.$$

*Sketch of Proof.*  We sketch here a proof of Claim 1, highlighting the main differences with the two-groups setting. We begin by introducing some notations.

**Algorithm 6** FAIR PHASED ELIMINATION FOR MULTIPLE GROUPS

1: **input:** $\delta, T, \mathcal{X}, k = |\mathcal{X}|, \epsilon_l = 2^{2-l}$ for $l \geq 1$
2: **initialize:** $\widehat{\Delta}^1 \leftarrow (2, ..., 2), l \leftarrow 0, \mathcal{Z}_1 = \mathcal{Z}$
3: **for** $z \in \mathcal{Z}_1$ **do** $\mathcal{X}_1^{(z)} \leftarrow \{x : z_x = z\}$
4: **while** the budget is not spent **do** $l \leftarrow l + 1$
5:     **for** $z \in \mathcal{Z}_l$ **do**
6:         $\left( \widehat{\theta}^{(z)}, \mathcal{X}_{l+1}^{(z)} \right) \leftarrow$ G-EXP-ELIM $\left( \mathcal{X}_l^{(z)}, \frac{2(d+1)}{\epsilon_l^2} \log \left( \frac{kl(l+1)}{\delta} \right), \epsilon_l \right)$
7:     **if** $|\mathcal{Z}_l| > 1$ **then**
8:         Compute $\widetilde{\kappa}_{\mathcal{Z}_l}(\widehat{\Delta}^l) = \sum\limits_{z \in \mathcal{Z}_l, z \neq 1} \widetilde{\kappa}_z(\widehat{\Delta}^l)$.
9:         **if** $\epsilon_l \leq \left( \widetilde{\kappa}_{\mathcal{Z}_l}(\widehat{\Delta}^l) \log(T)/T \right)^{1/3}$ **then**         ▷ Stop bias estimation
10:             Sample best action in $\cup_{z \in \mathcal{Z}_l} \mathcal{X}_{l+1}^{(z)}$ for the remaining time
11:     **else**
12:         $\left( \mathcal{Z}_{l+1}, \widehat{\Delta}^{l+1} \right) \leftarrow \Delta$-MULT-EXP-ELIM $\left( \mathcal{X}, \mathcal{Z}_l, \left( \mathcal{X}_{l+1}^{(z)}, \widehat{\theta}_l^{(z)} \right)_{z \in \mathcal{Z}_l}, \widehat{\Delta}^l, \frac{8}{\epsilon_l^2} \log \left( \frac{Zl(l+1)}{\delta} \right), \epsilon_l \right)$

**Notations** We denote by $L_T$ the largest integer $l$ such that $\epsilon_l \geq \left( 2\widetilde{\kappa}_*^{1/3} \log(T)/T \right)^{1/3}$. For $z \in \mathcal{Z}$, we denote by $L^\Delta$ the last phase where $\widehat{\Delta}^l$-optimal Exploration and Elimination is performed. We denote by Exp-G$_l^{(z)}$ the time indices where G-exploration is performed on $\mathcal{X}_l^{(z)}$ and by Exp-D$_l^{(z)}$ the time indices where $\Delta$-exploration is performed at phase $l$ for estimating the difference $\omega_1 - \omega_z$. We also denote by Recovery the time indices subsequent to the stopping criterion, this set being empty when the stopping criterion is not activated.

We define a "good" event $\overline{\mathcal{F}}$ such that for all $z, z' \in \mathcal{Z}$ and all $x \in \mathcal{X}_1^{(z)}$, the errors $\left| a_x^\top \left( \theta^* - \widehat{\theta}_l^{(z)} \right) \right|$ and $|(\omega_z^* - \omega_{z'}^*) - ((\widehat{\omega}_l)_z - (\widehat{\omega}_l)_{z'})|$ are smaller than $\epsilon_l$ for all $l$ such that these quantities are defined. In the following, we use $c, c'$ to denote positive absolute constants, which may vary from line to line. With these notations, we decompose the regret as follows :

$$
R_T \leq 2T\mathbb{P}(\mathcal{F}) + \mathbb{E}_{|\overline{\mathcal{F}}} \left[ \underbrace{\sum_{l \leq L_T} \sum_{z \in \mathcal{Z}_l} \sum_{t \in \text{Exp-G}_l^{(z)}} (x^* - x_t)^\top \gamma^*}_{R_T^G} \right] + \mathbb{E}_{|\overline{\mathcal{F}}} \left[ \underbrace{\sum_{l \leq L^\Delta} \sum_{z \in \mathcal{Z}_l, z \neq 1} \sum_{t \in \text{Exp-D}_l^{(z)}} (x^* - x_t)^\top \gamma^*}_{R_T^\Delta} \right]
$$

$$
+ \mathbb{E}_{|\overline{\mathcal{F}}} \left[ \underbrace{\sum_{l \geq L_T+1} \sum_{z \in \mathcal{Z}_l} \sum_{t \in \text{Exp-G}_l^{(z)}} (x^* - x_t)^\top \gamma^* + \sum_{t \in \text{Recovery}} (x^* - x_t)^\top \gamma^*}_{R_T^{Rec}} \right].
$$

**Bound on** $T\mathbb{P}(\mathcal{F})$. Using arguments based on concentration of Gaussian variables, we can show that $\mathbb{P}(\mathcal{F}) \leq 2T^{-1}$.

**Bound on** $R_T^G$. The analysis is similar to the two-groups setting. We can show that on $\overline{\mathcal{F}}$, only actions with gaps smaller than $c\epsilon_l$ remain in the sets $\mathcal{X}_l^{(z)}$ for $z \in \mathcal{Z}_l$. The length of each G-optimal Exploration and Elimination phase for one group is of the order $(d+1) \log(klT)/\epsilon_l^2$, so the regret corresponding to phase $l$ is of the order $Z(d+1) \log(klT)/\epsilon_l$. Summing over the different phases, we find that

$$
R_T^G \leq c(d+1)Z \log(kL_T T)/\epsilon_{L_T}. \tag{63}
$$

Using the definition of $L_T$, we find that $R_T^G \leq c(d+1)Z \log(kL_T T)\widetilde{\kappa}_*^{-1/3} \log(T)^{-1/3} T^{1/3}$.

**Bound on $\mathbf{R_T^{Rec}}$.** On the one hand, the actions selected during the Phases Exp-G$_l^{(z)}$ for $l \geq L_T + 1$ are sub-optimal by a gap at most $c\epsilon_{L_T}$ on the event $\overline{\mathcal{F}}$. On the other hand, if the algorithm enters the Recovery phase at a phase $l$, then

$$\epsilon_l \leq \widetilde{\kappa}_{\mathcal{Z}_{L^\Delta}} (\widehat{\Delta}^{L^\Delta})^{1/3} T^{-1/3} \log(T)^{1/3} \leq 2\widetilde{\kappa}_*^{1/3} T^{-1/3} \log(T)^{1/3},$$

so we must have $l = L^\Delta + 1 \geq L_T + 1$. Therefore, all actions selected during the Recovery phase are sub-optimal by a gap at most $c\epsilon_{L_T}$. Then, $R_T^{Rec}$ can be bounded as $R_T^{Rec} \leq c\epsilon_{L_T} T$. This implies in particular that $R_T^{Rec} \leq c'\widetilde{\kappa}_*^{1/3} \log(T)^{1/3} T^{2/3}$.

**Bound on $R_T^\Delta$.** To bound $R_T^\Delta$, we introduce further notations. Let us denote by $l_1, ..., l_R$ the phases at which at least one group is eliminated, by $S_1i$ the sets of groups remaining at the beginning of phase $l_i$, and by $S_{R+1}$ the set of groups that are never eliminated. We also write $l_{R+1} = L^\Delta$. We abuse notations and denote Exp-D$_l^{(S)} = \cup_{z \in S}$ Exp-D$_l^{(z)}$. Then, we see that

$$R_T^\Delta \leq \sum_{i \leq R+1} \sum_{l \leq l_i} \sum_{t \in \text{Exp-D}_l^{(S_i)}} (x^* - x_t)^\top \gamma^*.$$

The rest of the proof is similar to that in the two-communities setting. We show that on $\mathcal{F}$, $\widehat{\Delta}^l \geq \Delta$ for all $l \geq 1$. Then, our choice of design $\widehat{\mu}_{z_l,z}$ at phase $l$ ensures that for $i \leq R + 1$, on $\overline{\mathcal{F}}$,

$$\sum_{t \in \text{Exp-D}_l^{(S_i)}} (x^* - x_t)^\top \gamma^* \leq c \sum_{z \in S_i} \left( \frac{\log(Zl(l+1)T)}{\epsilon_l^2} \widetilde{\kappa}_z(\widehat{\Delta}^l) + d + 1 \right)$$

for some constant $c > 0$. Using arguments similar to the two-groups setting, we can sum over the different phases $l \leq l_i$, and find that

$$\sum_{l \leq l_i} \sum_{t \in \text{Exp-D}_l^{(S_i)}} (x^* - x_t)^\top \gamma^* \leq c\widetilde{\kappa}_{S_i}(\widehat{\Delta}^{l_i}) \log(Zl_i T)/\epsilon_{l_i}^2. \tag{64}$$

By definition of $S_i$ we have that $\widetilde{\kappa}_{\mathcal{Z}_{l_i}}(\widehat{\Delta}^{l_i}) = \widetilde{\kappa}_{S_i}(\widehat{\Delta}^{l_i})$. Now, the algorithm does not enter the Recovery phase before phase $l_i + 1$, so we must have
$\epsilon_{l_i}^{-2} \leq T^{2/3} \log(T)^{-2/3} \widetilde{\kappa}_{\mathcal{Z}_{l_i}}(\widehat{\Delta}^{l_i})^{-2/3}$. This implies that

$$\sum_{l \leq l_i} \sum_{t \in \text{Exp-D}_l^{(S_i)}} (x^* - x_t)^\top \gamma^* \leq c\widetilde{\kappa}_{\mathcal{Z}_{l_i}}(\widehat{\Delta}^{l_i})^{1/3} \left( \log(T)^{1/3} + \log(Z) \right) T^{2/3}.$$

We use that $\widetilde{\kappa}_{\mathcal{Z}_{l_i}}(\widehat{\Delta}^{l_i}) \leq \widetilde{\kappa}_*$ and sum over $i \leq R + 1 < Z$, and we find that
$R_T^\Delta \leq CZ\widetilde{\kappa}_*^{1/3} \log(T)^{1/3} T^{2/3}$ for $T$ large enough.

When $T \geq T_{\widetilde{\kappa}_*,k,Z,d,k}$ for some $T_{\widetilde{\kappa}_*,k,Z,d,k}$ large enough, we find that $\mathbb{R}_T \leq c'Z\widetilde{\kappa}_*^{1/3} \log(T)^{1/3} T^{2/3}$. $\qquad\qquad\qquad\qquad\qquad\qquad\qquad\qquad\qquad\qquad\qquad\qquad\qquad \square$

## D.2 Gap-dependent regret

Before stating the bound on the gap-dependent regret, we introduce further notations. For $z \in \mathcal{Z}$, we denote $\Delta_{\neq,z} = \min_{x:z_x=z} \Delta_x$, $\Delta_{\neq} = \min_{x:z\neq z^*} \Delta_{\neq,z}$, $\Delta_{\min} = \min_{x \in \mathcal{X} \setminus x^*} \Delta_x$, and $\varepsilon_T = (\widetilde{\kappa}_* \log(T)/T)^{1/3}$.
Then, we claim that the following gap-dependent regret bound on the regret of Algorithm 5 holds.

**Claim 2.** *Assume that $x^* \in \operatorname{argmax}_{x \in \mathcal{X}} x^\top \gamma^*$ is unique. Then, there exists an absolute constant $C > 0$ and a constant $T_{\widetilde{\kappa}_*,k,Z,d,k,\Delta_{\neq},\Delta_{\min}}$ depending on $\widetilde{\kappa}_*, k, Z, d, k, \Delta_{\min}$, and $(\Delta_{\neq,z})_{z \neq z^*}$ such that the following bound on the regret of the* FAIR PHASED ELIMINATION FOR MULTIPLE GROUPS *algorithm 6 holds for $T \geq T_{\widetilde{\kappa}_*,k,Z,d,k,\Delta_{\neq},\Delta_{\min}}$*

$$R_T \leq C \left( \frac{d}{\Delta_{\min}} \vee \sum_{z \neq z^*, z \neq 1} \frac{\kappa_z(\Delta \vee \Delta_{\neq,z} \vee \varepsilon_T)}{(\Delta_{\neq,z})^2} + \frac{\kappa_{z^*}(\Delta \vee \Delta_{\neq} \vee \varepsilon_T)}{(\Delta_{\neq})^2} \right) \log(T).$$

*Sketch of Proof.* We sketch here a proof of Claim 2. We begin by introducing some notations.

**Notations** We define a "good" event $\overline{\mathcal{F}}$ such that for all $z, z' \in \mathcal{Z}$ and all $x \in \mathcal{X}_1^{(z)}$, the errors $\left| a_x^\top \left( \theta^* - \widehat{\theta}_l^{(z)} \right) \right|$ and $|(\omega_z^* - \omega_{z'}^*) - ((\widehat{\omega}_l)_z - (\widehat{\omega}_l)_{z'})|$ are smaller than $\epsilon_l$ for all $l$ such that these quantities are defined. For each group $z \in \mathcal{Z}$, we denote by Exp-G$_l^{(z)}$ the time indices where G-exploration is performed on $\mathcal{X}_l^{(z)}$. For $z \in \mathcal{Z}, z \neq 1$, we denote by Exp-D$_l^{(z)}$ the time indices where $\Delta$-exploration is performed at phase $l$ to estimate the difference $\omega_1 - \omega_z$, and by $L^{(z)}$ the last phase $l$ such that $z \in \mathcal{Z}_l$ and bias exploration is performed at this phase. We denote by $L^\Delta$ the last phase $l$ where bias estimation is performed. Moreover, we denote by $S$ the sets of groups eliminated before the stopping criterion is activated, and write $\overline{S} = \mathcal{Z} \setminus S$. We abuse notations and denote Exp-D$_l^{(S)} = \cup_{z \in S}$ Exp-D$_l^{(z)}$. We also denote by Recovery the time indices subsequent to the stopping criterion, this set being empty when the stopping criterion is not activated. In the following, we use $c, c'$ to denote positive absolute constants, which may vary from line to line.

**Fact 1** Let $l_{\Delta_{\min}}$ be the largest integer such that $\epsilon_{l_{\Delta_{\min}}} \geq C\Delta_{\min}$ for some well-chosen absolute constant $C > 0$. Similarly to the two-groups setting, we can show that on the good event $\overline{\mathcal{F}}$, no more than $l_{\Delta_{\min}}$ G-optimal Exploration and Elimination phases are needed to find the best action. For all phases $l \geq l_{\Delta_{\min}}$, the algorithm always chooses $x^*$, and suffers no regret.

**Fact 2** Similarly to the two-groups setting, we can show that on the good event $\overline{\mathcal{F}}$, for each phase $l$, $\widehat{\Delta}^l \leq c(\Delta \vee \epsilon_l)$ for some constant $c$. Moreover, for all $l \leq L^\Delta$, all groups $z \neq 1$, and all $\tau > 0$, $\widetilde{\kappa}_z(\widehat{\Delta}^l) \leq c\widetilde{\kappa}_z(\Delta \vee \epsilon_l) \leq c(1 + \epsilon_l\tau^{-1})\widetilde{\kappa}_z(\Delta \vee \tau)$.

**Fact 3** For $z \in \mathcal{Z} \setminus \{z^*\}$, let $l_{\Delta_{\neq,z}}$ be the largest integer such that $\epsilon_{l_{\Delta_{\neq,z}}} \geq C\Delta_{\neq,z}$ for some well-chosen absolute constant $C > 0$. On the good event $\overline{\mathcal{F}}$, if $\widehat{\Delta}^l$-optimal Exploration and Elimination is performed at phase $l \geq l_{\Delta_{\neq,z}}$, and $z \in \mathcal{Z}_l$, then the algorithm eliminates $z$ at this phase. This implies that $L^{(z)} \leq l_{\Delta_{\neq,z}}$, and that $L^\Delta \leq l_{\Delta_{\neq}}$.

**Fact 4** We denote by $L_T$ the largest integer $l$ such that $\epsilon_l \geq (2\widetilde{\kappa}_* \log(T)/T)^{1/3}$. Since $2\widetilde{\kappa}_* \geq \widetilde{\kappa}(\widehat{\Delta}^l)$ for all $l \geq 1$ and all $z \in \mathcal{Z}$, we see that if the algorithm enters the Recovery phase, we must have $L_T \leq L^\Delta$, and $\epsilon_{L^\Delta} \leq \epsilon_{L_T} \approx \varepsilon_T$.

Using **Fact 1**, we find that the regret can be written as

$$
\begin{aligned}
R_T \leq \ & 2T\mathbb{P}(\mathcal{F}) + \mathbb{E}_{|\overline{\mathcal{F}}}\left[ \underbrace{\sum_{l \leq l_{\Delta_{\min}}} \sum_{z \in \mathcal{Z}_l} \sum_{t \in \text{Exp-G}_l^{(z)}} (x^* - x_t)^\top \gamma^*}_{R_T^G} \right] + \mathbb{E}_{|\overline{\mathcal{F}}}\left[ \underbrace{\sum_{z \in S} \sum_{l \leq L^{(z)}} \sum_{t \in \text{Exp-D}_l^{(z)}} (x^* - x_t)^\top \gamma^*}_{R_T^{\Delta,S}} \right] \\
& + \mathbb{E}_{|\overline{\mathcal{F}}}\left[ \underbrace{\sum_{l \leq L^\Delta} \sum_{t \in \text{Exp-D}_l^{(\overline{S})}} (x^* - x_t)^\top \gamma^*}_{R_T^{\Delta,\overline{S}}} \right] + \mathbb{E}_{|\overline{\mathcal{F}}}\left[ \underbrace{\sum_{t \in \text{Recovery}} (x^* - x_t)^\top \gamma^*}_{R_T^{Rec}} \right].
\end{aligned}
$$

**Bound on $R_T^G$.** We rely on arguments similar to those used in Equation (63) to show that $R_T^G \leq c(d+1)\log(kl_{\Delta_{\min}}T)\epsilon_{l_{\Delta_{\min}}}^{-1}$. Since $\epsilon_{l_{\Delta_{\min}}} \geq C\Delta_{\min}$, this implies that

$$
R_T^G \leq \frac{c(d+1)\log(kl_{\Delta_{\min}}T)}{\Delta_{\min}} \leq \frac{c'd\log(T)}{\Delta_{\min}}
$$

if $T \geq k$.

**Bound on $R_T^{\Delta,S}$.** Using arguments similar to the two-groups settings, we can show that for all $z \neq 1$

$$
\sum_{l \leq L^{(z)}} \sum_{t \in \text{Exp-D}_l^{(z)}} (x^* - x_t)^\top \gamma^* \leq c\widetilde{\kappa}_z(\widehat{\Delta}^{L^{(z)}})\log(l_{L^{(z)}}T)\epsilon_{L^{(z)}}^{-2}. \tag{65}
$$

Using **Fact 2** with $\tau = \Delta_{\neq,z}$ together with **Fact 3**, we find that

$$R_T^{\Delta,S} \le c \sum_{z \in \mathcal{S}} \widetilde{\kappa}_z(\Delta \vee \Delta_{\neq,z}) \log(L^{(z)}T)(\Delta_{\neq,z})^{-2}.$$

**Bound on $\mathbf{R_T^{\Delta,\overline{S}} + R_T^{Rec}}$.** If the algorithm does not enter the Recovery phase, then $R_T^{Rec} = 0$ and $\overline{S} = \{z^*\}$. Then, the algorithms finds the best group, and the last bias exploration phase is performed at phase $\max_{z \neq z^*} L^{(z)} \le \max_{z \neq z^*} l_{\Delta_{\neq,z}} = l_{\Delta_{\neq}}$. Then, Equation (65) implies that

$$R_T^{\Delta,\overline{S}} \le c\widetilde{\kappa}_{z^*}(\Delta \vee \Delta_{\neq}) \log(L^{(z)}T)(\Delta_{\neq})^{-2}.$$

If the algorithms enters the Recovery phase, we can use again the same arguments to show that $R_T^{\Delta,\overline{S}} \le c \sum_{z \in \overline{S}} \widetilde{\kappa}_z(\widehat{\Delta}^{L^\Delta}) \log(l_{L^{(z)}}T)\epsilon_{L^{(z)}}^{-2}$. Using **Fact 2** and Equation (65), we find that for $\tau = \epsilon_{L^\Delta}$,

$$R_T^{\Delta,\overline{S}} \le c \sum_{z \in \overline{S}} \widetilde{\kappa}_z(\Delta \vee \epsilon_{L^\Delta}) \log(l_{L^\Delta}T)\epsilon_{L^\Delta}^{-2} = c\frac{\widetilde{\kappa}_{\overline{S}}(\Delta \vee \epsilon_{L^\Delta}) \log(l_{L^\Delta}T)}{\epsilon_{L^\Delta}^2}.$$

Since all actions selected during the Recovery phase belong to $\cup_{z \in \overline{S}} \mathcal{X}_l^{(z)}$, on $\overline{\mathcal{F}}$ these actions are sub-optimal by a gap at most $c\epsilon_{L^\Delta+1}$, so $R_T^{Rec} \le cT\epsilon_{L^\Delta+1}$. Now, since the algorithm enters the Recovery phase, we must have $\epsilon_{L^\Delta+1} \le (\widetilde{\kappa}_{\overline{S}}(\Delta^{L^\Delta+1}) \log(T)/T)^{1/3}$, which implies that

$$R_T^{Rec} \le \frac{c\widetilde{\kappa}_{\overline{S}}(\widehat{\Delta}^{L^\Delta+1}) \log(T)}{\epsilon_{L^\Delta+1}^2}.$$

Together with **Fact 2**, this implies that

$$R_T^{\Delta,\overline{S}} + R_T^{Rec} \le \frac{c\widetilde{\kappa}_{\overline{S}}(\Delta \vee \epsilon_{L^\Delta}) \log(T)}{\epsilon_{L^\Delta}^2}.$$

On the one hand, **Fact 3** guarantees that, since we entered the Recovery phase before eliminating any group in $\overline{S}$, we must have $L^\Delta \le \min_{z \in \overline{S} \setminus \{z^*\}} l_{\Delta_{\neq,z}}$, so $\epsilon_{L^\Delta} \ge c\max_{z \in \overline{S}} \Delta_{\neq,z}$. On the other hand, **Fact 4** ensures that $\epsilon_{L^\Delta} \le \varepsilon_T$. Thus,

$$R_T^\Delta + R_T^{Rec} \le \sum_{s \in \overline{S} \setminus \{z^*\}} \frac{c\widetilde{\kappa}_z(\Delta \vee \varepsilon_T) \log(T)}{(\Delta_{\neq,z})^2} + \frac{c\widetilde{\kappa}_{z^*}(\Delta \vee \varepsilon_T) \log(T)}{(\Delta_{\neq})^2}.$$

Conclusion Combining these results, we find that

$$R_T \le c\left(\frac{d}{\Delta_{\min}} \vee \sum_{z \neq z^*, z \neq 1} \frac{\widetilde{\kappa}_z(\Delta \vee \Delta_{\neq,z}) \vee \widetilde{\kappa}_z(\Delta \vee \varepsilon_T)}{(\Delta_{\neq,z})^2} + \frac{\widetilde{\kappa}_{z^*}(\Delta \vee \Delta_{\neq}) \vee \widetilde{\kappa}_{z^*}(\Delta \vee \varepsilon_T)}{(\Delta_{\neq})^2}\right) \log(T)$$

when $T \ge k$. Using Lemma 8, we get that $\widetilde{\kappa}_z(\Delta \vee \Delta_{\neq}) \vee \widetilde{\kappa}_z(\Delta \vee \varepsilon_T) \le \widetilde{\kappa}_z(\Delta \vee \Delta_{\neq} \vee \varepsilon_T)$, which concludes the proof of the results. $\qquad \square$