# OpenReview forum: "The price of unfairness in linear bandits with biased feedback"
_NeurIPS.cc/2022/Conference — NeurIPS 2022 Accept_

### Official Review · Reviewer_Vg3M · 2022-07-08

**Rating:** 6
**Confidence:** 4
**Soundness:** 4 excellent
**Presentation:** 3 good
**Contribution:** 3 good

**Summary:**

The authors study the linear bandit problem with biased feedback. Specifically, they assume that there are two distinct groups and that one group receives positively biased feedback and the other negatively biased feedback. The goal is to minimize regret as it is normally defined. They develop a phase-based elimination algorithm that uses ideas from experimental design to correct the bias. They prove matching upper and lower regret bounds and also give gap-dependent upper bounds on the regret.

**Questions:**

How would one adapt the ideas from this paper to a real-world problem?

It would be useful to clarify the technical challenges in this paper.

Minor:

Why in routine 1, does $\pi$ have a support of $\Theta(d^2)$ and in Routine $\hat{\mu}$ has a support of $\Theta(d)$? Doesn't $\Theta(d)$ suffice in both cases?

**Limitations:**

Yes.

**Strengths And Weaknesses:**

Strengths:

The problem setting is well-motivated because it is important to ensure that interactive algorithms are deployed in a fair way.

The worst case regret bounds are nice since they match and are based on a seemingly novel quantity $\kappa^*$.

Weaknesses:

The algorithm is a neat combination of ideas from the literature, but the novelty seems limited. G-optimal design is a standard tool often applied in the linear bandits literature and optimization problem (3) at the heart of $\Delta$-exp-elim has also been considered in
Wagenmaker, Andrew et al. "Experimental design for regret minimization in linear bandits." AISTATS 2021.

While the problem is of considerable practical interest, the work seems primarily theoretical. It is not clear how to go from the proposed algorithm to an algorithm that would be applied in a practical setting. If the authors could explain ways to apply the algorithm in practice and settings where this particular model is suitable, that would improve the paper.

There are no experiments exploring the empirical performance of the algorithm.

The paper could benefit from some proof sketches highlighting the technical difficulties.

Algorithm 4 from the appendix seems complicated and fairly difficult to implement. Guidance for practitioners would be useful.

Post author response: I read the author's response and the other reviews and most of my concerns were addressed.

---

> ### Author Response · Authors · 2022-08-02
> **Answer to Reviewer Vg3M**
>
> **Practical aspects** Linear bandit algorithms are widely used in recommendation systems and online advertisement. Fairness in such online decision-making problems is an important issue. In this paper, we introduce a simple biased linear model to use as a first step in the understanding of these problems. The objective and the main contributions of our work are primarily theoretical: in a situation of biased observations, we want to 1) construct an algorithm that handles biased data and corrects unfair evaluations, and 2) understand the price to pay to correct the bias, and what changes conceptually in the trade-off between exploration and exploitation. On this problem, we are able to characterize precisely the dependence of the regret on the geometry of the action set, which is an open question in most partial monitoring problems.
>
> To do so, we propose a first algorithm of phased elimination adapted for biased observations. Using this algorithm allows us to carefully analyze the regret, and to obtain sharp matching bounds. While it is theoretically optimal, its empirical performances can perhaps be improved (for example by choosing in a less arbitrary manner $\epsilon_l$, which, in phased elimination algorithms, governs the length of the phases and has a strong impact on the performances, or by using algorithms more complex to analyze, such as IDS).
>
> **Implementation** The algorithm can be implemented easily by following its description: the only steps not explicitly written are the computation of the solutions of (2) and (3). To fill this gap, the reader can use optimal design libraries (such as the R package OptimalDesign) together with Lemma 9.  Following the reviewer’s suggestion, we discuss implementation in Appendix A.7 of the rebuttal revision. We underline that Algorithm 4 is only more complex than Algorithm 3 because we need to introduce new quantities required in the mathematical analysis, but not needed in the implementation.
>
> **Relation to [1]** We thank the reviewer for pointing out the interesting reference [1], which will be included in the manuscript. The spirit of optimization problem (3) is indeed related to that of problem (2) in [1]. Nonetheless, these problems differ significantly:
> - The optimization problem (2) in [1] solves a linear bandit problem, and as such could be used in our algorithm to replace the G-Exploration step. By contrast, the optimization problem (3) in our paper solves a bandit problem with *biased observations*. We emphasize that G-optimal design cannot be used to solve this problem. To overcome this issue, we use new results from broader experimental design problems, specifically designed for estimating the scalar product of the parameter with arbitrary vectors.
> - The bounds used to control the error are different, and so are the optimization problems defining the sampling strategy. In Lemma 9, we show that problem (3) can be reduced to a c-optimal design problem on the set of normalized actions. Then, one can use the machinery developed for optimal design to efficiently compute the optimal distribution.
>
> **Technical challenges** Following the reviewer's suggestion, we included sketches of proofs in Appendix C.1 of the rebuttal revision. We list below some of the most significant technical difficulties overcome in this paper:
> - We propose an algorithm for, and characterize the geometrical difficulty of the fair linear bandit problem. We explain how it relates in some sense to a specific instance of partial monitoring, and quantify the price to pay for correcting unfair biases.
> - Our upper bounds require sharp control of the estimated gaps and of the term $\kappa(\hat \Delta^l)$ (which is introduced in this paper), as well as involved arguments to analyze the stopping criterion.
> - We rely on our reduction of $\Delta$-optimal design to c-optimal design on normalized actions, and on c-optimal design theory to construct the hard problem instances needed to derive sharp lower bounds. By contrast, existing lower bounds in linear partial monitoring problems are much looser, and they rely on non-explicit geometric constants.
> - We find that the relevant quantity $\kappa_*$ for characterizing the difficulty of the problem is related to c-optimal design. Building on Elfving’s characterization of c-optimal design, we derive in Lemma 5 an equivalent characterization, which allows us to provide an intuitive, geometrical explanation of $\kappa_*$, and to show its equivalence to the seemingly unrelated “worst-case alignment constant” introduced in [2]. We emphasize that contrary to [2] our bounds are dimension-free.
>
> [1] Wagenmaker, A. et al."Experimental design for regret minimization in linear bandits."AISTATS 2021.
>
> [2] Kirschner, J. et al."Information directed sampling for linear partial monitoring."COLT 2021

---

> > ### Author Response · Authors · 2022-08-02
> > **Answer to the minor remark by Reviewer Vg3M**
> >
> > G-optimal design can be chosen of size $(d+1)(d+2)/2$ : this classical result follows from writing the symetric covariance matrix $V(\pi^*)$ as a barycenter of matrices $a_x a_x^{\top}$ in $\mathbb{R}^{(d+1) \times (d+1)}$, and by using Carathéodory's theorem. By contrast, Elfving's theorem implies that $\kappa_*^{-1/2}e_{d+1}$ is a barycenter of points $a_x \in \mathbb{R}^{(d+1)}$, and thus can be written as a combination of $d+2$ points.

---

### Official Review · Reviewer_XxP7 · 2022-07-10

**Rating:** 6
**Confidence:** 3
**Soundness:** 3 good
**Presentation:** 4 excellent
**Contribution:** 3 good

**Summary:**

The author studies the effect of biased linear bandit feedback on fairness in decision making. They consider a linear bandit problem where each action (a d-dimensional vector) belongs to one of two groups. The grouping is known. The reward is standard, i.e. for each action the reward is inner product of the reward and action vector. However, the observation is biased depending on the group the action belongs to alongside zero-mean stochastic noise. They show this problem is globally observable but not locally observable partial monitoring problem which ensures a O(T^2/3) regret guarantee. More importantly, they derive the exact constant for the regret upper bound scaling as a function of a geometric property of the system. Finally, they prove that the dependence on this quantity is tight by considering some worst case instances.

**Questions:**

- Please try and address the limitation mentioned below.
- Some concrete examples of $\kappa^*$  should be discussed.
- Given that this problem belongs to standard partial monitoring setup, what happens if we apply IDS methods directly? Maybe elaborate on the discussions after Lemma 1.

**Limitations:**

Some more discussion on the relation of this work to the unfairness literature is important. Using the knowledge of the group in algorithm design is typically discouraged in fairness literature, so the authors should address this point with use cases.

**Strengths And Weaknesses:**

Strengths
- The authors introduce (to the best of my knowledge) the biased linear bandits problem where the bias depends on grouping of the action, with known grouping and unknown bias.
- The devised algorithm seems intuitive. A G-optimal design and phased elimination helps narrow down the optimal arm in each group. Whereas, a novel $\Delta$-optimal design and group elimination is employed to estimate the bias parameter, and debias the rewards.
- The authors establish tight minimax lower and upper regret bounds for this problem, while augmenting that with gap-dependent regret upper bounds.
- Lemma 1 relating the term $\kappa^*$ to the geometry of the problem, and Lemma 2 connecting it to the minimax regret for estimating the bias are insightful.

Weakness
- (Major) How the grouping of actions relates to the fairness for end users in a decision making problem is unclear. This mapping is very important for properly motivating the problem. The use of knowledge of the grouping is generally discouraged in fair decision making. The authors tend to deviate from this.
- Examples of how $\kappa^*$ varies in different situation (e.g. action set is the unit sphere, or given a condition number of the action set matrix) should be discussed.
- An example highlighting the need for $\Delta$-optimal design will be insightful. In particular, why can't we simply use the estimates of $\omega$ obtained from the G-optimal design to debias.
- A discussion on the scenario where both bias and grouping is unknown will be interesting to have.  What if we have access to a small subset of actions where grouping is known?
- A discussion on generalization to multiple groups will be useful. Seems like most of the techniques and results will readily extend to that case. How the regret scales with the number of groups, and the $\Delta_{\neq}$  for each group will be interesting to see.

---

> ### Author Response · Authors · 2022-08-02
> **Response to Reviewer XxP7**
>
> **Major concern** We thank the reviewer for raising the interesting question of the awareness of the learner, which we will discuss in the manuscript. As pointed out by the reviewer, the use of the sensitive attribute can be discouraged (it can also be allowed, for example when used for affirmative action). For this reason, some works on statistical fairness avoid using this attribute at the time of prediction (in the unawareness framework); by contrast, our work falls in the awareness framework. We emphasize that in both cases the algorithms are trained on data with known sensitive attributes. In many practical cases, the sensitive attribute (e.g. gender) is indeed known.
>
> Recent works have highlighted critical issues related to unawareness. For example, [1] provide empirical evidence showing that algorithms based on disparate learning processes use non-sensitive features correlated with the sensitive attribute as a proxy for the later, thus inducing within-class discrimination, and leading to sub-optimal trade-off between accuracy and fairness. More recently, [2] study a problem of fair online learning and show that some problems feasible in the awareness framework become infeasible in the unawareness one (such as no-regret learning under demographic parity constraints). These examples, and many others, advocate for the use of the sensitive attribute, as it allows for better fairness guarantees while preventing unfair discrimination based on (possibly irrelevant) non-sensitive features correlated with the sensitive attribute.
>
> **Clarifications** We thank the reviewer for the valuable suggestions, and for pointing out some aspects of our work that we will clarify in a new version of the paper:
> - By contrast to classical complexity measures (such as condition number) that give equal weight to all observations, optimal design gives flexibility to choose $d+1$ best actions to estimate the bias, and therefore allows for sharper bounds. In the rebuttal revision, this topic is discussed in Appendices A.3, and illustration of the behavior of $\kappa_*$ is provided in Appendix A.2.
> - The estimator obtained during the G-optimal exploration phase comes with uniform bounds on the error for estimating $\gamma^{\top}x + \omega^{\top}z_x$, but without bounds on the error for estimating $\omega^{\top} z_x$ itself. If the labels and the actions are very correlated, the error for estimating the bias can be much larger than that for estimating the biased evaluation. An extreme case occurs when there are fewer than $d+1$ good actions left : then, no estimator obtained by sampling these actions can be used to estimate $\omega$. We will underline this point in the manuscript.
> - We will elaborate the discussion after Lemma 1, stating clearly that IDS can be used in our setting, however with sub-optimal worst-case regret bounds, and without guarantee on the gap-dependent regret.
>
> **Extension to more than two groups** We thank the reviewer for suggesting this natural extension to our work. Although it requires slightly heavier notations, it does not present major difficulties compared to the two-groups setting. We discuss in Appendix D of the rebuttal revision how to extend our results to this setting. Due to space constraint, we refer the reviewer to our answer to reviewer R63S, where we summarize the main results.
>
> **Extension to the unawareness framework** Extending the biased linear bandit problem to the unawareness framework is both interesting and challenging. As a first step, we can consider that we know the labels of some actions : in this case, we can start by alternating phases of estimation of the evaluation and of the bias, with decreasing noise level $\epsilon$. The error of the naive LS estimator for $x^{\top}\gamma$ (not taking the bias into account) is the sum of a noise term, plus a term of order $\omega$. As long as $\omega$ is smaller than $\epsilon$, we can inflate the confidence bounds to account for the bias term, and use them to eliminate actions. When $\epsilon$ is of order $omega$, we can estimate the sign of the bias, and use it to recover the unknown labels. Then, we can run the algorithm for the awareness framework.
>
> In the awareness setting, the complexity of the problem depends crucially on the distribution of the labels between the actions in $\mathcal{X}$, and on the separation between the groups. In the unawareness framework, not knowing which action to choose so as to estimate its group would change the difficulty of the problem. From a fairness perspective, estimating the sensitive attributes rather than using them directly could arguably be considered as no better. For these reasons, extending our results to this problem is complex, and beyond the scope of the paper.
>
> [1] Lipton, Z. et al. (2018) Does mitigating ml’s impact disparity require treatment disparity? NeuRIPs 2018
>
> [2] Chzhen, E. et al. (2021). A Unified Approach to Fair Online Learning via Blackwell Approachability. NeuRIPs 2021

---

> > ### Comment · Reviewer_XxP7 · 2022-08-04
> > **Response to Rebuttal**
> >
> > I thank the authors for their response. The response covers most of the questions raised by me.  I agree that the *awareness framework* can be used to motivate the system proposed. However, please make sure the pros and cons of both these frameworks are discussed properly (e.g. where we may want to avoid awareness framework).
> >
> > One concern that still remains mostly unanswered is the following (I clubbed two questions together, and the authors missed one):
> >
> > > How the grouping of actions relates to the fairness for end users in a decision making problem is unclear.

---

> > > ### Author Response · Authors · 2022-08-05
> > > **Clarification**
> > >
> > > We thank the reviewer for giving us an opportunity to clarify a fairness-related aspect of our model, which we will further discuss in the manuscript.
> > >
> > > In this paper, we study a decision-making problem in which the evaluations are unfairly biased against a group of actions (representing for instance individuals described by covariates). This bias can be harmful both for the agent, who can receive a very sub-optimal cumulative reward, due to negatively biased evaluations of optimal actions belonging to the discriminated group, and for the individuals (represented here by the actions), who are systematically discriminated according to their group, instead of being assessed according to their true value $x^{\top}\gamma^*$.
> > >
> > > We underline that in this paper (by contrast, e.g., to the literature on demographic parity), we do not seek to correct for the possibly unequal distribution of features $x$ and values $x^{\top}\gamma^*$ across the different groups. Our approach is related to causal fairness [1]: in the causal fairness framework, the dependencies between prediction, sensitive attributes and non-sensitive attributes are captured by a causal model. The goal is then to ensure that the sensitive attribute does not *directly* influence the prediction (in other words, that conditionally on selected *resolving variables*, the prediction is independent of the sensitive attribute). Here the resolving variables may depend on the sensitive attribute in a manner that is considered as non-discriminatory. For example, one group may have, on average, more physical strength than the other one, and this skill can be considered as fair when it comes to recruit a piano mover.
> > >
> > > The biased linear model studied in this paper is a simple example of causal model with linear structural model equations  $x=f(z,\xi')$ and $y=x^{\top}\gamma^*+\omega^* z+ \xi$, where $\xi$ and $\xi'$ are noise terms: the covariates $x$ may depend on the sensitive attribute $z$, and the biased evaluation $y$ depends on both. In our work, we treat $x^{\top}\gamma^*$ as a fair evaluation of the value of action $x$, since it is independent of $z$ conditionally on the resolving variable $x$.
> > >
> > > [1] Avoiding Discrimination through Causal Reasoning (2017). Kilbertus, N. et al. NeurIPs 2017.

---

> > > > ### Comment · Reviewer_XxP7 · 2022-08-05
> > > > **Rebuttal Addresses the Concerns**
> > > >
> > > > Thanks for the response. Please add these details to the paper, so that the paper is clearly placed in the fairness literature.

---

### Official Review · Reviewer_R63S · 2022-07-11

**Rating:** 6
**Confidence:** 1
**Soundness:** 4 excellent
**Presentation:** 3 good
**Contribution:** 3 good

**Summary:**

The paper considers sequential decision making in a bandit scenario with biased feedback. In the model, the player chooses an action. Each action is associated with covariates and a sensitive attribute, both visible to the player. The reward is a sum of three terms: the intrinsic reward of the action (which is a linear function of the covariates), a bias term, and an independent noise term. The bias term depends on the binary sensitive attribute. The goal of the player is to choose actions sequentially, minimizing the regret compared to the optimal action. The algorithm attempts to de-bias the observations, by deliberately sampling suboptimal actions. The purpose of de-biasing is to determine the group with the best action. Then, G-exploration and elimination is used within that group in order to identify the best action. If the algorithm cannot confidently determine the better group, then an iterative stopping criterion is used. Once the estimated “gaps” between the groups are small enough, the algorithm commits to one group. The theoretical results include upper and lower bounds, as well as gap-dependent bounds.

**Questions:**

It is unclear how the approach generalizes to more than two groups. Could you please comment on this?


**Limitations:**

Yes

**Strengths And Weaknesses:**

•	Interesting model; the results seen significant and they are nearly tight.

•	There is no experiment, which is a shortcoming in a paper proposing an algorithm.

•	There are some writing issues, but the presentation is generally good.

•	The motivation is nice, but could be more explicitly connected to the formal model.

•	The key challenge is knowing how much effort to put into estimating the bias: too little, and you explore the wrong group; too much, and you waste time with the wrong group. This concept is a theme throughout the paper, and is thoroughly addressed from both a theoretic and experimental standpoint.

•	The regret bounds are nearly tight. The gap-dependent bounds are interesting, given how crucial the gaps are in the algorithm.

---

> ### Author Response · Authors · 2022-08-02
> **Response to Reviewer R63S**
>
> We thank the reviewer for the valuable suggestions and comments. In particular, considering a setting with more than two groups is indeed a rather natural extension. Adapting our algorithm to this setting does not present major difficulties compared to the two-groups setting, although it requires introducing slightly more cumbersome notations. We discuss this extension in detail in Appendix D of the rebuttal revision. The main changes in the algorithm as well as the bounds on the regret are summarized below.
>
> **Model**
> We extend the biased linear bandits to $Z$ groups denoted $\mathcal{Z} = \{1, ..., Z\}$. The evaluations are given by
> $$ y_t = x_{t}^{\top}\gamma + Z_x^{\top}\omega,$$
> where $Z_x$ is the $z$-th vector of the canonical basis in $\mathbb{R}^{Z}$ if $x$ is in group $z_x$, and $\omega = \{\omega_1, ..., \omega_Z\}\in \mathbb{R}^{Z}$ is the vector of biases. With no loss of generality, we assume that the model does not contain an intercept, so that it remains identifiable.
>
> **Algorithm** The G-optimal Exploration and Elimination algorithm 1 can be used to estimate and eliminate actions with a given group. By contrast, we must adapt the bias estimation phase. Rather than estimating the bias $\omega_z$, we choose to estimate the differences $\omega_1-\omega_z$ between the bias of group $z$ and the first group, chosen as baseline. Thus, we only need to do $Z-1$ rounds of bias estimation to estimate all differences $\omega_z - \omega_z’$. To estimate $\omega_1 - \omega_z$, we sample actions according to
> $$\mu_z = argmin_{\mu} \sum_x\mu(x)\Delta_x \quad  \text{such that } \quad \left(e_{d+1}-e_{d+z}\right)^{\top}V(\mu)^+\left(e_{d+1}-e_{d+z}\right) \leq  1.$$
> We denote the regret for estimating $\omega_1 - \omega_z$ using actions sampled according to $\mu_z$ as
> $\kappa_{z}'(\Delta) = \sum_x\mu_{z}(x)\Delta_x$ .
> We modify the stopping criterion to adjust to the new cost of bias estimation.
>
> **Bound on the worst-case regret**
> Define $\kappa_* '= \sum_{z>1} \min_{\mu} \left(e_{d+1}-e_{d+z}\right)^{\top}\left(V(\pi)\right)^{+}\left(e_{d+1}-e_{d+z}\right).$ Then, $R_T = O(Z\kappa_*'^{1/3}\log(T)^{1/3}T^{2/3})$.
>
> **Bound on the gap-dependent regret**
> Define $\Delta_{\neq, z} = \min_{x : z_x = z}\Delta_{x}$, $\Delta_{\neq} = \min_{z\neq z^*}\Delta_{\neq, z}$, and $\epsilon_T = \left(\kappa_*'\log(T)/T\right)^{1/3}$. Then,
> $$R_T = O\left(\left(\frac{d}{\Delta_{\min}} + \sum_{z>1, z\neq z^*}\frac{\kappa_z'(\Delta \vee \Delta_{\neq, z}\vee \epsilon_T)}{\Delta_{\neq, z}^2} + \frac{\kappa_{z^*}'(\Delta \vee \Delta_{\neq}\vee \epsilon_T)}{\Delta_{\neq}^2}\right)\log(T)\right).$$

---

> > ### Comment · Reviewer_R63S · 2022-08-08
> > **Reply**
> >
> > Thank you for including the results on the case of more than two groups.

---

### Meta-Review · Area_Chair_VLBe · 2022-08-26

**Recommendation:** Accept
**Confidence:** Certain

**Metareview:**

The authors study a linear bandit problem with biased feedback, develop an algorithm and bound the corresponding regret. The bandit problem they study is meaningful and highly relevant. I therefore recommend to accept the paper.

**Award:**

No

---

### Decision · Program_Chairs · 2022-09-14

Accept